# Summer sea ice floe perimeter density in the Arctic: High-resolution optical satellite imagery and model evaluation

Yanan Wang[1], Byongjun Hwang[1], Adam W. Bateson[2], Yevgeny Aksenov[3], Christopher Horvat[4, 5]

[1]School of Applied Sciences, University of Huddersfield, Huddersfield, HD1 3DH, UK
5  [2]Centre for Polar Observation and Modelling, Department of Meteorology, University of Reading, Reading, RG2 7PS, UK
[3]National Oceanography Centre Southampton, Southampton, SO14 3ZH, UK
[4]Brown University, Providence, USA
[5]University of Auckland, Auckland, NZ

10  *Correspondence to*: Yanan Wang (yanan.wang@hud.ac.uk)

**Abstract.** Size distribution of sea ice floes is an important component for sea ice thermodynamic and dynamic processes, particularly in the marginal ice zone. Recently processes related to the floe size distribution (FSD) have been incorporated in sea ice models, but the sparsity of existing observations limits the evaluation of FSD models, so hindering model improvements. In this study, perimeter density has been applied to characterize the floe size distribution for evaluating three 15 FSD models – Waves-in-Ice module and Power law Floe Size Distribution (WIPoFSD) model and two branches of a fully prognostic floe size-thickness distribution model: CPOM-FSD and FSDv2-WAVE. These models are evaluated against a new FSD dataset derived from high-resolution satellite imagery in the Arctic. The evaluation shows an overall overestimation of floe perimeter density by the models against the observations. Comparison of the floe perimeter density distribution with the observations show that the models exhibit much larger proportion for small floes (the radius < 10–30 m) but much smaller 20 proportion for large floes (the radius > 30–50 m). Observations and the WIPoFSD model both show a negative correlation between sea ice concentration and the floe perimeter density, but the two prognostic models (CPOM-FSD and FSDv2-WAVE) show the opposite pattern. These differences between models and the observations may be attributed to limitations of the observations (e.g., the image resolution is not sufficient to detect small floes), or limitations of the model parameterisations, including the use of a global power-law exponent in the WIPoFSD model, as well as too-weak floe welding and enhanced 25 wave fracture in the prognostic models.

## 1 Introduction

Over the past decades, the extent and concentration of Arctic sea ice have been dramatically declining (Perovich et al., 2020). This results in the changing marginal ice zone (MIZ), defined as the ice-covered region affected by waves and swell by the World Meteorological Organization (WMO, 2014). Another alternative definition of MIZ is a sea ice-covered area with sea 30 ice concentration (SIC) of 15%–80% (e.g., Strong and Rigor, 2013; Aksenov et al., 2017; Rolph et al., 2020; Bateson et al., 2020; Horvat, 2021). Several SIC products are available to define the MIZ, whereas observing waves in sea ice using satellite-

derived observations is still an ongoing area of research. Similarly, sea ice modelling studies often do not include an explicit representation of waves in sea ice. Although this SIC-based definition of MIZ is not directly related to the dynamics in this region (Dumont, 2022), due to the lacking techniques for detecting wave-ice interactions (Horvat et al., 2020), this definition

is widely applied in previous studies. One of the major characteristics of the MIZ is the presence of discrete ice floes in different sizes and shapes, forming the floe size distribution (FSD) (Rothrock and Thorndike, 1984).

Previous studies have suggested the FSD is important for sea-ice processes in the MIZ. The FSD is linked to the total perimeter of the ice floes in the fragmented sea ice field, which is an important parameter influencing the sea ice melt occurring around the side of floes and ocean eddy processes (Steele, 1992; Tsamados et al., 2015; Arntsen et al., 2015; Horvat et al, 2016). Floe

size influences ocean surface heat budget and affects sea ice rheology (Shen et al., 1986; Feltham, 2005; Rynders, 2017), which in turn can affect lead dynamics. The FSD also affects the atmosphere-ocean momentum transfer (Tsamados et al., 2014). In the MIZ, small ice floes (the diameter < 100 m) significantly increase the floe edge contribution to form drag and surface roughness (Steele et al., 1989; Herman, 2010; Lüpkes et al., 2012; Tsamados et al., 2014; Rynders et al., 2018; Brenner et al, 2021). This increases the momentum transfer between the atmosphere and the ocean (Steele et al., 1989; Birnbaum and Lüpkes,

2002; Herman, 2010; Martin et al., 2016). The FSD affects the ocean surface waves propagation and attenuation through the ice, i.e., floes smaller than a characteristic wavelength of swells attenuate wave energy through dissipative processes, while larger floes attenuate the wave energy through scattering (Kohout and Meylan, 2008; Williams et.al., 2013a; Thomson and Rogers, 2014; Montiel et al., 2016; Meylan et al., 2021; Dumas-Lefebvre and Dumont, 2021; Horvat and Roach, 2022).

Given the crucial role of the FSD in various processes within the MIZ, a proper treatment of the FSD-related processes has

become a key issue in simulating sea ice. Recently FSD parameterisations have been incorporated into sea-ice models (Horvat and Tziperman, 2015; Zhang et al., 2016; Bennetts et al., 2017; Rynders, 2017; Roach et al., 2018a; Bateson et al., 2020). For example, current FSD prognostic models consider the FSD evolution driven by thermodynamic and dynamic processes, including lateral melt (Horvat and Tziperman, 2015; Zhang et al., 2015; Roach et al., 2018a), ice ridging and ice fragmentation (Zhang et al., 2015; Horvat and Tziperman, 2015), wave-induced fracture (Horvat and Tziperman, 2015; Bennetts et al., 2017;

Roach et al., 2018a), new ice formation (Roach et al., 2018a), floes welding (Roach et al., 2018a) and brittle fracture (Bateson et al., 2022). Some other modelling studies have assumed a particular shape of FSD (e.g., Bennetts et al., 2017; Rynders, 2017; Bateson et al., 2020). To enhance the understanding of the FSD evolution in various seasons and regions in the Arctic, a wide range of observations from aerial vehicles (e.g., Perovich and Jones, 2014; Toyota et al., 2016), optical satellites, e.g., Landsat (e.g., Rothrock and Thorndike, 1984; Gherardi and Lagomarsino, 2015; Wang et al., 2016), MEDEA (e.g., Denton and

Timmermans, 2021; Hwang and Wang, 2022), MODIS (Toyota et al., 2016; Stern et al., 2018a), and Synthetic Aperture Radar (SAR), e.g., TerraSAR-X (Hwang et al., 2017b; Stern et al., 2018a), have been used to derive floe sizes. Previous observational studies reported a power-law behaviour existing in the tail of FSD, i.e., a straight line in logarithmic axes, leading to the parametrisations of fixing the FSD as a truncated power law (Burroughs and Tebbens, 2001). However, the floe size power-law hypothesis has been contested by recent observations (Steer et al., 2008; Herman, 2010; Herman et al., 2021), which

suggest that different shapes or functions can be better in describing FSDs. Results from laboratory experiments (Herman et

al., 2017; Passerotti, 2022) and models (Herman, 2017; Montiel and Squire, 2017; Mokus and Montiel, 2021; Montiel and Mokus, 2021) indicate that a power-law FSD may be not the most appropriate way to describe the FSD due to wave-induced sea ice breakup.

Accurate model projections of Arctic climate change are needed to guide research and the response to climate change. The development of the FSD models is therefore essential to improve confidence in sea ice models. A major difficulty is the lack of FSD observations, especially high spatial resolution data to constrain the model parameters and evaluate model performance. Hence, we derived a new FSD dataset from 1-m resolution MEDEA imagery and 0.5-m resolution Worldview imagery products and used the dataset to assess the performance of three selected FSD models. The new FSD data can resolve small

floes (up to a few meters), providing a unique opportunity to evaluate the FSD model performance in the Arctic.
In this study, the three FSD models are evaluated against the new FSD dataset. The three models are a diagnostic Waves-in-Ice module and Power law Floe Size Distribution (WIPoFSD) model (e.g., Bateson et al., 2020; Bateson et al., 2022) and two fully prognostic FSD models branched from the FSTD model of Roach et al. (2018a, 2019), hereafter FSDv2-WAVE and CPOM-FSD. The FSDv2-WAVE model is developed by Roach et al. (2019), and CPOM-FSD is a branch of this model

developed by the Centre for Polar Observation and Modelling (CPOM) with additional features. This paper is organized as follows. The study regions are shown in Sect. 2. In Sect. 3, we introduce the FSD models and the new FSD dataset, and the methods applied to process satellite images to derive FSD and the metrics used to evaluate the models are described. Section 4 presents the model evaluation results. The discussion and conclusion are given in Sect. 5.

## 2 Study regions

Two study regions were selected for the model evaluation (Fig. 1). The Chukchi Sea region covers an area of 66°N–80°N, 156°W–180°W (blue box in Fig. 1), and the Fram Strait region covers an area of 77°N–87°N, 20°W–20°E (red box in Fig. 1). These regions are where the model outputs were extracted and analysed. The satellite images were acquired over a small area of 70°N and 170°W in the Chukchi Sea (black dot within the blue box in Fig. 1) and 84.9°N and 0.5°E in the Fram Strait region (black dot within the red box in Fig. 1). The grid cell size of the three models evaluated in this study is $1° \times 1°$, ranging

from 541 km$^2$ for the grid cells near the North Pole to 3802 km$^2$ for the grid cells on the far side of the North Pole in the study regions. Within every grid cell, the external forcing is homogeneous, and the model accuracy is dependent on the accuracy of external forcing. Compared to the satellite observation, much larger model study regions were selected, which could better represent the inhomogeneous effects of external forcing on the evolution of FSD and decrease the effects of external forcing bias in a few grid cells on FSDs. In this way, we minimize the bias caused by a lower resolution model outputs than

observations and ensure that the model outputs include the ice edge, so better representing the mean state of FSD in the models. Although both regions represent early-to-late spring sea ice conditions, the observations from the Chukchi Sea region captures

a more dynamic and fragmented ice condition (e.g., Fig. 1b), while the observations from the Fram Strait capture a less dynamic environment (e.g., Fig. 1c).

## 3 Data and Methods

### 3.1 Observations

#### 3.1.1 Satellite imagery

In this study we use two types of satellite imagery data. The first is 1-m resolution visual-band panchromatic images provided by Measurements of Earth Data for Environmental Analysis (MEDEA) group (Kwok and Untersteiner, 2011; Kwok, 2014). The images were accessed from the Global Fiducials Library (GFL) (http://gfl.usgs.gov/) of the United States Geological Survey (USGS), also known as Literal Image Derived Products (LIDPs). A total of 63 MEDEA images were acquired during May to August over the study period of 2000-2014 at the two fixed locations of the Chukchi Sea and the Fram Strait (Fig. 1). The original MEDEA images of area larger than 225 km$^2$ (Kwok, 2014) were cropped to remove cloud-covered areas and missing data. The size of the cropped images ranges between 30 km$^2$ and 250 km$^2$ (see Table S2 in the supporting information). We also collected one WorldView-1 (WV1) and four WorldView-2 (WV2) images at the Chukchi Sea and Fram Strait sites (Fig. 1), which are collected in a 50 cm panchromatic band (WV1) at 0.52 m spatial resolution (Satellite Imaging Corporation, https://www.satimagingcorp.com/satellite-sensors/worldview-1/, last access: 16 February 2023) and 50 cm panchromatic band and 2 m 4-band Multispectral Bundle (WV2) at 0.47 m and 0.58 m spatial resolution respectively (Satellite Imaging Corporation, https://www.satimagingcorp.com/satellite-sensors/worldview-2/, last access: 16 February 2023). The size of the WorldView (WV) images used in this study is ~40 km$^2$.

#### 3.1.2 FSD retrieval from satellite image

Both MEDEA and WV images were processed to derive the FSD using the algorithm developed by Hwang et al. (2017a). The algorithm combines speckle filtering, Kernel Graph Cutting (KGC) for the segmentation of water and ice regions, distance transformation and watershed transformation, a rule-based boundary revalidation to split ice floes boundaries and final manual validation. The minimum size of floes that can be resolved by the algorithm is dependent on the resolution and type of the images. For 1-m resolution MEDEA images, retrievable floe size ranges between tens of meters to a few kilometres. Small floes with radii less than 5 m can be difficult to resolve due to the limitation in splitting the floe boundaries, so the number of small floes are generally underestimated when applying the algorithm to MEDEA images (Hwang et al., 2017a). For the FSD retrieval, we applied the same filter parameter and KGC algorithm parameter as in Hwang et al. (2017b). The SIC was calculated by counting the number of ice pixels out of the total number of image pixels. Segmented water-ice images were then used to split boundaries of sea ice floes using distance transformation and watershed transformation described by Ren et al. (2015).

### 3.1.3 Sea ice concentration

Two types of SIC products were used in this study: National Snow and Ice Data Center (NSIDC) SIC of 25 km spatial resolution (Meier et al., 2017; Peng et al., 2013) and ARTIST Sea Ice (ASI) algorithm 6.25 km SIC (Spreen et al., 2008;
Melsheimer and Spreen, 2019, 2020). The collected SIC data cover between May and July over the analysis period of 2000–2014. We note that ASI algorithm SIC is unavailable in April- August 2000-2001, April-May in 2002 and April-June in 2012. During these periods, we only applied the NSIDC SIC for the model-observation comparison. The SIC data were extracted for the study areas of the Chukchi Sea and Fram Strait (blue and red boxes in Fig. 1) to compare them with the SIC outputs from the FSD models.

**3.2 Sea ice models with floe size distribution**

In this study, three FSD models are evaluated. An overview of the configuration of these three FSD models is given in Table 1. In Sects. 3.2.1–3.2.3, we will briefly describe the three models and highlight the major differences between them in simulating the FSD related processes.

#### 3.2.1 FSDv2-WAVE model

FSDv2-WAVE model is based on a sub-grid scale floe size and thickness distribution (FSTD) model by Horvat and Tziperman (2015, 2017). It uses the Los Alamos Sea Ice model CICE version 5.1 (Hunke et al., 2015) and is an upgraded version of FSDv2 by coupling with a wave model. Roach et al. (2018a) further implemented this FSTD model into a global ocean-sea ice model on the displaced gx1v6 grid (320x384 for horizontal grid), which is approximately 1-degree resolution horizontally. This is the first global model that simulates emergent floe size evolution by physical processes, including lateral melt/growth,
new ice formation, floes welding, and wave-induced fracture. FSDv2-WAVE uses the slab ocean model (SOM) (Bitz et al., 2012) coupled with the global ocean surface wave model Wavewatch III v5.16 (WAVEWATCH III Development Group, 2016), and incorporates a new wave-dependent ice production scheme (Roach et al., 2019). Attenuation of wave energy in the MIZ is modelled using multiple wave scattering theory developed by Meylan et al. (2021), which accounts for the sea ice floe size, sea ice thickness and sea ice concentration. To address the deficiency of attenuation of wave when wavelengths are much
greater than floe sizes in this theory, an additional attenuation scheme is applied as the reciprocal of wave period squared in FSDv2-WAVE. Among the three selected models, FSDv2-WAVE is the only one that has a fully coupled ocean surface wave model to improve the modelling of wave attenuation, wave-ice interactions and the associated ice thermodynamic/ dynamic processes in the MIZ (Roach et al., 2019).

#### 3.2.2 CPOM-FSD model

CPOM-FSD model is initiated with the ice-free Arctic and run with the tripolar 1° (129 × 104) grids for 37 years from 1 January 1980, including a 10-year period spin-up during 1980–1989 in a pan-Arctic domain excluding Hudson Bay and the

Canadian Arctic Archipelago. CPOM-FSD is adapted from the global FSTD model developed by Roach et al. (2018a, 2019) and built on a modified version of CICE v5.1.2 for sea ice simulation (hereafter referred to as CPOM-CICE) (Hunke et al., 2015; Bateson et al., 2022). CPOM-CICE is an updated version by CPOM at the University of Reading to include (i) a modified

prognostic mixed-layer ocean model to better capture sea ice-ocean feedbacks resulting from lateral and basal melt rate (Petty et al., 2014; Bateson et al., 2022), (ii) a form drag scheme for a better simulation of turbulent heat and momentum fluxes between the sea ice, ocean and atmosphere interface and representing the FSD effects on the form drag scheme (Tsamados et al., 2014; Bateson et al., 2022) and (iii) further amendments to alter maximum meltwater and snow erosion and add the "bubbly" conductivity formulation (Pringle et al., 2007; Schröder et al., 2019). A description of detailed differences between

CPOM-CICE and standard CICE is available in Bateson et al. (2022). CPOM-FSD is not coupled with a wave model but instead retains the internal wave scheme from Roach et al. (2018a) and uses 3-hourly ERA-Interim reanalysis as ocean surface wave forcing. CPOM-FSD incorporates the in-plane brittle fracture and the associated FSD processes, which was shown to improve model performance in simulating the FSD (Bateson et al., 2022).

### 3.2.3 WIPoFSD model

WIPoFSD is a diagnostic power law FSD model within CPOM-CICE (Bateson et al., 2020, 2022) and has the same horizontal grid, spin up and run period as CPOM-FSD. The WIPoFSD model implements the wave-in-ice model (WIM), originally based on the ice-wave interaction theory described by Williams et al. (2013a, 2013b) and updated to coupled ocean–waves–in–ice model NEMO–CICE–WIM at the National Oceanography Centre (NOC), UK (Hosekova et al., 2015; Rynders, 2017; Aksenov et al., 2022). Unlike the two prognostic models (FSDv2-WAVE and CPOM-FSD), WIPoFSD model simulates an FSD

following a power law with a fixed exponent of $\alpha = 2.56$ to constrain the FSD shape over a variable range of floe sizes. The fixed power-law exponent is determined from the satellite imagery data in Bateson et al. (2022), which is derived from MEDEA images acquired at three locations: Chukchi Sea (70°N, 170°W), East Siberian Sea (82°N, 150°E) and Fram Strait (84.9°N, 0.5°E). Bateson et al. (2022) applied a power law fit for the floe size in each image first and then calculated the averaged power-law exponent in each location. The fixed power-law exponent $\alpha = 2.56$ is the mean of these three locations. There are

three differences between the MEDEA imagery data used by Bateson et al. (2022) for calculating the power-law exponent (hereafter Obs1) and in this study for model evaluation (hereafter Obs2). 1) The image resolution of the satellite data used for deriving Obs1 was reduced to 2 m, while the original image resolution (1 m) was kept for Obs2. 2) Obs1 was derived from three sites: the Chukchi Sea, East Siberian Sea and Fram Strait, while Obs2 from 2 sites (the Chukchi Sea and Fram Strait). 3) The Obs1 datasets include 14 cases in the Chukchi Sea, 9 cases in the East Siberian Sea and 12 cases in the Fram Strait. The

Obs2 datasets however include 24 cases in the Chukchi Sea and 32 cases in the Fram Strait. In addition to the exponent, the model also simulates FSD evolution through the floe size parameter $r_{var}$, varying between minimum floe radius $r_{min}$ and maximum floe radius $r_{max}$ in the distribution. $r_{var}$ evolves according to four FSD processes: lateral melt, wave-induced fracture, floe growth in winter and ice advection (Bateson et al., 2020, 2022). The full details of $r_{var}$, including a physical interpretation of this value, are provided in Bateson et al. (2020).

## 3.3 FSD definition and evaluation metrics

In this study, the floe effective radius $r = \sqrt{(a/\pi)}$ is used to define floe size, which is the radius of a circle that has the same area, $a$, as a floe. The FSD is usually defined as the fractional-area distribution, $f(r)dr$, and number-density distribution, $n(r)dr$ (Rothrock and Thorndike, 1984; Toyota et al., 2006; Perovich and Jones, 2014; Horvat and Tziperman, 2015; Zhang et al., 2015; Hwang et al., 2017b; Bateson et al., 2020), corresponding to the area and the number of floes per unit ocean surface area with radius between $r$ and $r + dr$. For the evaluation of FSD models, we consider the perimeter density distribution, $p(r)dr$ (units: km km$^{-2}$), which is proportional to $r \cdot n(r)dr$. Perimeter density distribution defined here as the perimeter of floes per unit ice area with radius between $r$ and $r + dr$. The integral of $p(r)$ between $r_{min}$ and $r_{max}$ in radius is defined as total perimeter density,

$$P = \int_{r_{min}}^{r_{max}} p(r)dr, \tag{1}$$

which is the perimeter per unit ice area of floes between $r_{min}$ and $r_{max}$. Roach et al. (2019) suggested that $P$ is more related to the thermodynamic processes of sea ice floes, which is an important metric in evaluating a FSD model. Additionally, the concept of a perimeter density to characterize the overall floe size has been used in several previous observational studies (e.g., Perovich, 2002; Arntsen et al., 2015), because $P$ reduces the impacts of partially captured floes at the edge of the image for the FSD retrieval (Perovich, 2002; Perovich and Jones, 2014). In this study, $P$ (units: km km$^{-2}$) is used to present comparisons between observations and models. Whilst information about FSD shape is lost when calculating $P$, we also apply $p(r)$ (units: km km$^{-2}$ km$^{-1}$) to evaluate the model performance (e.g., Fig. 2), which can refer to the full perimeter density distribution. There are different ways to calculate $P$. In the following, we describe how the FSD models calculate $P$, as well as how $P$ can be calculated from the observational FSD data, which also show the relationships between number/areal FSD and $P$.

In prognostic FSD models, $P$ is calculated from areal FSD $f_i$ distributed into floe size categories $i$ as follows:

$$P_{prog} = 2\sum_{i=1}^{12} \frac{\gamma f_i w_i}{\pi r_i c_{ice}}, \tag{2}$$

where $r_i$ and $w_i$ are the midpoint and the bin width respectively for each floe size category $i$. Here $c_{ice}$ represents the area-weighted SIC in the selected region. Here $\gamma$ is a floe shape parameter, the ratio of floe perimeter to twice the floe effective radius. For example, $\gamma = \pi$ is for circular floes. From the analysis of MEDEA-derived FSD results, the mean floe shape parameter $\gamma$ is 3.60 in the Chukchi Sea region and 3.69 in the Fram Strait region, which will be used for the calculation of $P$ in this study. Due to the limitation of image resolution in capturing the shapes of small floes, the floes of d < 5 m are discarded for calculating $\gamma$.

$P$ for WIPoFSD can be calculated from

$$P_{wipofsd} = \frac{2\gamma(3-\alpha)(r_{var}^{2-\alpha} - r_{min}^{2-\alpha})}{\pi(2-\alpha)(r_{var}^{3-\alpha} - r_{min}^{3-\alpha})}. \tag{3}$$

In this equation, the power-law exponent was set as a constant, $\alpha = 2.56$, for WIPoFSD.

In this study, we used daily outputs from the FSD models to calculate $P$. To obtain $P$ from the daily model outputs, we calculated an area-weighted mean, on the same date as the observations, over the grid cells within the study areas of the

Chukchi Sea and the Fram Strait (Fig. 1). In supporting information, we note that the $P$ varies depending on the choice of binning and calculation methods (see Sec. S2 and Fig. S1 in the supporting information). To ensure matching with the model outputs, the FSD observation data were binned into the same 12 Gaussian spacing floe size categories used by the FSD models and estimated from areal FSD,

$$P_{obs} = \sum_{i=1}^{12} \frac{2\gamma A_{floe_i}}{\pi r_i A_{ice}}.$$ (4)

$A_{ice}$ is the total area of sea ice within the image. More details on the calculation of $P$ is provided in the Supplementary Materials (S1).

## 4 Results

### 4.1 Model evaluation: perimeter density

The comparison of $P$ between observations and models is shown in Fig. 2. Observations show a substantial difference in $P$ between the two regions (t-test, t (47) = 6.12, p < 0.001) (Fig. 2a). It shows a significantly higher $P$ of 23.76 ± 7.53 km km$^{-2}$ in the Chukchi Sea site than the $P$ of 14.28 ± 4.45 km km$^{-2}$ in the Fram Strait site (Fig. 2a). Higher $P$ in the Chukchi Sea indicates a larger fraction of small floes in that region. It should be noted that the $P$ values in this study are higher than the reported values from previous observational studies, which range between 5.26 km km$^{-2}$ and 13.68 km km$^{-2}$ from May to July in the Beaufort Sea and the Chukchi Sea (Perovich, 2002; Perovich and Jones, 2014; Arntsen et al., 2015). FSDv2-WAVE $P$ values spread out over a wide range and show the opposite regional difference to the observations (a higher $P$ 178.37 ± 89.28 km km$^{-2}$ in the Fram Strait region than the value 136.95 ± 70.58 km km$^{-2}$ in the Chukchi Sea region) (Fig. 2c). WIPoFSD (Chukchi Sea: 120.93 ± 1.66 km km$^{-2}$, Fram Strait: 138.99 ± 12.98 km km$^{-2}$) and CPOM-FSD (Chukchi Sea: 59.55 ± 19.13 km km$^{-2}$, Fram Strait: 61.19 ± 29.95 km km$^{-2}$) also show a general overestimation of $P$ to the observations and the opposite regional difference (Figs. 2b and 2d).

Figs. 2e–2k show the comparison of $p(r)$, i.e., the perimeter of floes per unit sea ice area per unit bin width. The observation results show a declining $p(r)$ with increasing floe radius $r$. The FSDv2-WAVE results show the same relationship but with a steeper slope than the observation, showing a much larger value of $p(r)$ for small floes ($r < 10$–30 m) whilst showing a much smaller value of $p(r)$ for large floes (30–50 m $< r <$ 400–800) than the observations (Figs. 2e–2k). This pattern is consistent in different months and regions. The CPOM-FSD results also show a similar pattern, yet the model $p(r)$ values are in a much better agreement with the observations for large floes ($r > O$ (10) m), especially during July and August in the Fram Strait region (Figs. 2g and 2h). This better match for larger floes has shown to be due to the effects of in-plane brittle fracture (Bateson et al., 2022). The results from the two prognostic models (FSDv2-WAVE and CPOM-FSD) consistently show an 'uptick' (a steepening upward slope in the largest floe size categories) in $p(r)$ (Figs. 2e–2k). This type of 'uptick' in the prognostic models has been reported by Bateson et al. (2022) and Roach et al. (2018a), and is an artificial feature derived from the model setup of an upper floe size limit and an incomplete representation of fragmentation processes in the prognostic model

for large floes. The WIPoFSD results also show a steeper slope than the observation, but a better agreement with the observations than the two other model results. Similar to the two other models, the WIPoFSD model also shows an overestimation of $p(r)$ in small floes ($r < 10$–30 m) (Figs. 2e–2k).

Now we examine the relationship between SIC and $P$. The observation results show a negative relationship between SIC and $P$ (correlation coefficient $r_{cor} = -0.47$, p < 0.01), which means higher $P$ in a lower SIC (i.e., the presence of smaller floes in a lower SIC). A similar relationship was found by Perovich (2002) and Perovich and Jones (2014) in July to September. The WIPoFSD model shows the same negative relationship between SIC and $P$ with stronger correlation ($r_{cor} = -0.78$, p < 0.01), but overall $P$ values are much larger than the observations (Fig. 3). In the Chukchi region, the $P$ values are mostly located within the 'pack ice' region (SIC > 80%) for both observations and WIPoFSD model outputs (Fig. 3a). In Fram Strait, however, the WIPoFSD $P$ values become shifted toward a lower SIC than the observations (Fig. 3b).

The two prognostic models show an opposite correlation to the observations and WIPoFSD results. Both FSDv2-WAVE and CPOM-FSD data show positive relationships between SIC and $P$ ($r_{cor} = 0.35$ and 0.38, p < 0.01) (Fig. 3). In the pack ice region (SIC > 80%), the two prognostic models simulate much higher $P$ than the observations, in particular the $P$ values from FSDv2-WAVE are almost 7–16 times higher than the observations in both study regions (Fig. 3). This indicates a much higher floe fragmentation in the model simulations than the observations in a pack ice condition. In a low ice concentration, the difference becomes smaller, especially for CPOM-FSD (Fig. 3).

### 4.2 Effects of image resolution on the FSD retrieval

In Sect. 4.1, the three models all show larger value of $p(r)$ for small floes ($r < 10$–30 m) than the observations (Figs. 2e–2k). The limited image resolution may hinder the retrieval of small floes. To test this, we investigate $p(r)$ derived from MEDEA ($\delta = 1$ m) images and from WV ($\delta = 0.5$ m) images (Fig. 4). The results show that the $p(r)$ values from the images are in a good agreement for the floes with $r > {\sim}15$ m (Figs. 4e and 4f). This confirms the compatibility of the FSD retrieval from the images with different resolutions. Importantly, however, for the floes with the floe radius $r$ smaller than ~15 m, $p(r)$ derived from the WV image becomes significantly higher than the MEDEA-derived $p(r)$ values (Figs. 4e and 4f). The difference in the $p(r)$ integrated in the two smallest bins ($r < 14.29$ m) between the WV and MEDEA images reaches 1.12 km km$^{-2}$ (Fig. 4e) in the cases in Figs. 4a and 4b and 3.48 km km$^{-2}$ (Fig. 4f) in the cases in Figs. 4c and 4d.

### 4.3 Model evaluation: sea ice concentration

The FSD models considered in this study include parameterisations with dependencies on SIC. For example, the floe welding rate is set to be proportional to the square of SIC in the two prognostic models, FSDv2-WAVE and CPOM-FSD based on the work of Roach et al. (2018b). It is therefore useful to evaluate how well the models simulate the observed SIC and consider the extent to which errors in the simulated SIC could explain the differences between models and observations in simulating floe perimeter density. In the Chukchi Sea, CPOM-FSD shows a good agreement in SIC with the observations (correlation

coefficient $r_{cor} > 0.98$, RMS error $< 7\%$) (Table 2). FSDv2-WAVE, however, shows a considerable bias, underestimating SIC

by 16–17% compared to the observations (Fig. 5a). In the Fram Strait, FSDv2-WAVE better agrees with the observations ($r_{cor} > 0.90$, RMS error $< 7\%$, Table 2) than CPOM-FSD (Fig. 5b). For example, the RMS errors for CPOM-FSD are more than two times larger than FSDv2-WAVE (Table 2). FSDv2-WAVE slightly underestimates SIC by 2–4% compared to the observations in the MIZ (SIC<80%). CPOM-FSD strongly underestimate the SIC by 13%–15% in the MIZ compared to the observations.

This difference in SIC between FSDv2-WAVE and CPOM-FSD can be attributed to different atmospheric forcing that is used in the models (Schröder et al., 2019). FSDv2-WAVE uses JRA55b reanalysis data for the atmospheric forcing, whilst CPOM-FSD and WIPoFSD use 6-hourly NCEP-2 reanalysis data (Table 1). The underestimated SIC from the two prognostic models will result in a too small floe welding rate during spring and early summer. A negative bias in spring SIC shown in the prognostic models may partially explain the overestimation of $P$, and in particular the overestimation of $p(r)$ for small floes

and the underestimation of $p(r)$ for large floes (Fig. 2).

For WIPoFSD model, the evolution of $P$ is constrained by the floe size parameter $r_{var}$ (Eq. 3), which is also impacted by a simple floe growth restoration scheme including floe welding, lateral growth and new ice formation (Bateson et al., 2020). However, this floe growth restoration scheme is not closely related to SIC. In contrast to the other schemes, changes in $r_{var}$ are linked to SIC in the WIPoFSD model via lateral melt, which acts to reduce both (Bateson et al., 2020; Bateson et al., 2022).

For WIPoFSD, SIC decreases 40% in the Chukchi Sea from May to July, similar to the observed decrease (39%) from NSIDC SIC and ASI SIC. In contrast, the SIC for WIPoFSD decreased about 20% in the Fram Strait, 2 times more than the observations (9%). Bateson et al. (2020) conducted a sensitivity study to test the role of lateral melting in affecting FSD by removing the lateral melt feedback on floe size. The results demonstrate that lateral melt is less important in changing the FSD in WIPoFSD. This could explain a smaller discrepancy $P$ between the WIPoFSD model and the observations than the other two models (Fig.

305 2).

## 4.4 Processes controlling floe size distribution evolution

In the prognostic models, FSD evolution is constrained by the parameterized processes. In the period of May–August, the dominant FSD evolution processes are lateral melt and wave-induced breakup, as lateral growth, new ice formation and floe welding are negligible during this season. To test the effects of lateral melt and wave breakup, we acquired two data sets from

model outputs (Fig. 6): monthly changes of $P$ arising from lateral melt and FSD changes arising from wave breakup. The results show that FSDv2-WAVE produces larger, positive changes in $P$ from wave fracture (Figs. 6b and 6d) in summer relative to CPOM-FSD (Figs. 6f and 6h). This indicates that the wave-induced fracture process is much more significant for the floe fragmentation in FSDv2-WAVE than CPOM-FSD. The more significant wave breakup in FSDv2-WAVE may be attributed to the fact that FSDv2-WAVE uses a coupled ocean wave scheme rather than in-ice wave scheme used in CPOM-

FSD model, and that the SIC in the Chukchi Sea is significantly lower in FSDv2-WAVE, which is forced with the JRA55a reanalysis, while CPOM-FSD are forced with the NCEP-2 reanalysis.

CPOM-FSD shows a stronger reduction in $P$ arising from lateral melt (Figs. 6e and 6g) in summer than FSDv2-WAVE. This indicates that the lateral melt process is much more dominant in CPOM-FSD than FSDv2-WAVE. The difference in lateral melt is likely to be related to the difference in sea surface temperature in the models. CPOM-FSD uses a prognostic mixed-
layer ocean model and form drag scheme to simulate ocean mixed-layer properties and the impact of topography of sea ice on sea ice-ocean-atmosphere heat exchange (Bateson et al., 2022). On the other hand, FSDv2-WAVE uses a single ocean layer and ocean heat content diagnosed from a run of Community Climate System Model version 4 (Roach et al., 2019). This difference in ocean components can produce different oceanic heat fluxes in determining the strength of lateral melt between FSDv2-WAVE and CPOM-FSD.

In the northern Chukchi Sea and the Fram Strait (blue boxes in Fig. 6), the change in $P$ arising from processes driving FSD change during May–August is almost zero in the two prognostic models. The $P$ in the northern regions can be regarded as a fixed value over the period of May–August. Our FSD observations lie in the southern part (red box in Fig. 6a) of the Chukchi Sea region, where both models show large changes in $P$ due to wave fracture and lateral melt compared with the northern Chukchi Sea region (blue box in Fig. 6a). For the Fram Strait region, the observation site is located in the northern region
where sea ice floes experience weaker lateral melt and wave fracture (blue box in Fig. 6c). To test the sensitivity of model $P$ between the northern (weak wave fracture and lateral melt) and southern (strong wave fracture and lateral melt), we calculated and compared $P$ from both models between the southern and northern regions of the Chukchi Sea and the Fram Strait. The purpose of this comparison is to explore whether the differences between observations and models arising from the inaccurate summer FSD evolution processes simulated by models (lateral melt and wave fracture) or the processes that determine FSD
shape prior to summer breakup.

As expected, the results show a considerable difference in $P$ between the two regions (Fig. 7). The $P$ values from the northern regions are smaller than the values from the southern regions for the two models (Fig. 7). In the Chukchi Sea region, the SIC values from the northern region are clustered between 90% and 100%, and the $P$ values for both models are comparable to the observation values (Fig. 7a). In the Fram Strait, the SIC values from the northern region spread over a wider range of 50–100%
(Fig. 7b). Interestingly, CPOM-FSD results from the northern Fram Strait region become very comparable to the observations in terms of the $P$ values and the range of SIC (Fig. 7a), while the $P$ values from FSDv2-WAVE still show much larger values (Fig. 7b and Table 3). Note that the observation site in the Fram Strait is located within the northern region. In a direct comparison encompassing a larger model region (Fig. 3), the $P$ values from CPOM-FSD were larger than the observation values (and a positive correlation with SIC). The $P$ in the northern Fram Strait simulated by CPOM-FSD is of the same order
of magnitude of observations, indicating a closer match between CPOM-FSD and the observations in the northern Fram Strait region than the southern region. This suggests that no significant wave fracture and lateral melt has occurred in the observation site. This can be supported by the fact that most of the satellite observations in the Fram Strait represent regions where the sea ice has experienced less thermodynamic and dynamic impacts (e.g., Fig. 1c), so the effects of lateral melt and wave fracture were likely small. It should be noted that CPOM-FSD implements in-plane brittle fracture into the model. Recent studies
suggest that brittle fracture can determine the initial FSD in spring before wave fracture and lateral melt (Gherardi and

Lagomarsino, 2015). Therefore, the close agreement between CPOM-FSD and the observations may represent the initial state of FSD before any significant wave fracture and lateral melt occur. In case of FSDv2-WAVE, the $P$ values from the northern Fram Strait region still show much larger numbers. It is difficult to pinpoint the exact causes of this overestimation as the effects of wave fracture would be quite small in the northern region (Fig. 6d).

## 5 Discussion and conclusion

In this study, we evaluate three state-of-art FSD models (FSDv2-WAVE, CPOM-FSD and WIPoFSD) against new observation data derived from 1-m resolution MEDEA imagery. The observation results show clear regional differences between the two study regions, i.e., much larger perimeter density $P$ in the Chukchi Sea region than in the Fram Strait region. Model outputs, however, fail to show such a regional difference.

The direct comparison between the observations and daily model outputs reveals that the models consistently show (i) overall overestimation of $P$, (ii) much larger value of the $p(r)$ for small floes (r < 10–30 m) and (iii) much smaller value of the $p(r)$ for the larger floes (30–50 m < r < 400–800 m). Among the three FSD models, WIPoFSD and CPOM-FSD show a much smaller difference to the observations than FSDv2-WAVE. The observations and WIPoFSD model both show a negative correlation between SIC and $P$ (i.e., smaller floes in a lower SIC), while the two prognostic models show the opposite (positive) correlation. The causes of such differences include (i) differences in the coverage area of study regions between models and observations, (ii) the limitations within the observations such as the image resolution, (iii) underestimation of SIC and the associated effects on floe welding parameterisation, and (iv) too much wave fragmentation in the models, as suggested by Cooper et al. (2022).

The satellite observation sites are fixed and much smaller than the regions selected for models, which include grid cells in both MIZ and interior pack ice. Nevertheless, the observations also include a range of different sea ice conditions. For example, the sea ice concentration of half of the cases in the Chukchi Sea (Figure 2i–2k) is below 80%, indicating almost half of the images within the MIZ and the other half within the interior ice pack. Therefore, the monthly means from the models (as the solid lines in Figure 2e-2k) can be comparable to the observations in the Chukchi Sea. In the Fram Strait, most of the observations represent the pack ice (>80% SIC), while the model outputs mainly represent the MIZ conditions (<80% SIC). To evaluate the effects of the size of modelling study regions, we reduced the size of the model domains down to a 5×5 grid-cells region at the centre of the observation sites, which shows the same overestimation of $P$ and $p(r)$ of small floes (see Figure S2 and Table S3). Based on this evaluation, we decided to use a larger model domain to better capture spatial variability caused by different external forcing fields in the models.

The effects of the limited image resolution are examined by comparing (1-m resolution) MEDEA-derived $p(r)$ with (~0.5-m resolution) WV-derived $p(r)$. It shows that WV-derived $p(r)$ integrated for the two smallest bins ($r$ < 14.29 m) is 1.12 to 3.48 km km$^{-2}$ larger than the value derived from MEDEA. However, this difference is still too small to explain the difference between the observations and model outputs, varying between 20.42 km km$^{-2}$ and 218.95 km km$^{-2}$ in Figs 2e–2k (See Table

S1). We do not know how much additional change we would see in $P$ and $p(r)$ if we had access to imagery at even higher resolutions. This suggests that based on the recent satellite imagery, the image resolution could be one of the contributors but is not the main contributor to the overestimation of modelled $p$ of small floes. It is also worth noting that both 1-m and 0.5-m resolution observations show two distinct regimes of $p$ for small floes ($r < O\,(10)$ m) and the larger floes (Figs 2e-2k, 4e and 4f). This situation is associated with several possible reasons: (i) Image resolution is not high enough to identify all small floes accurately; (ii) Lateral melt reduced the number of small floes (Hwang and Wang, 2022); (iii) Other statistical models, e.g., log-normal distribution, is better to describe the FSDs rather than power laws (Montiel and Mokus, 2022; Mokus and Montiel, 2022). It requires much higher resolution images (e.g., aerial photographs) and further research in the future to properly investigate the effects of the image resolution and the reasons of this deviation of the small floes perimeter density distribution. The strength of floe welding is strongly contributed from the SIC in the prognostic models evaluated in this study (Roach et al., 2018a, b; Bateson, 2021; Bateson et al., 2022). Previous studies have identified the dominant role of floe welding in the formation processes of large floes (Toyota et al., 2011; Roach et al., 2018a, b). In particular, Bateson (2021) has assessed the effects of floes welding on the $p(r)$ in the CPOM-FSD model, suggesting that floe welding occurring in spring can influence the FSD in summer. A low ice concentration reduces the floe welding during spring and consequently results in an initial over-fragmented state in early summer. Therefore, a negative bias in spring SIC shown in the prognostic models can partially explain the overestimation of $p(r)$ for small floes and the underestimation of $p(r)$ for larger floes in the two prognostic models (Fig. 2).

For WIPoFSD, the bias in the $p(r)$ is not related to the underestimation of SIC and the consequent floe welding parameterisation. Instead, the bias in $p(r)$ is likely due to the fixed power-law exponent of $\alpha = 2.56$ for non-cumulative distribution used in the model. This value is larger than the exponent from our dataset (i.e., in the Chukchi Sea $\alpha = 2.34$, in the Fram Strait $\alpha = 2.07$). Previous studies have found the exponent ranges vary seasonally and regionally (Stern et al., 2018a, b). The typical exponent value ranges from about 1.8 to 3.6 for non-cumulative distribution in the Chukchi Sea and the Beaufort Sea during May–August (Holt and Martin, 2001; Wang et al., 2016; Hwang et al., 2017b; Stern et al., 2018a, b) and from 2.0 to 2.8 (non-cumulative distribution) in the Fram Strait in June (Kergomard, 1989). Thus, we suggest that employing a seasonally and spatially variable exponent in the model may improve the model performance.

In terms of overactive wave fracture in the prognostic models, the wave fracture model applied by Horvat and Tziperman (2015) and Horvat and Roach (2022) has been implicated as producing unrealistically fragmented FSDs in the Chukchi Sea (Cooper et al., 2022). As wave events episodically propagate hundreds of kilometres into the sea ice, the impact of this oversensitivity may be to produce unrealistically high perimeter densities in our study regions. To investigate this, we examined the $P$ in the northern regions where wave-induced breakup is negligible. In these regions, most modelled $P$ match our observations better. However, we found that the $P$ from FSDv2-WAVE still show positive bias in the Fram Strait region. These biases may be attributed to the initially over-fragmented ice conditions in early spring set in the models.

In conclusion, the new FSD dataset was found to be valuable in evaluating the FSD models, which shows considerable differences from the observations in terms of $P$ and the relationship between $P$ and SIC. The summer $P$ change in the models

depends strongly on initial floe size distribution before melting starts, which is affected by floe formation and growth processes (e.g., the welding of meter-scale floes) in the models. Our findings also indicate positive biases of $P$ are closely linked to overactive wave fracture in the models. This suggests accurate parameterisation of wave-induced sea ice breakup is essential for simulating the summer FSD correctly. It should be noted that both the prognostic FSD model and power-law FSD model are still in development, as are methodologies to determine floe size from satellite imagery. In addition, the ability to resolve the shape of the FSD for small floes remains constrained by the limited resolution of satellite images. Nevertheless, this study shows how model evaluation using such imagery can be used to produce key insights for model development, thus allowing us to improve sea ice model performance for climate research and operational applications.

## Data availability

MEDEA images are openly available at the Global Fiducials Library website (http://gfl.usgs.gov/). WorldView images cannot be shared due to the license. However, the images can be ordered from LAND INFO Satellite Imagery Search Portal (https://search.landinfo.com/) or other satellite imagery providers. FSD imagery data retrieved from satellite imagery in this study can be accessible from UK Polar Data Centre soon (DOI will be generated before the publication of this paper). The model outputs used for analysing the monthly change of FSD in the study are available from https://doi.org/10.5281/zenodo.3463580 for FSDv2-WAVE and from http://dx.doi.org/10.17864/1947.300 for CPOM-FSD and WIPoFSD. The daily model outputs used in this study for model-observation comparison will be available before the publication of this paper.

## Author contribution

YW conceived the study under the supervision of BH, YA and CH. YW prepared the FSD observations with the support from BH. AB completed simulations of CPOM-FSD and WIPoFSD and shared model outputs from Bateson et al. [2022]. YW performed the data analysis and completed the comparison of the FSD observations to model output, with support from BH, AB, YA and CH. YW prepared the manuscript with guidance and contributions from all authors.

## Competing interests

Yevgeny Aksenov is a member of the editorial board of The Cryosphere. The peer-review process was guided by an independent editor, and the authors have also no other competing interests to declare.

**Acknowledgments**

YW and BH are funded by UK Natural Environment Research Council (NERC) under the project *Toward a Marginal Arctic Sea Ice*, grant NE/R000654/1, and by the NERC project *Arctic Sea Ice Breakup and Floe Size during the Autumn-to-Summer Transition (MOSAiCFSD)*, grant NE/S002545/1 to generate FSD observation. AB is funded by NERC, reference NE/R016690/ and NE/R000654/1. YA acknowledges support from NERC grants NE/N018044/1 (the North Atlantic Climate System Integrated Study LTS-M Programme ACSIS), NE/R000085/1, NE/R000085/2, and NE/T000260/1 and from NERC National Capability CLASS (Climate Linked Atlantic Sector Science LTS-S Programme), grant number NE/R015953/1. CH was supported by NASA grant 80NSSC20K0959 and by Schmidt Futures. We are grateful to Lettie Roach for providing daily model outputs from the FSDv2-WAVE model described by Roach et al. [2019].

**Financial support.**

This research has been supported by the UK Natural Environment Research Council (grant NE/R000654/1, NE/S002545/1, NE/R016690/, NE/N018044/1, NE/R000085/1, NE/R000085/2, NE/T000260/1 and NE/R015953/1), NASA (grant 80NSSC20K0959), Schmidt Futures and University of Huddersfield.

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

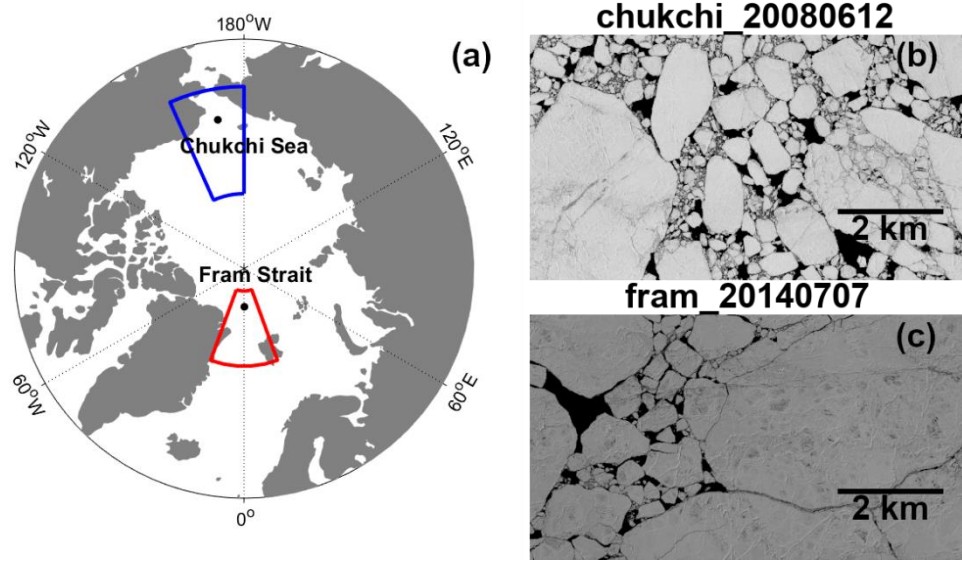


**Figure 1: (a) Map of the study regions. Satellite images acquired on (b) 12 June 2008 in the Chukchi Sea and (c) 7 July 2014 in the Fram Strait. The blue and red boxes are the boundary of the Chukchi Sea region and the Fram Strait region respectively. The black dots within the study regions mark the locations where satellite imagery data were acquired (70°N and 170°W in the Chukchi Sea and 84.9°N and 0.5°E in the Fram Strait).**


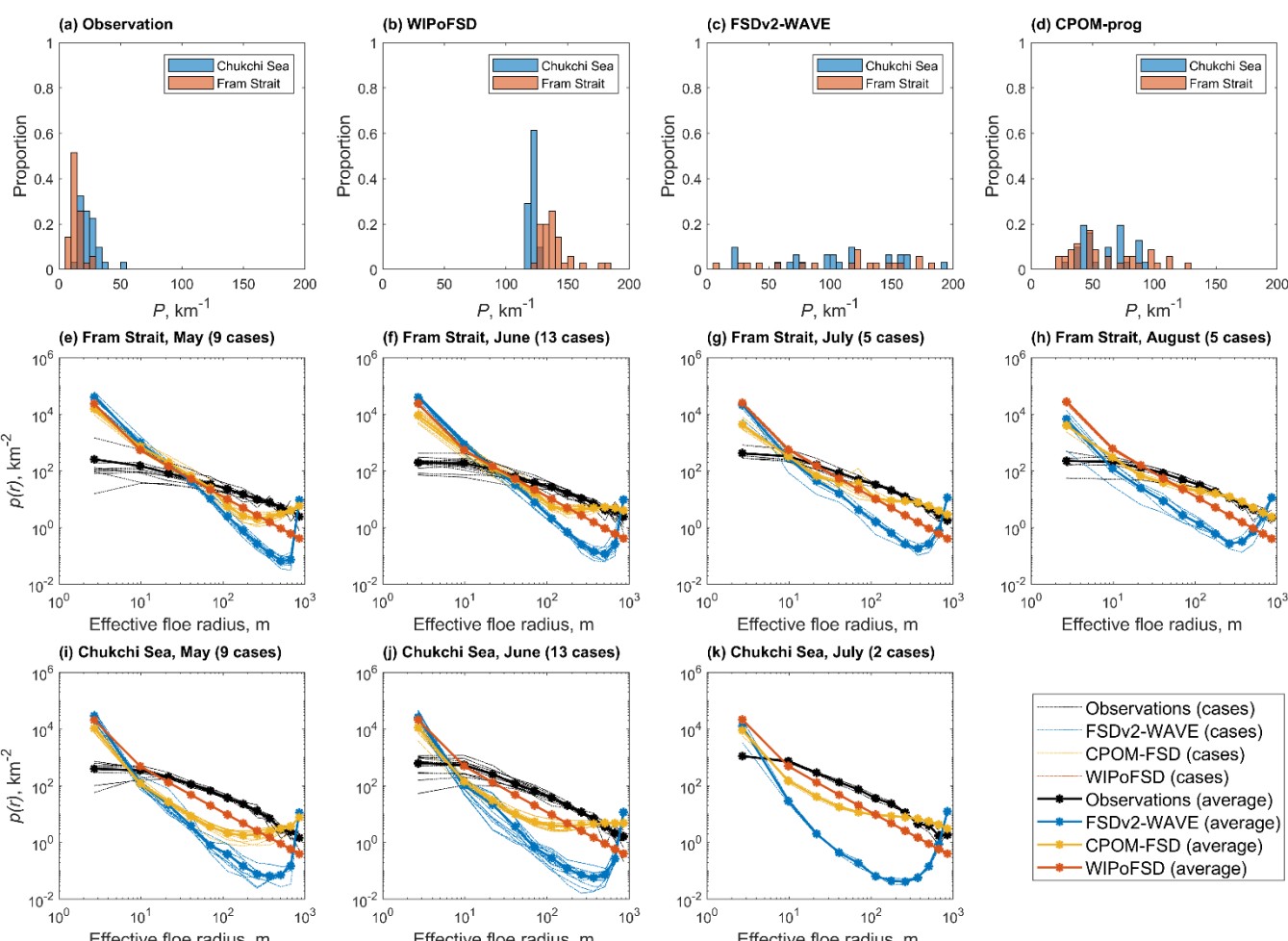

**Figure 2: Frequency histograms of floe $P$ from (a) observation, (b) WIPoFSD, (c) FSDv2-WAVE and (d) CPOM-FSD. In (a)–(d), blue colour indicates the frequency distribution of $P$ for the Chukchi Sea and red colour for the Fram Strait. $p(r)$ are shown for (e) May, (f) June, (g) July and (h) August in Fram Strait, as well as for (i) May, (j) June and (k) July in the Chukchi Sea. In (e)–(k), the observations are shown in black line and three models in different colours (FSDv2-WAVE—blue, CPOM-FSD—yellow, WIPoFSD—red). In (e)–(k), dash lines correspond to the frequency distribution of $P$ of individual cases in each month and solid lines are the mean of them.**

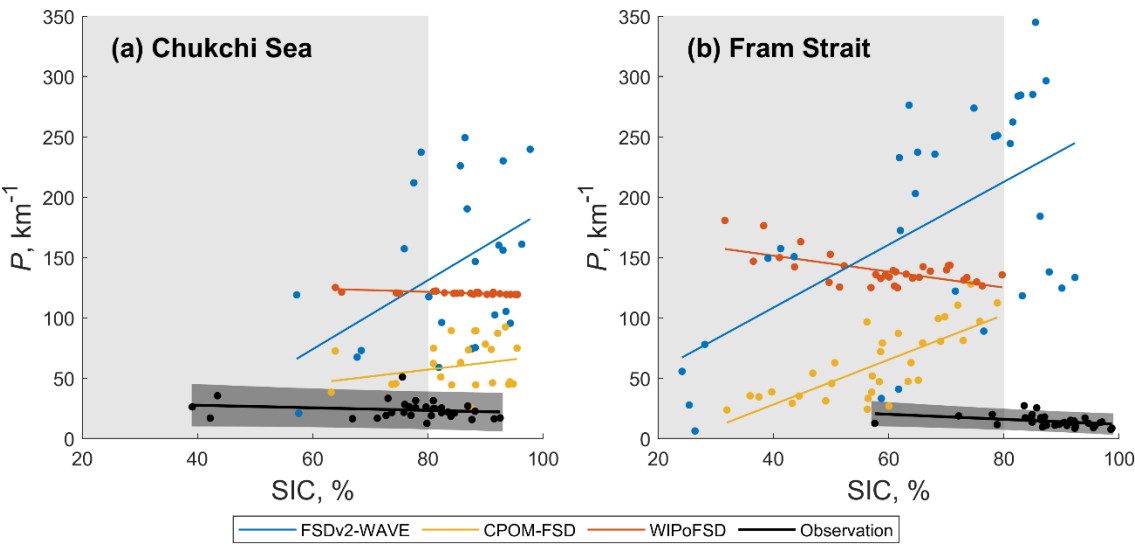

**Figure 3: Perimeter density *P* according to SIC for (a) the Chukchi Sea region and (b) the Fram Strait region. Dark grey shades along the regression lines of the observations mark a 95% confidence interval. The light grey shade marks the MIZ, defined as SIC between 15% and 80%.**

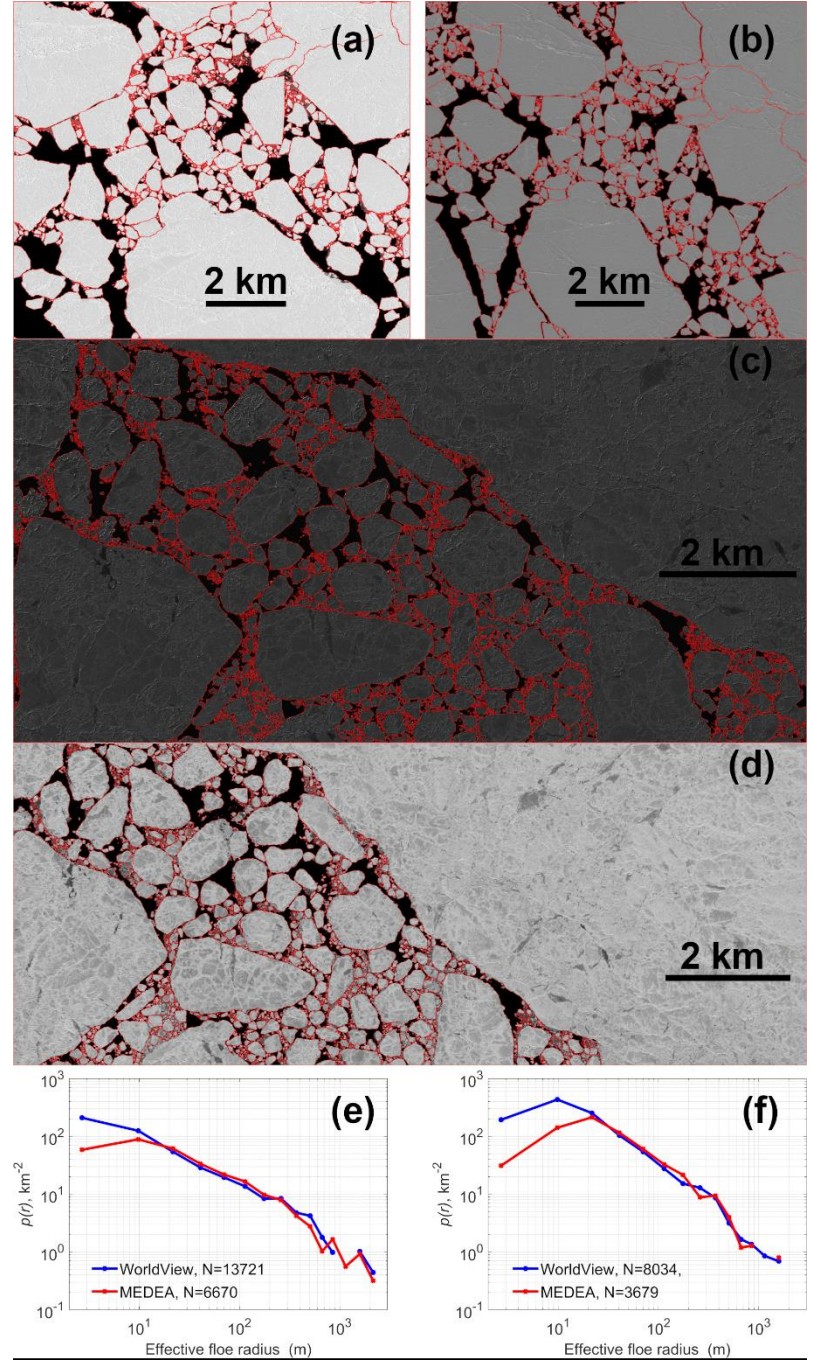

Figure 4: Comparison of perimeter density distribution $p(r)$ between (a) WV© image ($\delta$ = 0.5 m) and (b) MEDEA image ($\delta$ = 1 m) on 5 June 2013 at 84.9°N, 0.1°E (Fram Strait) is shown in (e) and between (c) WV image ($\delta$ = 0.5 m) on 1 June 2013 and (d) MEDEA image ($\delta$ = 1 m) on 31 May 2013 at 70°N, 170°W (Chukchi Sea) is shown in (f). The image size of the co-located scenes shown in (a), (b), (c) and (d) cover an area of 106 km², 82 km², 66 km² and 64 km² respectively. In panels (e)–(f), $N$ is the number of floes derived from satellite images.

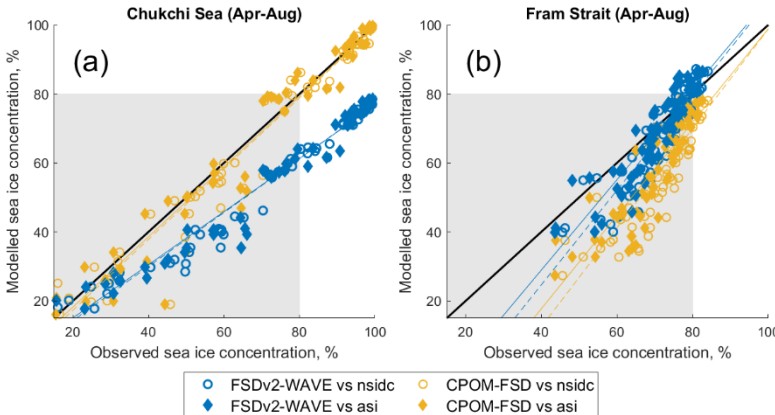

**Figure 5: A comparison of SIC between observations and prognostic models in the Chukchi Sea (a) and (b) the Fram Strait. Monthly SIC data from April to August for the period 2000–2014 were used for the comparison. In (a)–(b), the comparison between the observations and two prognostic models are shown in different colours (FSDv2WAVE: blue, CPOM-FSD: yellow). The comparison between NSIDC and models are marked with circles and their linear fits are shown as dashed line. Diamonds and solid lines indicate the comparison between ASI and models.**


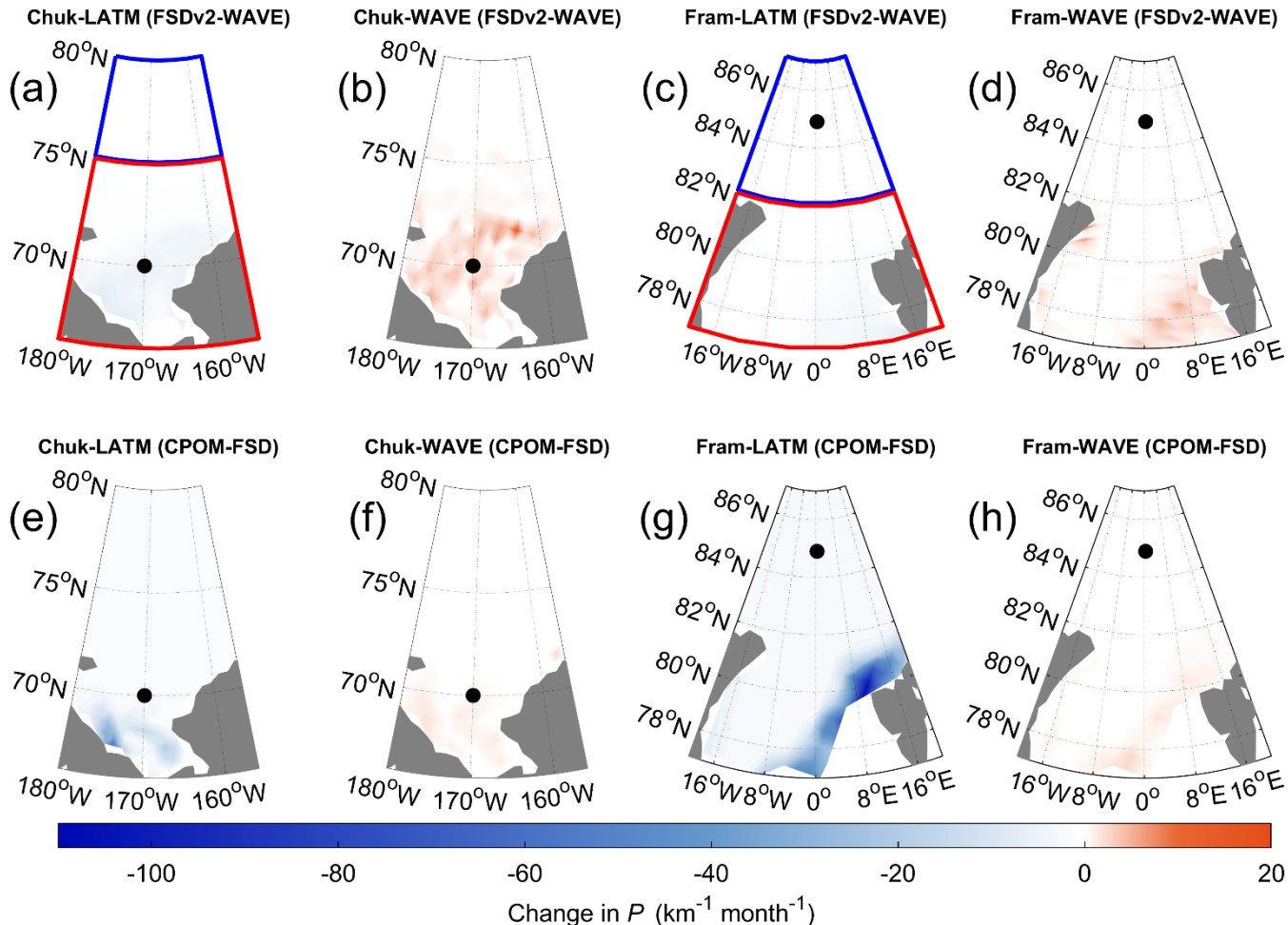

**Figure 6: Monthly changes of *P* simulated by the two prognostic models over the period May to July during 2000–2014. (a) Change of *P* arising from lateral melt for FSDv2-WAVE in the Chukchi Sea. (b) is same as (a) but for wave induced *P* change. (c) and (d) are same as (a)and (b) but in the Fram Strait. (e)–(h) is same as (a)–(d) but for CPOM-FSD. The blue and red box in (a) and (c) show the northern and southern region of the two study regions. Black dots indicate the location of observations in the study regions.**


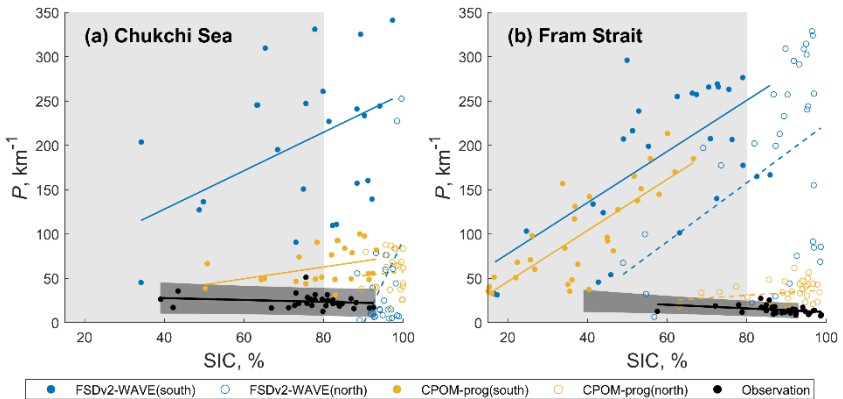

**Figure 7: Similar to Figure 3 but show the comparison between observations (black) and two prognostic models (FSDv2-WAVE— blue, CPOM-FSD—yellow) in southern regions (solid circle) and northern regions (hollow circle) of the study region in the Chukchi Sea (a) and the Fram Strait (b).**

**Table 1. Summary of model simulations used in this study.**

| Simulation | Sea ice model | Ocean coupling | Atmosphere forcing | Wave forcing | Grid | Run period |
|---|---|---|---|---|---|---|
| **FSDv2-WAVE** | CICE v5.1 | SOM[a] | 6-hourly Atmospheric reanalysis JRA55[c] | Coupled Wavewatch III v5.16[e] | Displaced pole 1° (320 x 384) | 2000–2014 |
| **CPOM-FSD** | CICE v5.1.2 | Mixed layer ocean model[b] | 6-hourly NCEP-2 reanalysis[d] | 3-hourly ERA-Interim reanalysis[f] | Tripolar 1° (129 × 104) | 1980–2016 |
| **WIPoFSD** | | | | | | |

[a] Slab Ocean Model (SOM) (Bitz et al., 2012)

[b] Petty et al. (2014)

[c] Japan Meteorological Agency (2013).

[d] Kanamitsu et al. (2002).

[e] WAVEWATCH III Development Group (2016).

[f] Dee et al. (2011)

**Table 2. Statistical summary for the two prognostic FSD models against the NSIDC SIC and ASI SIC. NSIDC and ASI SIC data used for the comparison are between April and August for the analysis period of 2000–2014.**

| | Correlation coefficient | | | | RMS error | | | |
|---|---|---|---|---|---|---|---|---|
| | NSIDC | | ASI | | NSIDC | | ASI | |
| | CS[a] | FS[b] | CS | FS | CS | FS | CS | FS |
| **FSDv2-WAVE** | 0.99 | 0.91 | 0.98 | 0.90 | 18% | 7% | 18% | 7% |
| **CPOM-FSD** | 0.98 | 0.86 | 0.98 | 0.86 | 6% | 16% | 7% | 14% |

[a] Chukchi Sea.

[b] Fram Strait.

**Table 3. The mean $P$ (km km$^{-2}$) and standard deviation simulated by FSDv2-WAVE and CPOM-FSD in the southern regions and northern regions of the Chukchi Sea (CS) region and the Fram Strait (FS) region.**

|  | Southern CS | Northern CS | Southern FS | Northern FS |
|---|---|---|---|---|
| **FSDv2-WAVE** | 197.50±85.80 | 44.12±67.99 | 188.51±83.32 | 179.18±104.69 |
| **CPOM-FSD** | 61.39 ± 20.11 | 55.84±22.47 | 98.93±51.34 | 33.71±12.23 |
