# Peer review of "Summer sea ice floe size distribution in the Arctic: High-resolution optical satellite imagery and model evaluation"

_The Cryosphere, 2022_

## Author Comment (AC1)

Dear Referee,

We would like to thank you so much for reviewing our manuscript and giving valuable comments and suggestions. Please find our responses to your comments in blue text and revised manuscript in red text below.

Best regards,

Yanan Wang and co-authors

**Response to Anonymous Referee #1**

This paper compares the sea-ice "floe perimeter density," as calculated from three models, to satellite observations in the Chukchi Sea (CS) and Fram Strait (FS).

The length, or density (length per unit area), of floe perimeter is a factor in the lateral melting of ice floes in summer, and is therefore a potential diagnostic for models. For a given field of ice floes, the floe perimeter density is a scalar.

The sea-ice floe size distribution (FSD) is the number of floes as a function of floe size. The FSD may be normalized (e.g. by the total number of floes) or not.

The analysis in this paper is all about perimeter density, denoted P_i (P sub i) by the authors, and PD by this reviewer. However, the authors treat PD and FSD as if they are interchangeable or equivalent. They are not. Completely different FSDs can give rise to the same PD, and identical FSDs can give rise to different PDs. There is not a one-to-one correspondence between PD and FSD. The authors point out that a larger PD implies more smaller floes, and this is true, but the PD says nothing about the FSD. In light of this fundamental confusion between PD and FSD, I must recommend that this paper be rejected. Specific comments follow.

PD and FSD

==========

The title implies that the paper is about the FSD, but it is really about the PD.

In the Abstract, lines 11-16 are about the FSD, and lines 17-25 are about the PD, without making any connection between the two.

Suppose n(r) is the number of floes of size r. Consider the case of circular floes. The perimeter of each floe is 2*pi*r and the area is pi*r^2. Therefore the perimeter density is:

   PD = INTEGRAL(2*pi*r * n(r)dr) / INTEGRAL(pi*r^2 * n(r)dr)

Now suppose the mean value of n(r) is MU, and the variance is SIGMA^2. Then the above equation yields:

   PD = 2*MU / (SIGMA^2 + MU^2)

Now consider two cases:

(1) n(r) has a uniform distribution on [0,L], i.e. n(r) = 1/L.

Then MU = L/2 and SIGMA^2 = (L^2)/12, so PD = 3/L.

(2) n(r) has an exponential distribution with parameter LAMBDA,

i.e. n(r) = (1/LAMBDA) * exp(-r/LAMBDA).

Then MU = LAMBDA and SIGMA^2 = LAMBDA^2, so PD = 1/LAMBDA.

   By choosing L = 3*LAMBDA, the uniform FSD has the same perimeter density as the exponential FSD.  Same PD, different FSDs.

   Now consider a set of circular floes with FSD n(r).  Construct a set of elliptical floes with semi-major axis "a" and semi-minor axis "b" such that pi*a*b = pi*r^2.  Each elliptical floe has the same area as its corresponding floe in the circular set.  Therefore the FSD of the elliptical set is also n(r), by construction.  But the perimeters of the elliptical floes are longer than the perimeters of the circular floes, so the PD for the elliptical set is larger than the PD for the circular set.  Same FSD, different PDs.

Same number FSD, different area FSD.

Lines 359-361.  "positive biases of P_i are closely linked to overactive wave fracture in the models.  This suggests accurate parameterisation of wave-induced sea ice breakup is essential for simulating the summer FSD correctly."

The implication here (and throughout the paper) is that the PD tells us about the FSD.  But a connection between PD and FSD has not been demonstrated, and the simple theoretical examples in the previous comment show that a connection need not exist.

Figure 6 caption.  "(a) Change of FSD arising from lateral melt" and "(b) ... wave induced FSD change"

According to the scale bar in the figure, the panels show the change in perimeter density, not the change in FSD.  But here (and throughout the paper) the authors seem to equate PD and FSD.

In summary, the authors have not said how the PD is related to the FSD, and therefore why it can be used to assess the FSD produced by the models.  According to my calculations, the PD and FSD are not necessarily related, so any statements or conclusions derived from the analysis of the PD do not necessarily apply to the FSD.  Since there is no easy way to rectify the confusion between PD and FSD in this paper, it should be rejected.

Thanks for your comments. Regarding your comment 'the PD and FSD are not necessarily related' and 'there is no easy way to rectify the confusion between PD and FSD in this paper', we disagree with these points and have different opinions.

1) In previous studies, the FSD is not only presented as number density. Perimeter density distribution and floe area fraction are also used. Rothrock and Thorndike (1984) first defined the FSD as both number density n(r) and area fraction f(r). Yes, perimeter density distribution is not defined as FSD by Rothrock and Thorndike (1984), but it is a useful proxy for FSD and widely used in previous FSD studies. For example, previous observational studies (Perovich, 2002; Perovich and Jones, 2014 and Arntsen et al., 2015) applied the total floe perimeter per unit ocean area as a proxy for floe size distribution. As discussed by Perovich (2002), the use of the perimeter of the sea ice floe reduces the impacts of partially captured floes at the edge of the image for FSD retrieval. Besides, perimeter density per unit ice area is also used in FSD models as a metric to show the evolution of the FSD, e.g., Roach et al. (2019) and Bateson et al. (2022). As explained by Roach et al. (2019), "$P_{ice}$ is weighted more heavily by smaller sizes, so Pi is more relevant for thermodynamic melting and freezing of floes", which is important in evaluating the FSD model.

2) The same PD corresponding to different number FSDs or the same FSD corresponding to different PDs does not mean there is no connection between PD and FSD. In previous studies, Rothrock and Thorndike (1984) first defined the FSD as both number density n(r) and area fraction f(r). These two definitions are related by $f(r) = \gamma r^2 n(r)$. Now, if we consider a set of circular floes and a set of elliptical floes with semi-major axis "a" and semi-minor axis "b" with the same number FSD n(r) in a given region. But assume that pi*a*b ≠ pi*r^2. In this situation, although these two groups of floes have the same n(r), their areal FSD f(r) is different. Similarly, if we consider 5 circular floes with the same radius of 10 m and 100 elliptical floes with semi-major axis "a=1 m" and semi-minor axis "b = 5 m" in a given region. Then we can get the same areal FSD, different number FSDs. Even the traditional FSD concepts n(r) and f(r) still shows the same situation that there is not always a unique 1:1 relationship between them. Similarly, it is not abundant evidence to prove that Pi is not a metric related to FSD by showing the same PD corresponding to different number FSDs or the same FSD corresponding to different PDs.

3) Although your example and ours given above means that there is not always a unique 1:1 relationship between the number density distribution and perimeter density distribution or between the number density distribution and area fraction distribution. It is worth noting that most examples are related to the variable shape of floe, i.e., γ is not the same for the two group of circle and elliptical floes given in these examples. However, it is common in FSD studies to make assumptions about floe shape, e.g., FSD models assume a fixed shape parameter γ. This should not be a cause for concern.

4) Additionally, the reason for the same PD corresponding to different number FSDs or the same FSD corresponding to different PDs can also be partially explained by equations (4)-(11) in Roach et al. (2019). They define the notation $\langle r^i \rangle_N$ as the ith moment of the number FSD.

    The fractional sea ice concentration in a grid cell, c, is
$$c = \int_{r_0}^{r_{max}} \gamma r^2 n(r) dr = \gamma \langle r^2 \rangle_N.$$
    then the perimeter density per unite ocean area is

$$P_{ocean} = \int_{r_0}^{r_{max}} 2\gamma r n(r) dr = 2\gamma \langle r^1 \rangle_N,$$

The perimeter density per unite sea ice area is

$$P_{ice} = \frac{P_{ocean}}{c} = \frac{\int_{r_0}^{r_{max}} 2\gamma r n(r) dr}{\int_{r_0}^{r_{max}} \gamma r^2 n(r) dr} = 2\frac{\langle r^1 \rangle_N}{\langle r^2 \rangle_N}$$

Different metrics have different emphases. The area fraction of floe $F(r) = \int_{r_0}^{r_{max}} f(r) dr = \int_{r_0}^{r_{max}} \gamma r^2 n(r) dr = \gamma \langle r^2 \rangle_N$, is a positive moment of the number FSD. For $P_{ice}$, Roach et al. (2019) suggest that "$P_{ice}$ is a negative moment of the FSD, weighted more heavily by smaller sizes, so Pi is more relevant for thermodynamic melting and freezing of floes", which is important in evaluating a FSD model.

Whilst this paper may benefit from some tightening of nomenclature and clarification, useful information about the FSD can still be obtained from the perimeter density. For example, in Figure 2, we explicitly present the perimeter density distribution for floe size categories (perimeter of floes in floe size category i per unit bin width per unit sea ice area) rather than just perimeter density Pi, which therefore provides additional information about whether it is smaller floes or larger floes that contribute to differences in Pi calculated across the distribution overall.

5) The reason why we choose perimeter per unit sea ice area rather than per unit ocean area as in previous observational studies (Perovich, 2002; Perovich and Jones, 2014; Arntsen et al., 2015) can be explained by an example. If we consider a circular floe with a radius r = 5 km in a 10x10 km region. Sea ice concentration $SIC_1 = \frac{\pi}{4}$, $P_{ocean1} = \frac{\pi}{10} km \cdot km^{-2}$ and $P_{ice1} = 0.4 km \cdot km^{-2}$. We also consider 50 circular floes with radius r = 0.1 km in a 10 x10 km region. $SIC_2 = \frac{\pi}{200} < SIC_1$, $P_{ocean2} = P_{ocean1} = \frac{\pi}{10} km \cdot km^{-2}$ and $P_{ice2} = 20 km \cdot km^{-2} > P_{ice1}$. Obviously, perimeter per unit sea ice area is more related to the distribution of small floes. The floe perimeter per unit ice area is a more suitable and important metric for the evaluation of FSD models as the development of the FSD models are particularly important in the marginal ice zone, where small-floe-related processes are active, such as the thermodynamic freezing and melting of sea ice floes. Besides, Denton and Timmermans (2022) also suggested the SIC should be considered in studying the temporal evolution of FSD, e.g., for use in sea-ice models. The use of perimeter density per unit sea ice area for FSD models evaluation corresponds to this result.

6) To avoid the confusion caused by the definition, we revised the title to "Summer sea ice floe perimeter density in the Arctic: High-resolution optical satellite imagery and model evaluation".

Comparison of Models and Observations

========================================

The lack of agreement between the models and the observations, and between the models themselves, is truly remarkable. The histograms of PD are all completely different (Figure 2, panels (a) through (d)). The plots of PD vs. sea-ice concentration (SIC) (Figure 3) show that FSDv2-WAVE has values of PD that are an order of magnitude larger than the observations, with vastly greater variability, and a slope (vs. SIC) with the wrong sign. CPOM-FSD is hardly any better. WIPoFSD has a slope similar to the observations but with PD values five times larger. The model/obs differences are noted by the authors at lines 196-206 and 219-231.

In Section 5, the causes of the differences between observations and models are attributed to three factors (lines 322-324). The first factor, image resolution, cannot explain such large differences (line 327). The second factor, underestimation of SIC in the models, "can partially explain" (line 338) such large differences. The third factor, overactive wave fragmentation in the models, was investigated by dividing each region (CS and FS) into north and south portions, and comparing observations vs. models in these sub-regions. In the CS region, all the observations are in the south sub-region. In the FS region, all the observations are in the north sub-region.

In Figure 7(a) for the CS region, the agreement between observations and FSDv2-WAVE(south) is still terrible, and the agreement between observations and CPOM(south) doesn't appear to be any better than in Figure 3(a). In Figure 7(b) for the FS region, the agreement between observations and FSDv2-WAVE(north) is still terrible, and the agreement between observations and CPOM(north) is not particularly good. In my opinion, the analysis by sub-region has not resolved or shed light on the large differences between observations and models.

Lines 352-353. To investigate "unrealistically high perimeter densities in our study regions" the authors "examined the P_i in the northern regions where wave-induced breakup is negligible. In these regions, most modelled P_i match our observations better."

This seems to be saying that the authors have compared model results from the northern sub-regions with the observations. But for the CS region, all the observations are in the south sub-region, so it makes no sense to compare models in the north with observations in the south. For the FS region, Figure 7(b) does not show that "most modelled P_i match our observations better." I don't see any kind of match between models and observations, nor much improvement over Figure 3(b).

Lines 359-360. "positive biases of P_i are closely linked to overactive wave fracture in the models."

I don't believe that the authors have demonstrated this.

Looking at the big picture, I can only think of two possible explanations for the enormous differences between the models and the observations, and between the models themselves: either the models are complete junk, or the PD is meaningless as a diagnostic of model performance. Do the authors have any thoughts on this?

Thanks for your comments.

1) In response to the following comments, *"the analysis by sub-region has not resolved or shed light on the large differences between observations and models"*,

   In the northern Chukchi Sea and the Fram Strait, the change in Pi arising from all FSD evolution processes is almost zero in the two prognostic models during our research period. For checking the Pi in these regions, we recognize the Pi value in the northern region is the value at the end of early spring or the beginning of our research period. This comparison could help us determine whether the differences between observations and models arising from FSD evolution processes in summer (lateral melt and wave fracture) or before that.

   Thanks for pointing out it. We will give more explanation about the reason for comparing Pi in northern and southern regions in the Chukchi Sea and the Fram Strait.

2) In response to the following comments, *"Looking at the big picture, I can only think of two possible explanations for the enormous differences between the models and the observations, and between the models themselves: either the models are complete junk"*,

   The development of the FSD models is essential to improve confidence in sea ice models. Both prognostic FSD models and the power-law FSD model are in the development phase, although models show differences relative to the observations in small floes, it could partially give the ideas of the FSD evolution at this stage. For example, in Fig. 2h, CPOM-FSD (yellow lines) shows good performance in simulating Pi for floes with radius at hundreds of meter scales. As discussed by Bateson et al. (2022), the inclusion of the novel brittle fracture scheme does improve the model performance in hundreds of meter scales compared with observations. There are still many unknowns of FSD evolution for both observations and models. The evaluation of recent FSD models can provide useful insights for model developers, while until now there has been no detailed and comprehensive study comparing modelling FSD with high-resolution observation at metres to thousands of metres scales on a seasonal scale in a long-term time series. This is the first time evaluating the FSD model by using high-resolution satellite imagery resolving small floes in different regions and long-term periods covering more than 10 years. FSD models are still in development and a work in progress. This study is very valuable in contributing to the future development of FSD models by pointing out the differences between models and observations and the possible reason causing these differences, which could help modelers find a way to improve the FSD simulations in the sea ice models and thus the confidence of climate projection in the future.

3) In response to the following comments, *"the PD is meaningless as a diagnostic of model performance... "*,

   At the beginning of this research, we tried to use areal FSD as a comparison metric, showing similar differences between models and observations (models show overestimation for small floes and underestimation for larger floes in general). This difference is not caused by the metric used. For the reasons for the

use of perimeter per unit sea ice area, please see our response as mentioned previously.

Other Comments

==============

About the MEDEA imagery used in this study (lines 59 and 88-90), please see Denton and Timmermans (2022), and say briefly how their data set and analysis relates to the present work.

It is an interesting study. The SIC-FSD relationship could give evidence about the processes governing FSD evolution and give us more confidence in the use of perimeter density per unit sea ice area rather than per unit ocean area. Their test of FSD sensitivity to this choice reveals that the FSD can appear divided into two power-law regimes if this choice does not adequately identify small floes. This result corresponds to the difference between Pi derived from 1-m and 0.5-m resolution images (Fig. 4 in our manuscript). Better identifying small floes could expand the power-law range. As in the comments from Referee #2 Fabien Montiel and my response to him, there are also two possibilities leading to these two regimes. One is the limitation of the power-law hypothesis (see details in Montiel and Mokus, 2022). The other one is that small floes are more vulnerable to lateral melt relative to large floes, which causes the deviation of the small-floe distribution prior to the distribution tail (see details in section 4(d) in Hwang and Wang, 2022).

Lines 79-81. "the observations from the Chukchi Sea region captures a more dynamic and fragmented ice condition (e.g., Fig. 1b), while the observations from the Fram Strait capture a less dynamic environment (e.g., Fig. 1c)."

Looking at Figure 1, I don't see that the Chukchi Sea image indicates more dynamic and fragmented ice than the Fram Strait image. The two images look similar to me. How does a single image convey dynamics?

Fig. 1b shows more fragmented sea ice condition and includes more small floes between mid-size floes relative to Fig. 1c.

Thanks for your comments, we have replaced this figure with their partially enlarged image cuts shown below, which can show more details of the difference between them.

[Figure]

Section 3.1.2. What measure of floe size is used? All I can gather is that floe size is characterized by a radius. Is it half the mean caliper diameter? Is it the radius that a circular floe of the same area would have?

In this study, we define floe size as floe effective radius, the radius of a circle that has the same area as a floe. We have added this information about the measure of floe size to the revised manuscript.

Also, at lines 103-105, "we first applied combined filters: median, bilateral and Gaussian filter" and "The smoothing term, KGC algorithm parameter, was set as 0.0001" -- either provide more detail so that the reader can understand what this means, or leave it out and just refer to Hwang et al (2017).

Thanks, we will just cite Hwang et al (2017) in the revised manuscript.

Section 3.1.3. What is the spatial resolution of the sea-ice data? Also, the analysis period is 2000-2014 but AMSR-E is only available 2002-2011 and AMSR-2 is only available since 2012. What sea-ice products (with what resolution) were used during what time periods?

Thanks for your comments. The resolution for AMSR-E and AMSR2 applied in this study is 6.25 km. The resolution for NSIDC sea ice concentration is 25 km.

Yes, in our study period, there isn't SIC derived from AMSR-E in April- August 2000-2001, April-May in 2002 and April-June in 2012. That is one of the reasons that we also applied the SIC from NSIDC, covering April to August during 2000-2014.

Thanks for pointing out this, we have added the explanation of the resolution of the SIC products and the period for each product that we used for analysis.

Lines 121-123. Does 1-degree grid mean 1-degree in latitude and 1-degree in longitude? What does "gx1v6" mean (line 122)? Also, the models are run "for 37 years from 1 January 1980, followed by a 10-year period spin-up" so the spin-up period is 2017-2026 i.e. partially in the future. Is that correct?

Yes, 1-degree grid means 1 degree in latitude and 1 degree in longitude, which is around 30-63 km per gird. gx1v6 is one of the displaced pole grids of the global model, which is approximately 1-degree resolution horizontally (320x384 for horizontal grid with the horizontal resolution of nominal 1 degree). Thanks for pointing out this inaccurate description. The spin-up period is the first 10 years, which is 1980-1989.

We have added a more detailed explanation of the grid used in models and spin-up period as follows.

FSDv2-WAVE model has the displaced 1° gx1v6 grid (320x384 for horizontal grid) over a global domain, which is approximately 1-degree resolution horizontally. The other two models are initiated with the ice-free Arctic and run with the tripolar 1° ($129 \times 104$) grids for 37 years from 1 January 1980, including a 10-year period spin-up during 1980–1989 in a pan-Arctic domain excluding Hudson Bay and the Canadian Arctic Archipelago.

Lines 155-157. "the model also simulates FSD evolution through the floe size parameter r_var ..."

Please define r_var. I see that it varies between r_min and r_max, and I see that it evolves according to four FSD processes, but no definition of r_var is given. What is it?

The $r_{var}$ is defined in the equations 17-19 in Bateson et al. (2020). We will quote that the definition of r_var is given in Bateson et al. (2020) in the revised manuscript.

In addition to the exponent, the model also simulates FSD evolution through the floe size parameter $r_{var}$, varying between minimum floe radius $r_{min}$ and maximum floe radius $r_{max}$. $r_{var}$ evolves according to four FSD processes: lateral melt, wave-induced fracture, floe growth in winter and ice advection (Bateson et al., 2020, 2022). The detailed definition of $r_{var}$ is given by Bateson et al. (2020), i.e., a variable FSD tracer reflecting the evolution of FSD in response to the physical processes.

Section 3.3. This section (FSD definition, lines 158-171) is confusing and unnecessarily complicated.

-- "The FSD is usually defined as the floe areal FSD..."

It's confusing to use FSD in the definition of FSD!

-- "By integrating f(r) over floe radius between r and r+dr, f(r)dr (dimensionless) is obtained"

This makes no sense.

Thanks. We revised the definition of FSD as follows.

The FSD is usually defined as the fractional-area distribution f(r)dr and number-density distribution n(r)dr (Rothrock and Thorndike, 1984; Toyota et al., 2006; Perovich and Jones, 2014; Horvat and Tziperman, 2015; Zhang et al., 2015; Hwang et al., 2017b; Bateson et al., 2020). Areal FSD f(r)dr corresponds to the area of floes per unit ocean surface area with radii between r and r+dr.

-- It's really not necessary to introduce the Heaviside function and equation (1) in order to define the FSD, especially since they're not used in the rest of the paper.

-- The cumulative floe number distribution, defined at lines 169-171, is also not used in the rest of the paper.

Thanks for your comment. We will delete the definition of F(r) in equation (1) and only keep the definition of n(r) and f(r).

Line 207. "normalized" perimeter density is not defined in the text. The caption for Figure 2 says "The normalized perimeter density distributions were obtained by dividing the width of every floe size category into P_i at each region." The original P_i has units of 1/km and the "normalized P_i" has units of 1/km^2. In what way is this a normalized quantity? Usually I think of normalization as producing a dimensionless result such as a percentage. What is the point of "normalizing" by dividing by the width of the floe size category to produce another dimensional quantity?

Thanks. Yes, the "normalized" perimeter density distribution is not normalized to 1. $P_i$ is defined as the perimeter of floes per unit sea ice area (units: km km$^{-2}$). We will define the "normalized" perimeter density $p_{ice}$ (units: km$^{-2}$) as the perimeter of floes in floe size category i per unit bin width per unit area in the revised manuscript. So, the integration of p(r) will be perimeter density $P_{ice}$, $P_{ice} = \int_{r_i}^{r_{i+1}} p_{ice}(r)dr$. In Figure 2, what we want to show is the perimeter density at the midpoint of each bin rather than Pi, the integral of the perimeter density over each category. That's why we produce $p_{ice}$.

Lines 270-271. "we constructed two data sets: monthly changes of P_i arising from lateral melt and FSD changes arising from wave breakup." How were these two data sets created?

These two variables are from model outputs. For the FSDv2-WAVE model, please see the first two terms on the right-hand side in equation (14) given by Roach et al. (2018), representing the growth and melt of existing floes in thickness and lateral size. For CPOM-FSD, the fraction of sea ice area lost due to lateral melting is shown in equation (8) in Bateson et al. (2022). The change in the FSTD f(r, h), per unit time due to fracture by ocean surface waves in both FSDv2-WAVE and CPOM-FSD is given in equation (20) by Roach et al (2018).

Thanks for pointing out this, we will add more information about these two variables in the revised manuscript.

Line 274. "CPOM-FSD produces negative changes in P_i from wave fracture (Figs. 6f and 6h)." But Fig. 6h shows changes arising from lateral melt, not wave fracture. Furthermore, Figs. 6e and 6f show that the change is positive, not negative. Finally, note that the panels in the bottom row of Fig. 6 are labelled (g) (e) (h) (f) from left to right. This might be the source of some confusion.

Line 280. "CPOM-FSD shows a stronger reduction in P_i arising from lateral melt (Figs. 6e and 6g)." But Fig. 6e shows changes arising from wave fracture, not lateral melt. See also the previous comment.

Thanks for pointing out this. Have revised Line 274 as below and the label of the bottom panels in Fig. 6.

The results show that FSDv2-WAVE produces larger positive changes in Pi from wave fracture (Figs. 6b and 6d) in summer relative to CPOM-FSD (Figs. 6f and 6h).

[Figure]

Figure 6: Monthly changes of $P_i$ simulated by the two prognostic models over the period May to July during 2000–2014. (a) Change of FSD arising from lateral melt for FSDv2-WAVE in the Chukchi Sea. (b) is same as (a) but for wave induced FSD change. (c) and (d) are same as (a)and (b) but in the Fram Strait. (e)–(h) is same as (a)–(d) but for CPOM-FSD. The blue and red box in (a) and (c) show the northern and southern region of the two study regions. Black dots indicate the location of observations in the study regions.

Lines 302-303. "The close match between CPOM-FSD and the observations for the northern Fram Strait region..."

I'm looking at Figure 7(b) for the Fram Strait region. The observations (black circles) were acquired in the northern part of the region. The CPOM(north) results are indicated by open yellow circles. I don't see a close match between the black circles and the open yellow circles.

Thanks. The Pi in the northern Fram Strait simulated by CPOM-FSD is of the same order of magnitude of observations. Will use a more rigorous description: "The much closer

match between CPOM-FSD and the observations for the northern Fram Strait region suggest no significant wave fracture and lateral melt has occurred in the observation site".

Lines 314-316. "The observation results show clear regional differences between the two study regions, i.e., much larger perimeter density P_i (smaller floes) in the Chukchi Sea region than in the Fram Strait region."

I'm looking at Figs. 3(a) (Chukchi Sea) and 3(b) (Fram Strait). The observations are indicated by black circles. When I look back and forth at the black circles in (a) and (b), I just don't see the "clear" regional differences.

In lines 196-197, observations show a substantial difference in P_i between the two regions (t-test, t (47) = 6.43, p < 0.001) (Fig. 2a). It shows a significantly higher P_i of $20.77 \pm 6.54$ km km$^{-2}$ in the Chukchi Sea site than the P_i of $12.16 \pm 3.79$ km km$^{-2}$ in the Fram Strait site (Fig. 2a). Regional differences shown in observations is at tens km$^{-1}$ scale. For the black circles in 7(a) and 7(b), due to the overestimation of Pi in models, the y-axis limits for Fig. 7 range from 0 to 350 km km$^{-2}$. At this range, the regional difference in observations is not clearly shown in this plot. But the result from the t-test could statistically support this view.

Lines 320-322. "The observations and WIPoFSD model both show a positive correlation between SIC and P_i ... while the two prognostic models show the opposite (negative) correction."

(Note that the word "correction" should be "correlation").

Thanks. We revised this.

In Figure 3, I see the opposite of what's stated here: observations and WIPoFSD both show NEGATIVE correlations between SIC and P_i; FSDv2-WAVE and CPOM-FSD both show POSITIVE correlations between SIC and P_i.

Thanks. We have corrected.

Supplementary Materials

========================

Equation (1) and following.

-- It's bad notation to use "i" as a subscript on the left-hand side and "i" as an index of summation on the right-hand side. See also my comments below about equation (3) of the main text.

Thanks. We changed $P_i$ to $P_{ice}$ now.

-- The parameter GAMMA is not defined.  "Here GAMMA is a floe shape parameter, for example..." (line 22) -- this is not a definition.  I gather from equation (1) that the floe perimeter is 2*GAMMA*r, which perhaps defines GAMMA (if so, that should be stated explicitly).  In that case, for circular floes, GAMMA = pi, as noted on line 22, but for square floes, GAMMA is not equal to 1, as erroneously stated on line 22.  Consider a square floe of side s, perimeter 4*s, area s^2.  If r is the radius of a circular floe of the same area, then s^2 = pi*r^2.  The perimeter of the square floe is 4*s = 4*sqrt(pi)*r.  If this is equal to 2*GAMMA*r then GAMMA = 2*sqrt(pi) = 3.54.

-- Rothrock and Thorndike (1984, hereafter RT84) calculated a "shape parameter" similar to GAMMA, finding that AREA / MCD^2 = 0.66 +/- 0.05, where AREA is the area of a floe and MCD is its mean caliper diameter.  In the present context, if AREA = GAMMA*(r^2) and MCD = 2*r then the RT84 shape parameter is GAMMA/4 which implies GAMMA = 2.64 +/- 0.20, which is not too different from the values given on lines 23 and 24.

Thanks for your helpful comments. We will give a clear definition of gamma and have revised all equations in both the supplement and main text.

As r applied here is effective floe radius $r_{eff} = \sqrt{a/\pi}$, the radius of a circle that has the same area a as a floe. We assume $p = 2\gamma r_{eff}$. Here the gamma should be defined as $\gamma = \frac{p}{2r_{eff}} = \frac{p}{2\sqrt{a/\pi}} = \frac{p\sqrt{\pi}}{2\sqrt{a}}$, where p and a are the perimeter and area of a floe respectively. The gamma value we give in L23-L24 (the mean floe shape parameter $\gamma$ is 2.27 in the Chukchi Sea region and 2.23 in the Fram Strait region) is $\frac{4a}{d^2}$, which corresponda to 4*0.66=2.64 in Rothrock and Thorndike (1984). Thanks for pointing out this, which make us realize that there was a mistake in the value of gamma for the calculation of Pi. Now we have corrected both the equations and the Pi value shown in the results sections.

For $\gamma = \frac{p}{2r_{eff}} = \frac{p}{2\sqrt{a/\pi}} = \frac{p\sqrt{\pi}}{2\sqrt{a}}$, the related value given by Rothrock and Thorndike is $\frac{a}{p^2} = $ 0.06, which means $\gamma = \frac{p\sqrt{\pi}}{2\sqrt{a}} \approx 3.62$ in Rothrock and Thorndike (1984). This ratio is for floes with diameters over about 1 km. For our data, if we ignore the floes with caliper diameters smaller than 1 km, $\gamma = 4.57$ in the Chukchi Sea and $\gamma = 4.55$. For $r_{eff} > $ 0.56 m (the effective radius of a 1-pixel floe) and $r_{eff} > 945.81$m (the upper limit of floe size simulated in models), $\gamma = 2.17$ in the Chukchi Sea and $\gamma = 1.96$. This ratio between floe perimeter and floe effective radius is closely related to the lower limit that we choose. As in this study, the comparison between models and observations is in the range of 0.07<r<945.81m, we prefer to use the shape parameter calculated within this range. The corrected gamma value is given in the revised manuscript as below.

Here $\gamma$ is a floe shape parameter, $\gamma = \frac{p}{2r_{eff}}$. From the analysis of MEDEA-derived FSD results, the mean floe shape parameter $\gamma$ is 2.17 in the Chukchi Sea region and 1.96 in the Fram Strait region, respectively.

The relationship between the areal FSD $f(r)$ and CFND $N(r)$ is revised to $\int_{r_0}^{\infty} f(r)dr = \int_{r_0}^{\infty} \pi r^2 dN(r)$.

$P_{ice}$ for prognostic model $P_{prog}$ is revise to $P_{prog} = 2\sum_{i=1}^{12} \frac{\gamma f_i(r_{i_{max}}-r_{i_{min}})}{\pi r_i c_{ice}}$.

$P_{ice}$ for WIPoFSD is revised to $P_{wipofsd} = \frac{\int_{r_{min}}^{r_{var}} 2\gamma r n(r)dr}{c_{ice}} = \frac{2\gamma(3-\alpha)(r_{var}^{2-\alpha}-r_{min}^{2-\alpha})}{\pi(2-\alpha)(r_{var}^{3-\alpha}-r_{min}^{3-\alpha})}$.

$P_{ice}$ for observations is revised to $P_{obs} = \sum_{i=1}^{12} \frac{2\gamma A_{floe_i}}{\pi r_i A_{ice}}$.

-- The meaning of the terms in equation (1) should be explained more clearly. For example: (r_i_max - r_i_min) is the bin width of the i-th bin; n_i is the number of floes in bin i per unit bin width per unit area; therefore their product is the number of floes in bin i per unit area. Therefore the quantity inside the summation is the perimeter of the floes in bin i, per unit area, and the numerator is the total floe perimeter per unit area. After dividing by the sea-ice concentration c_ice, one obtains the total floe perimeter per unit area of sea-ice -- the floe PD.

Thanks for your suggestions. We revised the explanations as follows.

In FSD models, $P_{ice}$ is calculated from number FSD $n_i$ distributed into floe size categories i as follows:

$$P_{ice} = \frac{\sum_{i=1}^{12}(2\gamma r_i n_i(r_{i_{max}}-r_{i_{min}}))}{c_{ice}}, \tag{1}$$

where $r_i$, $r_{i_{max}}$ and $r_{i_{min}}$ are the midpoint, upper and lower limit for each floe size category i. $r_{i_{max}} - r_{i_{min}}$ is the bin width of floe size category i. $n_i$ (m$^{-3}$) is the number of floes in floe size category i per unit bin width per unit area. So $n_i(r_{i_{max}} - r_{i_{min}})$ is the number of floes per unit area for each floe size category i. The upper term in the right hand of Eqs. (1) is the perimeter of floes per unit area. By dividing total floe perimeter per unit ocean area by the sea-ice concentration, $c_{ice}$, we obtain the total floe perimeter per unit sea ice area.

-- Lines 23-24, "From the analysis of MEDEA-derived FSD results..." This sentence belongs in the section on "Calculation of P_i from the observations" at line 47, not in the section on the calculation of P_i from models.

Prognostic models didn't output perimeter density, we also need a gamma for calculating perimeter density in equation (4), which is revised as $P_{prog} = 2\sum_{i=1}^{12} \frac{\gamma f_i(r_{i_{max}}-r_{i_{min}})}{\pi r_i c_{ice}}$. Here we use the same value as in observations for the same floe size range.

Equation (4) (line 33) is the same as equation (3) of the main text (line 181). Also, lines 42-46, including equation (8), are an exact repeat of lines 184-188 of the main text, including equation (4) there. Why repeat the same material in the Supplement?

Thanks for your comments. We deleted the same equation as in the main text.

Minor Comments

==============

Line 173. The units of perimeter density are given as 1/meter but in much of the rest of the paper the units are 1/kilometer. Figures 3 and 7 use 1/km but Figure S1 uses 1/m.

Thanks. We corrected it.

Equation (3) -- notation.

-- On the right-hand side, the index "i" is used in the summation, and on the left-hand side, the index "i" is used as a subscript on P. This is bad notation.

-- Same comment for equation (5) and most of the equations in the Supplementary Materials.

-- The bad notation is easily fixed by dropping the subscript "i" on the perimeter density -- just use "P" (line 173 and following). There's no reason for a subscript.

Thanks. We revised them to $P_{prog}$, $P_{wipofsd}$, $P_{obs}$.

-- Also, doubly-subscripted variables r_i_max and r_i_min are confusing and unnecessary. The quantity (r_i_max - r_i_min) is just the bin width of the i-th bin or category. It could be denoted w_i or something else with a single subscript i.

Thanks. We replaced the bin width $r_{i_{max}} - r_{i_{min}}$ as $w_i$.

Equation (4). GAMMA is not defined. Also, I believe ALPHA = 2.56 in this case, which is perhaps worth repeating here.

Thanks. We defined gamma and repeat the value of alpha in the main text.

Figure 1 caption. The date for panel (b) should probably be 12 June, not 6 June.

Thanks. Revised.

Figure 2. In panels (e) through (k), what is the meaning of "Effective" floe radius?

Thanks. We will define effective floe radius in section 3.3 as follows.

In this study, we define the floe size as effective radius $r = \sqrt{a/\pi}$, the radius of a circle that has the same area, a, as a floe.

Table 2. The "a" and "b" superscripts are missing from the table.

Thanks. Corrected.

Table 3. The caption says "FSDv2-WAVE and WIPoFSD" but the table itself lists FSDv2-WAVE and CPOM-FSD.

Thanks. Corrected.

**Citation in the response to Referee #1**:

Arntsen, A. E., Song, A. J., Perovich, D. K., and Richter-Menge, J. A.: Observations of the summer breakup of an Arctic sea ice cover, Geophys. Res. Lett., 42, 8057–8063, https://doi.org/10.1002/2015GL065224, 2015.

Bateson, A. W., Feltham, D. L., Schröder, D., Hosekova, L., Ridley, J. K., and Aksenov, Y.: Impact of sea ice floe size distribution on seasonal fragmentation and melt of Arctic sea ice, Cryosph., 14, 403–428, https://doi.org/10.5194/tc-14-403-2020, 2020.

Bateson, A. W., Feltham, D. L., Schröder, D., Wang, Y., Hwang, B., Ridley, J. K., and Aksenov, Y.: Sea ice floe size: its impact on pan-Arctic and local ice mass and required model complexity, 16, 2565–2593, https://doi.org/10.5194/tc-16-2565-2022, 2022.

Perovich, D. K.: Aerial observations of the evolution of ice surface conditions during summer, J. Geophys. Res., 107, 8048, https://doi.org/10.1029/2000JC000449, 2002.

Perovich, D. K. and Jones, K. F.: The seasonal evolution of sea ice floe size distribution, J. Geophys. Res. Ocean., 119, 8767–8777, https://doi.org/10.1002/2014JC010136, 2014.

Roach, L. A., Horvat, C., Dean, S. M., and Bitz, C. M.: An Emergent Sea Ice Floe Size Distribution in a Global Coupled Ocean-Sea Ice Model, J. Geophys. Res. Ocean., 123, 4322–4337, https://doi.org/10.1029/2017JC013692, 2018.

Roach, L. A., Bitz, C. M., Horvat, C., and Dean, S. M.: Advances in Modeling Interactions Between Sea Ice and Ocean Surface Waves, J. Adv. Model. Earth Syst., 11, 4167–4181, https://doi.org/10.1029/2019MS001836, 2019.

---

## Author Comment (AC2)

Dear Dr Fabien Montiel,

We would like to thank you for providing valuable comments and suggestions. Please find our responses to your comments in blue text and revised manuscript in red text below.

Best regards,

Yanan Wang and co-authors

**Response to Referee #2**

First, I would like to apologise to the authors and editor for the delay in submitting my review. The manuscript attempts to compare floe perimeter density data obtained via satellite imagery at 2 locations in the Arctic Ocean against the predicted density of 3 floe size distribution (FSD) models, which all include a parametrisation of wave-induced fracture. The goal is to evaluate the performance of the models. Results show that the discrepancy is quite significant in a number of ways. In particular, the models generally predict much larger perimeter densities than the observations, meaning an overestimation of small floes. The authors then attempt to explain the discrepancy by discussing potential issues with specific parameterised components of the models considered.

The manuscript presents original work, is overall well written and the topic is highly relevant to improving the modelling capabilities of FSD resolving sea ice models. That said I have a number of important issues with the way the study has been designed, which may explain the poor agreement between models and data. These are detailed below as well as other comments. I therefore recommend the manuscript undergoes major revisions before it can be further considered for publication.

**Main comments**

1. My main concern relates to the way the study regions for both models and observations were selected in relation to one another. If I understand correctly, the satellite images were chosen at 2 specific locations, one in the Chukchi Sea and one in the Fram Strait. In contrast, the regions selected for analysing model outputs are much larger. They do include the specific observational locations but extend over much wider regions, selected to include the ice edge. My issue is that the variability in FSD in these regions is likely to be much larger than that at the locations of the data. If the data is collected far from the ice edge (which could have been estimated), floe sizes are likely much larger than closer to ice edge. Therefore, I am not sure the comparison between model outputs and data is fair with respect to the models. In fact, when the comparison is refined to a subdivision of the initial study regions, the agreement is improved. I am not an expert in analysing satellite imagery, so my following suggestion could be naïve and/or uninformed, but this is what I would have done: (i) select all the imagery available in the model study regions, not just those at the specific locations selected, OR, if this is too much data to analyse, (ii) take a random sample of images spanning the study regions. Either

way, the variability in the data would be a lot more representative of that predicted by the models. I am not suggesting that the authors redo the entire analysis for this paper, but I feel like this limitations needs to be given a lot more emphasis in the manuscript. At the very least, bring this up in the Discussion section, but what would be even better is to add a sub-section looking at the comparison models/data when the model outputs are only selected in a smaller region around the data locations, even more localised than the subdivision shown in figure 6.

Thanks for your comments and suggestions. For models, we have tried to select a small region (5×5 grid cells) for comparison. However, for the comparison between observations and several models that have different atmospheric forcing, there are very different ice conditions within this small region, e.g., one model shows MIZ but the other two show interior ice pack. This situation will cause the unfair in the model-observation comparison. By choosing large regions including the ice edge, we can make sure the perimeter density simulated by all models indicate the average perimeter density in the region far from the ice edge and close to the ice region.

For observations, all MEDEA images used in this study in the Chukchi Sea and the Fram Strait are in the same fixed location, 70°N, 170°W for the Chukchi Sea and 84.9°N, 0.1°E for the Fram Strait. However, although the images applied in this study are in the fixed site, different imagery data can also indicate the perimeter density at different distances to the ice edge. For example, the sea ice concentration for 7 out of 13 cases in June in the Chukchi Sea is below 80%, indicating MIZ in the images. And the left 6 June cases indicate interior ice pack. Hence, when we compare the monthly mean (as the solid lines in Figure 2e-2k) rather than the case-by-case comparison, it is the comparison of the mean state of perimeter density both near the ice edge and far from the ice edge for both models and observations.

Thanks for pointing out this limitation. There are some extra MEDEA images for sea ice buoys available within the study regions, which is not in the fixed locations of "fiducial sites". However, for this study, we believe the variation of observations is enough for the model-observation comparison study. Thanks for your helpful suggestions. In future study, these images could be used to expand the spatial variability in the observations. We will add more explanations about these limitations in the Discussion and Conclusion section.

2. The title of the manuscript is misleading and should be changed, as the authors do not analyse the FSD but the floe perimeter density, which is different. The abstract also needs some work as it starts with statements on the FSD and then switches to perimeter density without making a link between them. I am not saying the perimeter density is a bad metric, but it is not the one advertised! It took some time to fully appreciate the meaning of Pi (again not an expert!). It would have helped me to show the Pi results with units km per km^2. I would have liked the relationship between these 2 quantities discussed in more details in relation to the results. For instance, do we expect FSDs to have similar qualitative and quantitative properties as the perimeter density distributions shown in figs 2e-k?

Thanks for your comments. In previous observational studies (Perovich, 2002; Perovich and Jones, 2014 and Arntsen et al., 2015), perimeter per unit area has been applied as a proxy for floe size distribution. As discussed by Perovich (2002), the use of the perimeter of the sea ice floe reduces the impacts of partially captured floes at the edge of the image for FSD retrieval. Besides, perimeter density per unit ice area is widely used as a metric to show the evolution of the FSD, e.g., Roach et al. (2019) and Bateson et al. (2022). As explained by Roach et al. (2019), "$P_{ice}$ is weighted more heavily by smaller sizes, so Pi is more relevant for thermodynamic melting and freezing of floes", which is important in evaluating the FSD models. For more details on the discussion of the definition of floe number density N(r), floe area fraction F(r) and perimeter density P, please see our response to referee #1.

Thanks for your suggestions. We will revise the title to " Summer sea ice floe perimeter density in the Arctic: High-resolution optical satellite imagery and model evaluation" and include the relationship between FSD and perimeter density in the abstract. We also changed the units of Pi to km km$^{-2}$ to make it easier to understand the meaning of Pi.

**Other comments**

3. L28-30: This statement is confusing, especially for the uninformed reader. Quantifying the MIZ is still an active an area of research. Mentioning the 2 definitions (i.e. wave-based and SIC-based) in this way makes it seem that they are equivalent, but that's not true (see Brouwer et al paper). I suggest you pick one definition or expand the discussion.

Thanks for your suggestions. We expanded the discussion as below.

This results in the changing marginal ice zone (MIZ), defined as the ice-covered region affected by waves and swell by the World Meteorological Organization (WMO, 2014). However, due to the difficulty in detecting wave-ice interaction data from satellite imagery (Horvat et al., 2020), another optional definition of MIZ widely applied in previous studies is a sea ice-covered area with sea ice concentration (SIC) of 15%–80% (e.g., Strong and Rigor, 2013; Aksenov et al., 2017; Rolph et al., 2020; Bateson et al., 2020; Horvat, 2021).

4. L43: "viscous dissipation" is not the process governing attenuation, it refers effective/homogenised dissipative rheology of continuum viscous layer models often used to approximate attenuation caused by a non-homogeneous ice cover. In any cases, "dissipative processes" would be a better choice of wording here.

Thanks for your suggestions. We changed the wording in the revised manuscript.

The FSD affects the ocean surface waves propagation and attenuation through the ice, i.e., floes smaller than a characteristic wavelength of swells attenuate wave energy through dissipative processes

5. L53-55: The limitations of the power-law should be discussed. See Montiel & Mokus (2022, Philosophical Transactions A) for an overview.

Thanks for your suggestions. We added more discussion about the limitation of the power-law hypothesis.

Some other modelling studies assumed a particular shape of FSD (e.g., Bennetts et al., 2017; Rynders, 2017; Bateson et al., 2020). Previous observational studies reported a power-law behaviour existing in the tail of FSD, i.e., a straight line in logarithmic axes, leading to the parametrisations of fixing the FSD as following a truncated power law (Burroughs and Tebbens, 2001). However, the floe size power-law hypothesis has been contested by recent observations (Steer et al., 2008; Herman, 2010; Herman et al., 2021), which suggest different shapes or functions for describing FSDs. Results from laboratory experiments (Herman et al., 2017; Passerotti, 2022) and models (Herman, 2017; Montiel and Squire, 2017; Mokus and Montiel, 2021; Montiel and Mokus, 2021) also express the concerns that a power-law FSD is not appropriate for wave-induced sea ice breakup.

6. L77: I'm not sure I follow the argument that a low-resolution model output justifies the choice of a large model study area.

Thanks for your comments. For the WIPoFSD model and CPOM-FSD model, the grid cell area ranges from about 2468 km2 to 3480 km2 in the study region. For FSDv2-WAVE, the grid cell area is between 541 km2 and 3802 km2 in the study region. Within every grid cell, the atmosphere forcing, wave forcing, etc. are homogeneous. The accuracy of FSDs in every grid cell relies heavily on the accuracy of external forcing. Applying a large region for models could better reveal the inhomogeneous effects of external forcing on the evolution of FSD and decrease the effects of external forcing bias in a few grid cells on FSDs.

We will add more details of this argument to the revised manuscript.

7. L90: What's the area of the uncropped images? 250 km^2? It would be interesting to know.

The size of the raw MEDEA image is approximately more than 15 by 15 km, i.e., covering an area larger than 225 $km^2$ (Kwok, 2014). For the World-View image applied in this study, the size of raw images is more than 200 $km^2$, but we only purchase part of it (~40 $km^2$). The image size information will be added in the revised supplement.

8. L91-93: A reference for the WV images and more details about the sensor should be given.

Thanks. A reference for WV images will be added to the revised manuscript. A more information is added as shown below.

We also collected one WorldView-1 (WV1) and four WorldView-2 (WV2) images at the Chukchi Sea and Fram Strait sites (Fig. 1), which are collected in a 50 cm panchromatic band (WV1) at 0.52 m spatial resolution and 50 cm panchromatic band and 2 m 4-band Multispectral Bundle (WV2) at 0.47 m and 0.58 m spatial resolution. The size of the WorldView (WV) images is ~40 km$^2$.

9. L98-102: how is "floe size" defined? There are multiple definitions out there.

   Thanks for pointing out this. In this study, we define floe size as floe effective radius, the radius of a circle that has the same area as a floe. We have added this information to the revised manuscript.

10. Section 3.2 is confusing. The intro paragraph discusses differences between the different models, before describing the models themselves in subsequent sub sub-sections. It would make more sense to describe the models first and then discuss their differences/similarities.

    Thanks for your comments. The introduction paragraph gives a broad description and review of the recent FSD models including but not limited to models evaluated in this study. Section 3.2 introduces three specific models within them. In our opinion, it may be better to keep them, i.e., describing the differences between recent FSD models first and then introducing the details of the three specific models.

11. Section 3.2.1: More details about the WW3 configuration are needed. Is it a global or regional run? What ice attenuation parametrisation did you choose? Also, what do you mean "attenuation in the open ocean"?

    Thanks for your comments. It is a global run. As in Roach et al. (2019), attenuation is the sum of two parts. "The first is an empirical fit to a multiple-floe scattering theory by Meylan et al. (2021) that accounts for floe size, thickness, and sea ice concentration. This theory finds that the scattering resembles the linear Boltzmann equation derived by Masson and Leblond (1989) with an error correction that accounts for multiple scattering. Attenuation from this theory becomes negligible when wavelengths are much greater than floe sizes; therefore, Roach et al. (2019) include an additional term based on observations to attenuate waves like the inverse of the wave period squared".

    More details will be given as a brief introduction to WW3 in the revised manuscript.

12. L149: "process" is the wrong word. Maybe "theory" or "model"?

    Thanks. The wording is changed as follows.

    The WIPoFSD model implements the wave-in-ice model (WIM), originally based on the ice-wave interaction theory described by Williams et al. (2013a, 2013b)

13. L153-154: I feel more details are needed about how the fixed power was determined. Was it the same data as those used later for comparisons with model outputs. Were all the floe size data from all the images collated into a single dataset and then a power law fit was performed or were power law fit done for each image and then averaged? What was the range of floe sizes considered for the fit(s)? Also is 3 significant figures appropriate? What's the uncertainty on alpha? Was a goodness-of-fit test evaluation conducted?

Thanks for your comments. It is not completely the same dataset. The exponent of -2.56 was determined for the dataset considered in the Bateson et al. (2022) study rather than the more complete dataset considered in our study. To describe the differences between the dataset used by Bateson et al. (2022) and our study, hereafter, we refer to the observations used for calculating the power-law fit used in WIPoFSD (Bateson et al., 2022) as Obs1 and the observations used for mode-observation comparison in this study as Obs2. The difference between Obs1 and Obs2 are:

1) The different resolution: The resolution of the images used for deriving Obs1 was reduced from 2 m, while for Obs2, we keep its 1-m resolution.

2) The different number of sites: Obs1 is calculated from the data derived from three sites: the Chukchi Sea, East Siberian Sea and Fram Strait. Obs2 only includes 2 sites: the Chukchi Sea and Fram Strait.

3) The different number of cases in every site: For Obs1, there are 14 cases in the Chukchi Sea, 9 cases in the East Siberian Sea and 12 cases in the Fram Strait. For Obs2, there are 24 cases in the Chukchi Sea and 32 cases in the Fram Strait.

We will clearly explain the WIPoFSD model output and any choices about model parameters from the Bateson et al. (2022) study in the revised manuscript.

We applied power law fit for the floe size in each image first and then averaged. The range of floe sizes considered for the fits is 0 m-3000 m for Obs1.

Yes, 3 significant figures are appropriate. If we keep 5 significant figures and set $\alpha = 2.5574$, $[P(\alpha = 2.5574) - P(\alpha = 2.56)]/P(\alpha = 2.56) \approx -1\%$, which only shows a 1% smaller perimeter density.

Half cases pass the goodness-of-fit test.

More details are added in the main text. A table of the power-law fit, uncertainty on alpha and p-value for the goodness-of-fit test of every case will be given in the revised supplement.

14. Eq (2): I don't n(r) has been defined.

Thanks. The definition is added as follows.

n(r) is defined such that n(r)dr is the number of floes per unit area for floe sizes between r and r+dr.

15. L175-177: Rephrase these 2 sentences as they appear to contradict each other.

Thanks. We will move this sentence, "More details on the calculation of $P_i$ is provided in the supporting information Sect. S1", to the end of Section 3.4.

16. Eq (4): gamma needs to be defined.

Thanks for your comments. We added a clear definition of gamma. When adding this definition, we realize that there was a mistake in the value of gamma for the calculation of Pi. As a result, we have revised all equations in both the supplement and main text.

As r applied here is effective floe radius $r_{eff} = \sqrt{a/\pi}$, the radius of a circle that has the same area a as a floe. We assume $p = 2\gamma r_{eff}$. Here the gamma is defined as $\gamma = \frac{p}{2r_{eff}} = \frac{p}{2\sqrt{a/\pi}} = \frac{p\sqrt{\pi}}{2\sqrt{a}}$, where p and a are the perimeter and area of a floe respectively.

The gamma value we give in L23-L24 (the mean floe shape parameter $\gamma$ is 2.27 in the Chukchi Sea region and 2.23 in the Fram Strait region) is $\frac{4a}{d^2}$. Now we have corrected both the equations and the Pi value shown in the results sections as follows.

Here $\gamma$ is a floe shape parameter, $\gamma = \frac{p}{2r_{eff}}$. From the analysis of MEDEA-derived FSD results, the mean floe shape parameter $\gamma$ is 2.17 in the Chukchi Sea region and 1.96 in the Fram Strait region, respectively.

The relationship between the areal FSD f(r) and CFND N(r) is revised to $\int_{r_0}^{\infty} f(r)dr = \int_{r_0}^{\infty} \pi r^2 dN(r)$.

$P_{ice}$ for prognostic model $P_{i\_prog}$ is revised to $P_{prog} = 2\sum_{i=1}^{12} \frac{\gamma f_i(r_{i_{max}} - r_{i_{min}})}{\pi r_i c_{ice}}$.

$P_{ice}$ for WIPoFSD is revised to $P_{wipofsd} = \frac{\int_{r_{min}}^{r_{var}} 2\gamma rn(r)dr}{c_{ice}} = \frac{2\gamma(3-\alpha)(r_{var}^{2-\alpha} - r_{min}^{2-\alpha})}{\pi(2-\alpha)(r_{var}^{3-\alpha} - r_{min}^{3-\alpha})}$.

$P_{ice}$ for observations is revised to $P_{obs} = \sum_{i=1}^{12} \frac{2\gamma A_{floe_i}}{\pi r_i A_{ice}}$.

17. L197: How do you estimate central tendency and spread in the figures provided for Pi?

By applying equation $P_{obs} = \sum_{i=1}^{12} \frac{2\gamma A_{floe_i}}{\pi r_i A_{ice}}$, we calculate the $P_{obs}$ of every images. Then calculate the average of Pi and the standard deviation.

18. Fig 2e-k: there appears to be a plateauing in the observations for small floes. It could be due to a resolution issue as discussed in section 4.2, but it could also be a

signature of the limitation of the power law. I know you are not trying to fit a power law here, but I believe this is an important point to make, especially when comparing to

Thanks for your comments. This is a very interesting point. We will discuss the possibility of the limitation of the power law in the Discussion and Conclusion section.

19. L212: use big O notation for order of magnitude.

Thanks. Revised.

The CPOM-FSD results also show a similar pattern, yet the model $P_i$ values are in much better agreement with the observations for large floes ($r > O$ (10) m), especially during July and August in the Fram Strait region (Figs. 2g and 2h).

20. L215-216: I believe an explanation was provided for the presence of the "uptick". It would be good to mention it here.

Thanks. The explanation is added (see below).

The "uptick" is an artificial feature derived from the model setup of an upper limit of floe sizes and a lack of fragmentation processes for large floes (Roach et al., 2018; Bateson et al., 2022).

21. L254-255: That statement seems to be quite a stretch at this stage and sort of comes out of nowhere. Given the seasons considered, I would expect lateral melting to be a more important contributor to underestimated SIC.

Thanks. In the prognostic models, the floe welding rate is set to be proportional to the square of SIC in the two prognostic models. What we would like to explain here is that the underestimated SIC is a contributor to the underpredicted floe welding rate. Will modify the sentences to make them clear.

The underestimated SIC from the two prognostic models may be a contributor to the underpredicted floe welding rate during spring and early summer.

22. L270: "floes welding are negligible during this season" seems to contradict the statement in the previous subsection (see comment 21).

Yes, the contribution of floe welding to the change of Pi in summer is negligible, i.e., floe welding does not happen in summer. But Bateson (2021) has assessed the effects of floes welding on the FSD in the CPOM-FSD model, suggesting that floe welding occurring in spring can influence the perimeter density in summer (Section 7.2.3 and Figure 7.9 in Bateson (2021)).

23. L334-339: again, I don't find the argument about underpredicting welding very convincing. What about potential overestimation in the data? See Roach et al (2018, The Cryosphere)

Thanks. Roach et al (2018) discuss the reason for the overestimation of sea ice concentration (SIC) as lateral melt. Similar to the response to comment 22, in this study, we discussed the underestimation of SIC will lead to the underestimated floe welding and thus the overestimation of perimeter density of small floes in summer. Our explanation may cause a misunderstanding.

Thanks for your comments. We will modify the sentences to make these causal relationships clear.

24. L243: Again, how was the fit performed? Across all floe sizes or a limited range? What about goodness-of-fit?

Thanks. I guess that you may mean L343.

This exponent is calculated across all floe sizes. By setting the floe size range of 0-3000m, 50% data pass the goodness-of-fit test. We will add this information in the main text and a table including the power-law fit, uncertainty on alpha and p-value for the goodness-of-fit test of every case in the revised supplement.

25. L245-247: What about the possibility that the power law is not appropriate? I don't see why a power fitted through historical data should predict a power law in the current dataset, even at the same location.

Thanks. I guess that you may mean L345-347.

As you mentioned in comment 18, almost flat FSD in the observations for small floes could result from the limitation of image resolution (section 4.2) or a signature of the limitation of the power law. Besides, as in Hwang and Wang (2022), this behaviour may be consistent with the fact that small floes are more vulnerable to lateral melt relative to large floes, which causes the deviation of the small-floe distribution prior to the tail. A discussion of all these possibilities is added in this section in the revised manuscript.

Through historical data, we would like to indicate that the spatial and seasonal variation of the power-law exponent does not only exist in our datasets but also in previous studies. This could emphasize the view that a constant exponent is not appropriate.

**Typographical errors**

26. L58: delete "the".

Thanks. Corrected.

27. "welding" is misspelled a couple of times.

Thanks. Corrected.

**Citation in the response to Referee #2**:

Arntsen, A. E., Song, A. J., Perovich, D. K., and Richter-Menge, J. A.: Observations of the summer breakup of an Arctic sea ice cover, Geophys. Res. Lett., 42, 8057–8063, https://doi.org/10.1002/2015GL065224, 2015.

Bateson, A. W.: Fragmentation and melting of the seasonal sea ice cover, Ph.D. thesis, Department of Meteorology, University of Reading, United Kingdom, 293 pp., 2021.

Bateson, A. W., Feltham, D. L., Schröder, D., Wang, Y., Hwang, B., Ridley, J. K., and Aksenov, Y.: Sea ice floe size: its impact on pan-Arctic and local ice mass and required model complexity, 16, 2565–2593, https://doi.org/10.5194/tc-16-2565-2022, 2022.

Horvat, C., Blanchard-Wrigglesworth, E., and Petty, A.: Observing Waves in Sea Ice With ICESat-2, Geophys. Res. Lett., 47, https://doi.org/10.1029/2020GL087629, 2020.

Hwang, B. and Wang, Y.: Multi-scale satellite observations of Arctic sea ice: new insight into the life cycle of the floe size distribution, Philos. Trans. R. Soc. A Math. Phys. Eng. Sci., 380, https://doi.org/10.1098/rsta.2021.0259, 2022.

Kwok, R.: Declassified high-resolution visible imagery for Arctic sea ice investigations: An overview, Remote Sens. Environ., 142, 44–56, https://doi.org/10.1016/j.rse.2013.11.015, 2014.

Masson, D., & Leblond, P. H.: Spectral evolution of wind-generated surface gravity waves in a dispersed ice field. JournalofFluid Mechanics, 202, 43–81, https://doi.org/10.1017/S0022112089001096, 1989.

Meylan, M. H., Horvat, C., Bitz, C. M., and Bennetts, L. G.: A floe size dependent scattering model in two- and three-dimensions for wave attenuation by ice floes, Ocean Model., 161, 101779, https://doi.org/10.1016/j.ocemod.2021.101779, 2021.

Perovich, D. K.: Aerial observations of the evolution of ice surface conditions during summer, J. Geophys. Res., 107, 8048, https://doi.org/10.1029/2000JC000449, 2002.

Perovich, D. K. and Jones, K. F.: The seasonal evolution of sea ice floe size distribution, J. Geophys. Res. Ocean., 119, 8767–8777, https://doi.org/10.1002/2014JC010136, 2014.

Roach, L. A., Horvat, C., Dean, S. M., and Bitz, C. M.: An Emergent Sea Ice Floe Size Distribution in a Global Coupled Ocean-Sea Ice Model, J. Geophys. Res. Ocean., 123, 4322–4337, https://doi.org/10.1029/2017JC013692, 2018.

Roach, L. A., Bitz, C. M., Horvat, C., and Dean, S. M.: Advances in Modeling Interactions Between Sea Ice and Ocean Surface Waves, J. Adv. Model. Earth Syst., 11, 4167–4181, https://doi.org/10.1029/2019MS001836, 2019.

---

## Author Response (AR1)

**Response to Referee #1**

This paper compares the sea-ice "floe perimeter density," as calculated from three models, to satellite observations in the Chukchi Sea (CS) and Fram Strait (FS).

The length, or density (length per unit area), of floe perimeter is a factor in the lateral melting of ice floes in summer, and is therefore a potential diagnostic for models. For a given field of ice floes, the floe perimeter density is a scalar.

The sea-ice floe size distribution (FSD) is the number of floes as a function of floe size. The FSD may be normalized (e.g. by the total number of floes) or not.

The analysis in this paper is all about perimeter density, denoted $P_i$ (P sub i) by the authors, and PD by this reviewer. However, the authors treat PD and FSD as if they are interchangeable or equivalent. They are not. Completely different FSDs can give rise to the same PD, and identical FSDs can give rise to different PDs. There is not a one-to-one correspondence between PD and FSD. The authors point out that a larger PD implies more smaller floes, and this is true, but the PD says nothing about the FSD. In light of this fundamental confusion between PD and FSD, I must recommend that this paper be rejected. Specific comments follow.

Dear Referee,

We would like to thank you so much for reviewing our manuscript and giving valuable comments and suggestions. Please find our responses to your comments in blue text and revised manuscript in red text below.

We outlined the major changes in this revision as follows:

1) We changed the title to "Summer sea ice floe perimeter density in the Arctic: High-resolution optical satellite imagery and model evaluation";
2) All equations and the Pi value related to floes shape parameter gamma have been corrected in the text and Figures of results sections.

Best regards,

Yanan Wang and co-authors

==========

The title implies that the paper is about the FSD, but it is really about the PD.

To avoid the confusion caused by the definition, we revised the title to "Summer sea ice floe perimeter density in the Arctic: High-resolution optical satellite imagery and model evaluation".

In the Abstract, lines 11-16 are about the FSD, and lines 17-25 are about the PD, without making any connection between the two.

We include the relationship between FSD and perimeter density in the abstract in the revised manuscript as follows.

*In this study, perimeter density has been applied as a proxy for floe size distribution for evaluating three FSD models.*

Suppose n(r) is the number of floes of size r.  Consider the case of circular floes.  The perimeter of each floe is 2*pi*r and the area is pi*r^2.  Therefore the perimeter density is:

   PD = INTEGRAL(2*pi*r * n(r)dr) / INTEGRAL(pi*r^2 * n(r)dr)

Now suppose the mean value of n(r) is MU, and the variance is SIGMA^2.  Then the above equation yields:

   PD = 2*MU / (SIGMA^2 + MU^2)

Now consider two cases:

(1) n(r) has a uniform distribution on [0,L], i.e. n(r) = 1/L.

Then MU = L/2 and SIGMA^2 = (L^2)/12, so PD = 3/L.

(2) n(r) has an exponential distribution with parameter LAMBDA,

i.e. n(r) = (1/LAMBDA) * exp(-r/LAMBDA).

Then MU = LAMBDA and SIGMA^2 = LAMBDA^2, so PD = 1/LAMBDA.

By choosing L = 3*LAMBDA, the uniform FSD has the same perimeter density as the exponential FSD.  Same PD, different FSDs.

   Now consider a set of circular floes with FSD n(r).  Construct a set of elliptical floes with semi-major axis "a" and semi-minor axis "b" such that pi*a*b = pi*r^2.  Each elliptical floe has the same area as its corresponding floe in the circular set.  Therefore the FSD of the elliptical set is also n(r), by construction.  But the perimeters of the elliptical floes are longer than the perimeters of the circular floes, so the PD for the elliptical set is larger than the PD for the circular set.  Same FSD, different PDs.

Same number FSD, different area FSD.

Lines 359-361.  "positive biases of P_i are closely linked to overactive wave fracture in the models.  This suggests accurate parameterisation of wave-induced sea ice breakup is essential for simulating the summer FSD correctly."

The implication here (and throughout the paper) is that the PD tells us about the

FSD. But a connection between PD and FSD has not been demonstrated, and the simple theoretical examples in the previous comment show that a connection need not exist.

Figure 6 caption. "(a) Change of FSD arising from lateral melt" and "(b) ... wave induced FSD change"

According to the scale bar in the figure, the panels show the change in perimeter density, not the change in FSD. But here (and throughout the paper) the authors seem to equate PD and FSD.

1) The same PD corresponding to different number FSDs or the same FSD corresponding to different PDs does not mean there is no connection between PD and FSD. In previous studies, Rothrock and Thorndike (1984) first defined the FSD as both number density $n(r)$ and area fraction $f(r)$. These two definitions are related by $f(r) = \gamma r^2 n(r)$. Now, if we consider a set of circular floes and a set of elliptical floes with semi-major axis "a" and semi-minor axis "b" with the same number FSD $n(r)$ in a given region. But assume that pi*a*b ≠ pi*r^2. In this situation, although these two groups of floes have the same $n(r)$, their areal FSD $f(r)$ is different. Similarly, if we consider 5 circular floes with the same radius of 10 m and 100 elliptical floes with semi-major axis "a=1 m" and semi-minor axis "b = 5 m" in a given region. Then we can get the same areal FSD, different number FSDs. Even the traditional FSD concepts $n(r)$ and $f(r)$ still shows the same situation that there is not always a unique 1:1 relationship between them. Similarly, it is not abundant evidence to prove that Pi is not a metric related to FSD by showing the same PD corresponding to different number FSDs or the same FSD corresponding to different PDs.

2) Additionally, the reason for the same PD corresponding to different number FSDs or the same FSD corresponding to different PDs can also be partially explained by equations (4)-(11) in Roach et al. (2019). They define the notation $\langle r^i \rangle_N$ as the ith moment of the number FSD.

The fractional sea ice concentration in a grid cell, $c$, is

$$c = \int_{r_0}^{r_{max}} \gamma r^2 n(r) dr = \gamma \langle r^2 \rangle_N.$$

then the perimeter density per unite ocean area is

$$P_{ocean} = \int_{r_0}^{r_{max}} 2\gamma r n(r) dr = 2\gamma \langle r^1 \rangle_N,$$

The perimeter density per unite sea ice area is

$$P_{ice} = \frac{P_{ocean}}{c} = \frac{\int_{r_0}^{r_{max}} 2\gamma r n(r) dr}{\int_{r_0}^{r_{max}} \gamma r^2 n(r) dr} = 2 \frac{\langle r^1 \rangle_N}{\langle r^2 \rangle_N}$$

Different metrics have different emphases. The area fraction of floe

$$F(r) = \int_{r_0}^{r_{max}} f(r) dr = \int_{r_0}^{r_{max}} \gamma r^2 n(r) dr = \gamma \langle r^2 \rangle_N,$$

is a positive moment of the number FSD. For $P_{ice}$, Roach et al. (2019) suggested that "$P_{ice}$ is a negative moment of the FSD, weighted more heavily by smaller sizes, so Pi

is more relevant for thermodynamic melting and freezing of floes", which is important in evaluating a FSD model.

3) Although your example and ours given above means that there is not always a unique 1:1 relationship between the number density distribution and perimeter density distribution or between the number density distribution and area fraction distribution. It is also worth noting that most examples are related to the variable shape of floe, i.e., γ is not the same for the two group of circle and elliptical floes given in these examples when calculating the perimeter density. However, as pointing out by previous study, the ratios between floe size properties is generally constant and almost independent of floe size (e.g., Rothrock and Thorndike, 1984; Toyota et al., 2006). Besides, it is common in FSD studies to make assumptions about floe shape, e.g., FSD models assume a fixed shape parameter γ. This should not be a cause for concern.

In summary, the authors have not said how the PD is related to the FSD, and therefore why it can be used to assess the FSD produced by the models.

In the response above, we have given the relationship between the PD and the FSD. The reason why we choose perimeter per unit sea ice area rather than per unit ocean area as in previous observational studies (Perovich, 2002; Perovich and Jones, 2014; Arntsen et al., 2015) can be explained by an example. If we consider a circular floe with a radius r = 5 km in a 10x10 km region. Sea ice concentration $SIC_1 = \frac{\pi}{4}$, $P_{ocean1} = \frac{\pi}{10} km \cdot km^{-2}$ and $P_{ice1} = 0.4 km \cdot km^{-2}$. We also consider 50 circular floes with radius r = 0.1 km in a 10 x10 km region. $SIC_2 = \frac{\pi}{200} < SIC_1$, $P_{ocean2} = P_{ocean1} = \frac{\pi}{10} km \cdot km^{-2}$ and $P_{ice2} = 20 km \cdot km^{-2} > P_{ice1}$. Obviously, perimeter per unit sea ice area is more related to the distribution of small floes. The floe perimeter per unit ice area is a more suitable and important metric for the evaluation of FSD models as the development of the FSD models are particularly important in the marginal ice zone, where small-floe-related processes are active, such as the thermodynamic freezing and melting of sea ice floes. Besides, Denton and Timmermans (2022) also suggested the SIC should be considered in studying the temporal evolution of FSD, e.g., for use in sea-ice models. The use of perimeter density per unit sea ice area for FSD models evaluation corresponds to this result.

According to my calculations, the PD and FSD are not necessarily related, so any statements or conclusions derived from the analysis of the PD do not necessarily apply to the FSD. Since there is no easy way to rectify the confusion between PD and FSD in this paper, it should be rejected.

Regarding your comment 'the PD and FSD are not necessarily related' and 'there is no easy way to rectify the confusion between PD and FSD in this paper', we disagree with these points and have different opinions. In previous studies, the FSD is not only presented as number density. Perimeter density distribution and floe area fraction are also used. Rothrock and Thorndike (1984) first defined the FSD as both number density n(r) and area fraction f(r). Yes, perimeter density distribution is not defined as FSD by

Rothrock and Thorndike (1984), but it is a useful proxy for FSD and widely used in previous FSD studies. For example, previous observational studies (Perovich, 2002; Perovich and Jones, 2014 and Arntsen et al., 2015) applied the total floe perimeter per unit ocean area as a proxy for floe size distribution. As discussed by Perovich (2002), the use of the perimeter of the sea ice floe reduces the impacts of partially captured floes at the edge of the image for FSD retrieval. Besides, perimeter density per unit ice area is also used in FSD models as a metric to show the evolution of the FSD, e.g., Roach et al. (2019) and Bateson et al. (2022). As explained by Roach et al. (2019), "$P_{ice}$ is weighted more heavily by smaller sizes, so Pi is more relevant for thermodynamic melting and freezing of floes", which is important in evaluating the FSD model.

Whilst this paper may benefit from some tightening of nomenclature and clarification, useful information about the FSD can still be obtained from the perimeter density. For example, in Figure 2, we explicitly present the perimeter density distribution for floe size categories (perimeter of floes in floe size category i per unit bin width per unit sea ice area) rather than just perimeter density Pi, which therefore provides additional information about whether it is smaller floes or larger floes that contribute to differences in Pi calculated across the distribution overall.

Comparison of Models and Observations

========================================

The lack of agreement between the models and the observations, and between the models themselves, is truly remarkable.  The histograms of PD are all completely different (Figure 2, panels (a) through (d)).  The plots of PD vs. sea-ice concentration (SIC) (Figure 3) show that FSDv2-WAVE has values of PD that are an order of magnitude larger than the observations, with vastly greater variability, and a slope (vs. SIC) with the wrong sign.  CPOM-FSD is hardly any better.  WIPoFSD has a slope similar to the observations but with PD values five times larger.  The model/obs differences are noted by the authors at lines 196-206 and 219-231.

    In Section 5, the causes of the differences between observations and models are attributed to three factors (lines 322-324).  The first factor, image resolution, cannot explain such large differences (line 327).  The second factor, underestimation of SIC in the models, "can partially explain" (line 338) such large differences.  The third factor, overactive wave fragmentation in the models, was investigated by dividing each region (CS and FS) into north and south portions, and comparing observations vs. models in these sub-regions.  In the CS region, all the observations are in the south sub-region.  In the FS region, all the observations are in the north sub-region.

    In Figure 7(a) for the CS region, the agreement between observations and FSDv2-WAVE(south) is still terrible, and the agreement between observations and CPOM(south) doesn't appear to be any better than in Figure 3(a).  In Figure 7(b) for the FS region, the agreement between observations and FSDv2-WAVE(north) is still terrible, and the agreement between observations and CPOM(north) is not particularly good.  In my

opinion, the analysis by sub-region has not resolved or shed light on the large differences between observations and models.

Lines 352-353. To investigate "unrealistically high perimeter densities in our study regions" the authors "examined the P_i in the northern regions where wave-induced breakup is negligible. In these regions, most modelled P_i match our observations better."

This seems to be saying that the authors have compared model results from the northern sub-regions with the observations. But for the CS region, all the observations are in the south sub-region, so it makes no sense to compare models in the north with observations in the south. For the FS region, Figure 7(b) does not show that "most modelled P_i match our observations better." I don't see any kind of match between models and observations, nor much improvement over Figure 3(b).

In the northern Chukchi Sea and the Fram Strait, the change in Pi arising from all FSD evolution processes is almost zero in the two prognostic models during our research period. For checking the Pi in these regions, we recognize the Pi value in the northern region is the value at the end of early spring or the beginning of our research period. This comparison could help us determine whether the differences between observations and models arising from FSD evolution processes in summer (lateral melt and wave fracture) or in spring.

Thanks for pointing out it. We provided more explanation about the reason for comparing Pi in northern and southern regions in the Chukchi Sea and the Fram Strait. Please see Lines 313–315 and 320–323 in Page 11 in the revised manuscript.

*In the northern Chukchi Sea and the Fram Strait (blue boxes in Fig. 6), the change in P arising from processes driving FSD change during May–August is almost zero in the two prognostic models. The P in the northern regions can be regarded as a fixed value over the period of May–August.*

*The purpose of this comparison is to explore whether the differences between observations and models arising from the inaccurate summer FSD evolution processes simulated by models (lateral melt and wave fracture) or the processes that determine FSD shape prior to summer breakup.*

Lines 359-360. "positive biases of P_i are closely linked to overactive wave fracture in the models."

I don't believe that the authors have demonstrated this.

Looking at the big picture, I can only think of two possible explanations for the enormous differences between the models and the observations, and between the models themselves: either the models are complete junk, or the PD is meaningless as a diagnostic of model performance. Do the authors have any thoughts on this?

Thanks for your comments.

Our strong views are that evaluation of FSD models provides us with key insights for model development through the analysis of model biases and their causes, thus allowing us improving sea ice models performance for climate research and operational applications. Until now, there have had been no comprehensive study comparing modelled and observed FSD. The reason for this was the absence of reliable satellite products resolving sea ice floes with the seizes from metres to thousands of metres, available for the timescales from seasons to the interannual. For the first time with this study, the evaluation of FSD simulations has become available for the widely used by the research and operational communities' sea ice models from the analysis of high-resolution satellite imagery, spanning more than 10 years.

Admittedly, both the prognostic FSD models and power-law FSD model are still as in the development, as much as satellite FSD-detection techniques are still finding their feet. A pertinent observational benchmarking can improve the model design, as it had been demonstrated by accounting for brittle sea ice fracture on hundred-meter scale (Bateson et al., 2022). This is also shown in Fig. 2h in our manuscript that CPOM-FSD (yellow lines) including a brittle fracture scheme shows good performance in simulating Pi for floes with radius at hundreds of meter scales.

We have also tested areal FSD as a validation metric, which shows a similar differences between models and observations (models show overestimation for small floes and underestimation for larger floes in general). Please see the figure below, which is a repeat of Figure 2e-2k from the manuscript, except the plots shown are for area-weighted FSD. We do not see any substantial difference in conclusions when this alternative metric is used. The differences between models and observations are not caused by the metric used. Please see the reasons for using the metric of perimeter per unit sea ice area, in our response above.

[Figure]

Other Comments

==============

About the MEDEA imagery used in this study (lines 59 and 88-90), please see Denton and Timmermans (2022), and say briefly how their data set and analysis relates to the present work.

It is an interesting study. The SIC-FSD relationship could give evidence about the processes governing FSD evolution and give us more confidence in the use of perimeter density per unit sea ice area rather than per unit ocean area. Their test of FSD sensitivity to this choice reveals that the FSD can appear divided into two power-law regimes if this choice does not adequately identify small floes. This result corresponds to the difference between Pi derived from 1-m and 0.5-m resolution images (Fig. 4 in our manuscript). Better identifying small floes could expand the power-law range. Similar to the response to Referee #2, there are two possibilities that may lead to these two regimes. The first one is the limitation of the power-law hypothesis (see details in Montiel and Mokus, 2022). The other one is that small floes are more vulnerable to lateral melt relative to large floes, which causes the deviation of the small-floe distribution prior to the distribution tail (see details in section 4(d) in Hwang and Wang, 2022).

Lines 79-81. "the observations from the Chukchi Sea region captures a more dynamic and fragmented ice condition (e.g., Fig. 1b), while the observations from the Fram Strait capture a less dynamic environment (e.g., Fig. 1c)."

Looking at Figure 1, I don't see that the Chukchi Sea image indicates more dynamic and fragmented ice than the Fram Strait image. The two images look similar to me. How does a single image convey dynamics?

Fig. 1b shows more fragmented sea ice condition and includes more small floes between mid-size floes relative to Fig. 1c.

We have replaced this figure with their partially enlarged image cuts in Figure 1 at Line 635 in the revised manuscript, which can show more details of the difference between them.

Section 3.1.2. What measure of floe size is used? All I can gather is that floe size is characterized by a radius. Is it half the mean caliper diameter? Is it the radius that a circular floe of the same area would have?

We define floe size as floe effective radius, the radius of a circle that has the same area as a floe. We have added this information about the measure of floe size to the revised manuscript (Lines 188–189, Page 7).

*In this study, the floe effective radius $r = \sqrt{(a/\pi)}$ is used to define floe size, which is the radius of a circle that has the same area, a, as a floe.*

Also, at lines 103-105, "we first applied combined filters: median, bilateral and Gaussian filter" and "The smoothing term, KGC algorithm parameter, was set as 0.0001" -- either

provide more detail so that the reader can understand what this means, or leave it out and just refer to Hwang et al (2017).

Thanks, we will just cite Hwang et al (2017) in the revised manuscript (Lines 118, Page 4).

*For the FSD retrieval, we applied the same filter parameter and KGC algorithm parameter as in Hwang et al. (2017b).*

Section 3.1.3.  What is the spatial resolution of the sea-ice data?  Also, the analysis period is 2000-2014 but AMSR-E is only available 2002-2011 and AMSR-2 is only available since 2012. What sea-ice products (with what resolution) were used during what time periods?

The resolution for AMSR-E and AMSR2 applied in this study is 6.25 km. The resolution for NSIDC sea ice concentration is 25 km.

Yes, in our study period, there isn't SIC derived from AMSR-E in April- August 2000-2001, April-May in 2002 and April-June in 2012. That is one of the reasons that we also applied the SIC from NSIDC, covering April to August during 2000-2014.

Thanks for pointing out this, we have added the explanation of the resolution of the SIC products and the period for each product that we used for analysis in the revised manuscript (Lines 123–127, Page 5).

*Two types of SIC products were used in this study: National Snow and Ice Data Center (NSIDC) SIC of 25 km spatial resolution (Meier et al., 2017; Peng et al., 2013) and ARTIST Sea Ice (ASI) algorithm 6.25 km SIC (Spreen et al., 2008; Melsheimer and Spreen, 2019, 2020). The collected SIC data cover between May and July over the analysis period of 2000–2014. We note that ASI algorithm SIC is unavailable in April-August 2000-2001, April-May in 2002 and April-June in 2012. During these periods, we only applied the NSIDC SIC for the model-observation comparison.*

Lines 121-123.  Does 1-degree grid mean 1-degree in latitude and 1-degree in longitude?  What does "gx1v6" mean (line 122)?  Also, the models are run "for 37 years from 1 January 1980, followed by a 10-year period spin-up" so the spin-up period is 2017-2026 i.e. partially in the future.  Is that correct?

Yes, 1-degree grid means 1 degree in latitude and 1 degree in longitude, which is around 30-63 km per gird. gx1v6 is one of the displaced pole grids of the global model, which is approximately 1-degree resolution horizontally (320x384 for horizontal grid with the horizontal resolution of nominal 1 degree). Thanks for pointing out this inaccurate description. The spin-up period is the first 10 years, which is 1980-1989.

We have added a more detailed explanation of the grid used in models and spin-up period in the revised manuscript (Lines 136–140, Page 5).

*FSDv2-WAVE model has the displaced gx1v6 grid (320x384 for horizontal grid) over a global domain, which is approximately 1-degree resolution horizontally. The other two models are initiated with the ice-free Arctic and run with the tripolar 1° (129 × 104)*

*grids for 37 years from 1 January 1980, including a 10-year period spin-up during 1980–1989 in a pan-Arctic domain excluding Hudson Bay and the Canadian Arctic Archipelago.*

Lines 155-157. "the model also simulates FSD evolution through the floe size parameter r_var ..."

Please define r_var. I see that it varies between r_min and r_max, and I see that it evolves according to four FSD processes, but no definition of r_var is given. What is it?

The $r_{var}$ is defined in the equations 17-19 in Bateson et al. (2020). We will quote that the definition of r_var is given in Bateson et al. (2020). Please see Lines 183–186 on Pages 6–7 in the revised manuscript.

*In addition to the exponent, the model also simulates FSD evolution through the floe size parameter $r_{var}$, varying between minimum floe radius $r_{min}$ and maximum floe radius $r_{max}$ in the distribution. $r_{var}$ evolves according to four FSD processes: lateral melt, wave-induced fracture, floe growth in winter and ice advection (Bateson et al., 2020, 2022). The full details of $r_{var}$, including a physical interpretation of this value, are provided in Bateson et al. (2020).*

Section 3.3. This section (FSD definition, lines 158-171) is confusing and unnecessarily complicated.

-- "The FSD is usually defined as the floe areal FSD..."

It's confusing to use FSD in the definition of FSD!

-- "By integrating f(r) over floe radius between r and r+dr, f(r)dr (dimensionless) is obtained"

This makes no sense.

Thanks. We revised the definition of FSD at Lines 189–192 on Page 7.

*The FSD is usually defined as the fractional-area distribution, $f(r)dr$, and number-density distribution, $n(r)dr$ (Rothrock and Thorndike, 1984; Toyota et al., 2006; Perovich and Jones, 2014; Horvat and Tziperman, 2015; Zhang et al., 2015; Hwang et al., 2017b; Bateson et al., 2020), corresponding to the area and the number of floes per unit ocean surface area with radius between $r$ and $r + dr$.*

-- It's really not necessary to introduce the Heaviside function and equation (1) in order to define the FSD, especially since they're not used in the rest of the paper.

-- The cumulative floe number distribution, defined at lines 169-171, is also not used in the rest of the paper.

Thanks for your comment. We will delete the definition of F(r) in equation (1) and only keep the definition of n(r)dr and f(r)dr at Lines 189–192 on Page 7.

*The FSD is usually defined as the fractional-area distribution, $f(r)dr$, and number-density distribution, $n(r)dr$ (Rothrock and Thorndike, 1984; Toyota et al., 2006; Perovich and Jones, 2014; Horvat and Tziperman, 2015; Zhang et al., 2015; Hwang et al., 2017b; Bateson et al., 2020), corresponding to the area and the number of floes per unit ocean surface area with radius between $r$ and $r + dr$.*

Line 207. "normalized" perimeter density is not defined in the text. The caption for Figure 2 says "The normalized perimeter density distributions were obtained by dividing the width of every floe size category into P_i at each region." The original P_i has units of 1/km and the "normalized P_i" has units of 1/km^2. In what way is this a normalized quantity? Usually I think of normalization as producing a dimensionless result such as a percentage. What is the point of "normalizing" by dividing by the width of the floe size category to produce another dimensional quantity?

Thanks. Yes, the "normalized" perimeter density distribution is not normalized to 1. $P_i$ is defined as the perimeter of floes per unit sea ice area (units: km km$^{-2}$). We will define the "normalized" perimeter density $p_{ice}$ (units: km$^{-2}$) as the perimeter of floes in floe size category i per unit bin width per unit area in the revised manuscript (Lines 233, Page 8). So, the integration of p(r) will be perimeter density $P_{ice}$, $P_{ice} = \int_{r_i}^{r_{i+1}} p_{ice}(r)dr$. In Figure 2, what we want to show is the perimeter density at the midpoint of each bin rather than Pi, the integral of the perimeter density over each category. That's why we produce $p_{ice}$.

*Figs. 2e–2k show the comparison of P per floe size category, i.e., the perimeter of floes per unit sea ice area per unit bin width.*

Lines 270-271. "we constructed two data sets: monthly changes of P_i arising from lateral melt and FSD changes arising from wave breakup." How were these two data sets created?

These two variables are from model outputs. For the FSDv2-WAVE model, please see the first two terms on the right-hand side in equation (14) given by Roach et al. (2018), representing the growth and melt of existing floes in thickness and lateral size. For CPOM-FSD, the fraction of sea ice area lost due to lateral melting is shown in equation (8) in Bateson et al. (2022). The change in the FSTD f(r, h), per unit time due to fracture by ocean surface waves in both FSDv2-WAVE and CPOM-FSD is given in equation (20) by Roach et al (2018). Thanks for pointing out this, we will add the information about these two variables at Line 297 on Page 10 in the revised manuscript.

*To test the effects of lateral melt and wave breakup, we acquired two data sets from model outputs (Fig. 6): monthly changes of P arising from lateral melt and FSD changes arising from wave breakup.*

Line 274. "CPOM-FSD produces negative changes in P_i from wave fracture (Figs. 6f and 6h)." But Fig. 6h shows changes arising from lateral melt, not wave fracture. Furthermore, Figs. 6e and 6f show that the change is positive, not negative. Finally, note that the panels in the bottom row of Fig. 6 are labelled (g) (e) (h) (f) from left to right. This might be the source of some confusion.

Line 280. "CPOM-FSD shows a stronger reduction in P_i arising from lateral melt (Figs. 6e and 6g)." But Fig. 6e shows changes arising from wave fracture, not lateral melt. See also the previous comment.

Thanks for pointing out this. Have revised at Lines 298–300 on Page 10 and the label of the bottom panels in Fig. 6 at Line 675 on Page 26 in the revised manuscript.

*The results show that FSDv2-WAVE produces larger, positive changes in P from wave fracture (Figs. 6b and 6d) in summer relative to CPOM-FSD (Figs. 6f and 6h).*

Lines 302-303. "The close match between CPOM-FSD and the observations for the northern Fram Strait region..."

I'm looking at Figure 7(b) for the Fram Strait region. The observations (black circles) were acquired in the northern part of the region. The CPOM(north) results are indicated by open yellow circles. I don't see a close match between the black circles and the open yellow circles.

Thanks. The Pi in the northern Fram Strait simulated by CPOM-FSD is of the same order of magnitude of observations. We provided a more rigorous description at L332-L334 on Page 11.

*The P in the northern Fram Strait simulated by CPOM-FSD is of the same order of magnitude of observations, indicating a closer match between CPOM-FSD and the observations in the northern Fram Strait region than the southern region. This suggests that no significant wave fracture and lateral melt has occurred in the observation site.*

Lines 314-316. "The observation results show clear regional differences between the two study regions, i.e., much larger perimeter density P_i (smaller floes) in the Chukchi Sea region than in the Fram Strait region."

I'm looking at Figs. 3(a) (Chukchi Sea) and 3(b) (Fram Strait). The observations are indicated by black circles. When I look back and forth at the black circles in (a) and (b), I just don't see the "clear" regional differences.

In lines 196-197, observations show a substantial difference in P_i between the two regions (t-test, $t\,(47) = 6.43$, $p < 0.001$) (Fig. 2a). It shows a significantly higher P_i of $20.77 \pm 6.54$ km km$^{-2}$ in the Chukchi Sea site than the P_i of $12.16 \pm 3.79$ km km$^{-2}$ in the Fram Strait site (Fig. 2a). Regional differences shown in observations is at tens km$^{-1}$ scale. For the black circles in 7(a) and 7(b), due to the overestimation of Pi in models, the y-axis limits for Fig. 7 range from 0 to 350 km km$^{-2}$. At this range, the regional difference in observations is not clearly shown in this plot. But the result from the t-test could statistically support this view.

Lines 320-322. "The observations and WIPoFSD model both show a positive correlation between SIC and P_i ... while the two prognostic models show the opposite (negative) correction."

(Note that the word "correction" should be "correlation").

Thanks. We revised this at Line 352 on Page12 in the revised manuscript.

*The observations and WIPoFSD model both show a negative correlation between SIC and P (i.e., smaller floes in a lower SIC), while the two prognostic models show the opposite (positive) correlation.*

In Figure 3, I see the opposite of what's stated here: observations and WIPoFSD both show NEGATIVE correlations between SIC and P_i; FSDv2-WAVE and CPOM-FSD both show POSITIVE correlations between SIC and P_i.

Thanks. We have corrected at Lines 351–353 on Page 12.

*The observations and WIPoFSD model both show a negative correlation between SIC and P (i.e., smaller floes in a lower SIC), while the two prognostic models show the opposite (positive) correlation.*

Supplementary Materials

========================

Equation (1) and following.

-- It's bad notation to use "i" as a subscript on the left-hand side and "i" as an index of summation on the right-hand side. See also my comments below about equation (3) of the main text.

Thanks. We changed $P_i$ to P now.

-- The parameter GAMMA is not defined. "Here GAMMA is a floe shape parameter, for example..." (line 22) -- this is not a definition. I gather from equation (1) that the floe perimeter is 2*GAMMA*r, which perhaps defines GAMMA (if so, that should be stated explicitly). In that case, for circular floes, GAMMA = pi, as noted on line 22, but for square floes, GAMMA is not equal to 1, as erroneously stated on line 22. Consider a square floe of side s, perimeter 4*s, area s^2. If r is the radius of a circular floe of the same area, then s^2 = pi*r^2. The perimeter of the square floe is 4*s = 4*sqrt(pi)*r. If this is equal to 2*GAMMA*r then GAMMA = 2*sqrt(pi) = 3.54.

As r applied here is effective floe radius $r_{eff} = \sqrt{a/\pi}$, the radius of a circle that has the same area a as a floe. We assume $p = 2\gamma r_{eff}$. Here the gamma should be defined as $\gamma = \frac{p}{2r_{eff}} = \frac{p}{2\sqrt{a/\pi}} = \frac{p\sqrt{\pi}}{2\sqrt{a}}$, where p and a are the perimeter and area of a floe respectively. The

gamma value we give in L23-L24 (the mean floe shape parameter γ is 2.27 in the Chukchi Sea region and 2.23 in the Fram Strait region) is $\frac{4a}{d^2}$, which corresponda to 4*0.66=2.64 in Rothrock and Thorndike (1984). Thanks for pointing out this, which make us realize that there was a mistake in the value of gamma for the calculation of Pi. This modification led to changes in the value of Pi. But there isn't any substantial difference in the comparison results when the value of gamma is modified.

Now we have added the definition of gamma and corrected both the equations and the Pi value shown in the text and Figures of results sections. Please see Lines 201, 204-206, 209, 217, 223-224, 229-234 and Figures 2, 3, 6 and 7 in the revised manuscript. For the supplementary materials, please see Line 54, Figure S1 and Table S1.

*Here γ is a floe shape parameter, the ratio of floe perimeter to twice the floe effective radius. For example, $\gamma = \pi$ is for circular floes. From the analysis of MEDEA-derived FSD results, the mean floe shape parameter γ is 3.60 in the Chukchi Sea region and 3.69 in the Fram Strait region, which will be used for the calculation of P in this study. Due to the limitation of image resolution in capturing the shapes of small floes, the floes of d < 5 m are discarded for calculating γ.*

-- Rothrock and Thorndike (1984, hereafter RT84) calculated a "shape parameter" similar to GAMMA, finding that AREA / MCD^2 = 0.66 +/- 0.05, where AREA is the area of a floe and MCD is its mean caliper diameter. In the present context, if AREA = GAMMA*(r^2) and MCD = 2*r then the RT84 shape parameter is GAMMA/4 which implies GAMMA = 2.64 +/- 0.20, which is not too different from the values given on lines 23 and 24.

For $\gamma = \frac{p}{2r_{eff}} = \frac{p}{2\sqrt{a/\pi}} = \frac{p\sqrt{\pi}}{2\sqrt{a}}$, the related value given by Rothrock and Thorndike is $\frac{a}{p^2} = 0.06$, which means $\gamma = \frac{p\sqrt{\pi}}{2\sqrt{a}} \approx 3.62$ in Rothrock and Thorndike (1984). This ratio is for floes with diameters over about 1 km. For our data, if we ignore the floes with caliper diameters smaller than 1 km, $\gamma = 4.57$ in the Chukchi Sea and $\gamma = 4.55$. For $r_{eff} > 5$ m, $\gamma = 3.60$ in the Chukchi Sea and $\gamma = 3.69$. Although the value of γ we used in this study is similar to the value given by Rothrock and Thorndike (1984), it is actually the shape factor for different size of floes. The corrected gamma value is given Lines 204–206 on Page 7 in the revised manuscript.

-- The meaning of the terms in equation (1) should be explained more clearly. For example: (r_i_max - r_i_min) is the bin width of the i-th bin; n_i is the number of floes in bin i per unit bin width per unit area; therefore their product is the number of floes in bin i per unit area. Therefore the quantity inside the summation is the perimeter of the floes in bin i, per unit area, and the numerator is the total floe perimeter per unit area. After dividing by the sea-ice concentration c_ice, one obtains the total floe perimeter per unit area of sea-ice -- the floe PD.

Thanks for your suggestions. We revised the explanations at Lines 21–24 on Page 1 in the revised supplementary materials.

*where $r_i$ and $w_i$ are the midpoint and bin width for each floe size category i. $n_i$ ($m^{-3}$) is the number of floes in floe size category i per unit bin width per unit area. So $n_i w_i$ is the number of floes per unit area for each floe size category i. The upper term in the right hand of Eq. S1 is the perimeter of floes per unit area. By dividing total floe perimeter per unit ocean area by the sea-ice concentration, $c_{ice}$, we obtain the total floe perimeter per unit sea ice area.*

-- Lines 23-24, "From the analysis of MEDEA-derived FSD results..." This sentence belongs in the section on "Calculation of P_i from the observations" at line 47, not in the section on the calculation of P_i from models.

Prognostic models didn't output perimeter density, we also need a gamma for calculating perimeter density in equation (4), which is revised as $P_{prog} = 2 \sum_{i=1}^{12} \frac{\gamma f_i (r_{i_{max}} - r_{i_{min}})}{\pi r_i c_{ice}}$. Here we use the same value as in observations for the same floe size range.

Equation (4) (line 33) is the same as equation (3) of the main text (line 181). Also, lines 42-46, including equation (8), are an exact repeat of lines 184-188 of the main text, including equation (4) there. Why repeat the same material in the Supplement?

Thanks for your comments. We deleted the same equation as in the main text.

Minor Comments

===============

Line 173. The units of perimeter density are given as 1/meter but in much of the rest of the paper the units are 1/kilometer. Figures 3 and 7 use 1/km but Figure S1 uses 1/m.

Thanks. We corrected the unit in Figure S2.

Equation (3) -- notation.

-- On the right-hand side, the index "i" is used in the summation, and on the left-hand side, the index "i" is used as a subscript on P. This is bad notation.

-- Same comment for equation (5) and most of the equations in the Supplementary Materials.

-- The bad notation is easily fixed by dropping the subscript "i" on the perimeter density -- just use "P" (line 173 and following). There's no reason for a subscript.

Thanks. We revised them to P, $P_{prog}$, $P_{wipofsd}$, $P_{obs}$ in equation 1–3 on Page 7-8 in the revised manuscript.

-- Also, doubly-subscripted variables r_i_max and r_i_min are confusing and

unnecessary. The quantity (r_i_max - r_i_min) is just the bin width of the i-th bin or category. It could be denoted w_i or something else with a single subscript i.

Thanks. We replaced the bin width $r_{i_{max}} - r_{i_{min}}$ as $w_i$ in all equations in the revised main text and supplementary materials.

Equation (4). GAMMA is not defined. Also, I believe ALPHA = 2.56 in this case, which is perhaps worth repeating here.

Thanks. We defined gamma at Lines 203-206 and repeat the value of alpha in the revised main text at Line 210.

*Here $\gamma$ is a floe shape parameter, the ratio of floe perimeter to twice the floe effective radius. For example, $\gamma = \pi$ is for circular floes. From the analysis of MEDEA-derived FSD results, the mean floe shape parameter $\gamma$ is 3.60 in the Chukchi Sea region and 3.69 in the Fram Strait region, which will be used for the calculation of P in this study.*

*In this equation, the power-law exponent was set as a constant, $\alpha = 2.56$, for WIPoFSD.*

Figure 1 caption. The date for panel (b) should probably be 12 June, not 6 June.

Thanks. Revised.

Figure 2. In panels (e) through (k), what is the meaning of "Effective" floe radius?

Thanks. In this study, we define the floe size as effective radius $r = \sqrt{a/\pi}$, the radius of a circle that has the same area, $a$, as a floe. We now defined effective floe radius at Lines 188-189 on Page 7 in the revised manuscript.

*In this study, the floe effective radius $r = \sqrt{(a/\pi)}$ is used to define floe size, which is the radius of a circle that has the same area, a, as a floe.*

Table 2. The "a" and "b" superscripts are missing from the table.

Thanks. Corrected.

Table 3. The caption says "FSDv2-WAVE and WIPoFSD" but the table itself lists FSDv2-WAVE and CPOM-FSD.

Thanks. Corrected in Table 3 on Page 30 in the revised manuscript.

**Citation in the response to Referee #1**:

Arntsen, A. E., Song, A. J., Perovich, D. K., and Richter-Menge, J. A.: Observations of the summer breakup of an Arctic sea ice cover, Geophys. Res. Lett., 42, 8057–8063, https://doi.org/10.1002/2015GL065224, 2015.

Bateson, A. W., Feltham, D. L., Schröder, D., Hosekova, L., Ridley, J. K., and Aksenov, Y.: Impact of sea ice floe size distribution on seasonal fragmentation and melt of Arctic sea ice, Cryosph., 14, 403–428, https://doi.org/10.5194/tc-14-403-2020, 2020.

Bateson, A. W., Feltham, D. L., Schröder, D., Wang, Y., Hwang, B., Ridley, J. K., and Aksenov, Y.: Sea ice floe size: its impact on pan-Arctic and local ice mass and required model complexity, 16, 2565–2593, https://doi.org/10.5194/tc-16-2565-2022, 2022.

Perovich, D. K.: Aerial observations of the evolution of ice surface conditions during summer, J. Geophys. Res., 107, 8048, https://doi.org/10.1029/2000JC000449, 2002.

Perovich, D. K. and Jones, K. F.: The seasonal evolution of sea ice floe size distribution, J. Geophys. Res. Ocean., 119, 8767–8777, https://doi.org/10.1002/2014JC010136, 2014.

Roach, L. A., Horvat, C., Dean, S. M., and Bitz, C. M.: An Emergent Sea Ice Floe Size Distribution in a Global Coupled Ocean-Sea Ice Model, J. Geophys. Res. Ocean., 123, 4322–4337, https://doi.org/10.1029/2017JC013692, 2018.

Roach, L. A., Bitz, C. M., Horvat, C., and Dean, S. M.: Advances in Modeling Interactions Between Sea Ice and Ocean Surface Waves, J. Adv. Model. Earth Syst., 11, 4167–4181, https://doi.org/10.1029/2019MS001836, 2019.

Toyota, T., Takatsuji, S., and Nakayama, M.: Characteristics of sea ice floe size distribution in the seasonal ice zone, 33, L02616, https://doi.org/10.1029/2005GL024556, 2006.

**Response to Referee #2**

First, I would like to apologise to the authors and editor for the delay in submitting my review. The manuscript attempts to compare floe perimeter density data obtained via satellite imagery at 2 locations in the Arctic Ocean against the predicted density of 3 floe size distribution (FSD) models, which all include a parametrisation of wave-induced fracture. The goal is to evaluate the performance of the models. Results show that the discrepancy is quite significant in a number of ways. In particular, the models generally predict much larger perimeter densities than the observations, meaning an overestimation of small floes. The authors then attempt to explain the discrepancy by discussing potential issues with specific parameterised components of the models considered.

The manuscript presents original work, is overall well written and the topic is highly relevant to improving the modelling capabilities of FSD resolving sea ice models. That said I have a number of important issues with the way the study has been designed, which may explain the poor agreement between models and data. These are detailed below as well as other comments. I therefore recommend the manuscript undergoes major revisions before it can be further considered for publication.

Dear Dr Fabien Montiel,

We would like to thank you for providing valuable comments and suggestions. Please find our responses to your comments in blue text and revised manuscript in red text below.

We would like to thank you for providing valuable comments and suggestions. Please find our responses to your comments in blue text and revised manuscript in red text below.

We outlined the major changes in this revision as follows:

3) We changed the title to "Summer sea ice floe perimeter density in the Arctic: High-resolution optical satellite imagery and model evaluation";

4) All equations and the Pi value related to floes shape parameter gamma have been corrected in the text and Figures of results sections.

Best regards,

Yanan Wang and co-authors

**Main comments**

1. My main concern relates to the way the study regions for both models and observations were selected in relation to one another. If I understand correctly, the satellite images were chosen at 2 specific locations, one in the Chukchi Sea and one in the Fram Strait. In contrast, the regions selected for analysing model outputs are

much larger. They do include the specific observational locations but extend over much wider regions, selected to include the ice edge. My issue is that the variability in FSD in these regions is likely to be much larger than that at the locations of the data. If the data is collected far from the ice edge (which could have been estimated), floe sizes are likely much larger than closer to ice edge. Therefore, I am not sure the comparison between model outputs and data is fair with respect to the models. In fact, when the comparison is refined to a subdivision of the initial study regions, the agreement is improved.

Thanks for your comments and suggestions. For models, we have tried to select a small region (5×5 grid cells) for comparison. However, for the comparison between observations and several models that have different atmospheric forcing, there are very different ice conditions within this small region, e.g., one model shows MIZ but the other two show interior ice pack. This will cause an unfair comparison between the model and observations. These biases related to the general sea ice conditions are due to the biases in atmospheric forcing and the absence of ocean beneath in models. We try to counter-weight these by using a wider area for comparison.

For observations, all MEDEA images used in this study in the Chukchi Sea and the Fram Strait are in the same fixed location, 70°N, 170°W for the Chukchi Sea and 84.9°N, 0.1°E for the Fram Strait. However, although the images applied in this study are in the fixed site, different imagery data can also indicate the perimeter density at different distances to the ice edge. For example, the sea ice concentration for 7 out of 13 cases in June in the Chukchi Sea is below 80%, indicating MIZ in the images. And the remaining 6 June cases indicate interior ice pack. Hence, when we compare the monthly mean (as the solid lines in Figure 2e-2k) rather than the case-by-case comparison, it is the comparison of the mean state of perimeter density both near the ice edge and far from the ice edge for both models and observations.

I am not an expert in analysing satellite imagery, so my following suggestion could be naïve and/or uninformed, but this is what I would have done: (i) select all the imagery available in the model study regions, not just those at the specific locations selected, OR, if this is too much data to analyse, (ii) take a random sample of images spanning the study regions. Either way, the variability in the data would be a lot more representative of that predicted by the models. I am not suggesting that the authors redo the entire analysis for this paper, but I feel like this limitations needs to be given a lot more emphasis in the manuscript. At the very least, bring this up in the Discussion section, but what would be even better is to add a sub-section looking at the comparison models/data when the model outputs are only selected in a smaller region around the data locations, even more localised than the subdivision shown in figure 6.

Thanks for pointing out this limitation. There are some extra MEDEA images of the drifting ice parcels available in the location of sea ice buoys within the study regions, which is not in the fixed locations of "fiducial sites". However, for this study, we believe the variation of observations is enough for the model-observation comparison study. Thanks for your helpful suggestions. In future study, these images could be used to expand the spatial variability in the observations. We added

more explanations about these limitations in the Discussion and Conclusion section including the comparison results within 5×5 grid-cells regions for models. Please see Lines 356-366 on Page 12 in the revised manuscript and Figure S2 and Table S2 in the supplementary materials.

*The satellite observation sites are fixed and much smaller than the regions selected for models, which include grid cells in both MIZ and interior pack ice. Nevertheless, the observations also include a range of different sea ice conditions. For example, the sea ice concentration of half of the cases in the Chukchi Sea (Figure 2i–2k) is below 80%, indicating almost half of the images within the MIZ and the other half within the interior ice pack. Therefore, the monthly means from the models (as the solid lines in Figure 2e-2k) can be comparable to the observations in the Chukchi Sea. In the Fram Strait, most of the observations represent the pack ice (>80% SIC), while the model outputs mainly represent the MIZ conditions (<80% SIC). To evaluate the effects of the size of modelling study regions, we reduced the size of the model domains down to a 5×5 grid-cells region at the centre of the observation sites, which shows the same overestimation of P and P of small floes (see Figure S2 and Table S3). Based on this evaluation, we decided to use a larger model domain to better capture spatial variability caused by different external forcing fields in the models.*

2. The title of the manuscript is misleading and should be changed, as the authors do not analyse the FSD but the floe perimeter density, which is different. The abstract also needs some work as it starts with statements on the FSD and then switches to perimeter density without making a link between them.

We will revise the title to " Summer sea ice floe perimeter density in the Arctic: High-resolution optical satellite imagery and model evaluation" and include the relationship between FSD and perimeter density in the abstract.

*In this study, perimeter density has been applied as a proxy for floe size distribution for evaluating three FSD models*

I am not saying the perimeter density is a bad metric, but it is not the one advertised! It took some time to fully appreciate the meaning of Pi (again not an expert!). It would have helped me to show the Pi results with units km per km^2.

We have changed the units of Pi to km km$^{-2}$ to make it easier to understand the meaning of Pi.

I would have liked the relationship between these 2 quantities discussed in more details in relation to the results.

In previous observational studies (Perovich, 2002; Perovich and Jones, 2014 and Arntsen et al., 2015), perimeter per unit area has been applied as a proxy for floe

size distribution. As discussed by Perovich (2002), the use of the perimeter of the sea ice floe reduces the impacts of partially captured floes at the edge of the image for FSD retrieval. Besides, perimeter density per unit ice area is widely used as a metric to show the evolution of the FSD, e.g., Roach et al. (2019) and Bateson et al. (2022). As explained by Roach et al. (2019), "$P_{ice}$ is weighted more heavily by smaller sizes, so Pi is more relevant for thermodynamic melting and freezing of floes", which is important in evaluating the FSD models. For more details on the discussion of the relationship between floe number density N(r), floe area fraction F(r) and perimeter density P, please see our response to referee #1.

For instance, do we expect FSDs to have similar qualitative and quantitative properties as the perimeter density distributions shown in figs 2e-k?

As in our response to referee #1, we have also tested areal FSD as a validation metric, which shows a similar difference between models and observations (models show overestimation for small floes and underestimation for larger floes in general). Please see the figure below, which is a repeat of Figure 2e-2k from the manuscript, except the plots shown are for area-weighted FSD.

[Figure]

**Other comments**

3. L28-30: This statement is confusing, especially for the uninformed reader. Quantifying the MIZ is still an active an area of research. Mentioning the 2 definitions (i.e. wave-based and SIC-based) in this way makes it seem that they are equivalent, but that's not true (see Brouwer et al paper). I suggest you pick one definition or expand the discussion.

Thanks for your suggestions. We expanded the discussion in the revised manuscript (Lines 28-35, Page 1).

*This results in the changing marginal ice zone (MIZ), defined as the ice-covered region affected by waves and swell by the World Meteorological Organization*

*(WMO, 2014). Another alternative definition of MIZ is a sea ice-covered area with sea ice concentration (SIC) of 15%–80% (e.g., Strong and Rigor, 2013; Aksenov et al., 2017; Rolph et al., 2020; Bateson et al., 2020; Horvat, 2021). Several SIC products are available to define the MIZ, whereas observing waves in sea ice using satellite-derived observations is still an ongoing area of research. Similarly, sea ice modelling studies often do not include an explicit representation of waves in sea ice. Although this SIC-based definition of MIZ is not directly related to the dynamics in this region (Dumont, 2022), due to the lacking techniques for detecting wave-ice interactions (Horvat et al., 2020), this definition is widely applied in previous studies.*

4. L43: "viscous dissipation" is not the process governing attenuation, it refers effective/homogenised dissipative rheology of continuum viscous layer models often used to approximate attenuation caused by a non-homogeneous ice cover. In any cases, "dissipative processes" would be a better choice of wording here.

Thanks for your suggestions. We changed the wording in the revised manuscript (L46, Page 2).

*The FSD affects the ocean surface waves propagation and attenuation through the ice, i.e., floes smaller than a characteristic wavelength of swells attenuate wave energy through dissipative processes, while larger floes attenuate the wave energy through scattering (Kohout and Meylan, 2008; Williams et.al., 2013a; Thomson and Rogers, 2014; Montiel et al., 2016; Meylan et al., 2021; Dumas-Lefebvre and Dumont, 2021; Horvat and Roach, 2022).*

5. L53-55: The limitations of the power-law should be discussed. See Montiel & Mokus (2022, Philosophical Transactions A) for an overview.

Thanks for your suggestions. We added more discussion about the limitation of the power-law hypothesis in the revised manuscript (L59-63, Page 2).

*However, the floe size power-law hypothesis has been contested by recent observations (Steer et al., 2008; Herman, 2010; Herman et al., 2021), which suggest that different shapes or functions can be better in describing FSDs. Results from laboratory experiments (Herman et al., 2017; Passerotti, 2022) and models (Herman, 2017; Montiel and Squire, 2017; Mokus and Montiel, 2021; Montiel and Mokus, 2021) indicate that a power-law FSD may be not the most appropriate way to describe the FSD due to wave-induced sea ice breakup.*

6. L77: I'm not sure I follow the argument that a low-resolution model output justifies the choice of a large model study area.

Thanks for your comments. For the WIPoFSD model and CPOM-FSD model, the grid cell area ranges from about 2468 km2 to 3480 km2 in the study region. For FSDv2-WAVE, the grid cell area is between 541 km2 and 3802 km2 in the study region. Within every grid cell, the atmosphere forcing, wave forcing, etc. are

homogeneous. The accuracy of FSDs in every grid cell relies heavily on the accuracy of external forcing. Applying a large region for models could better reveal the inhomogeneous effects of external forcing on the evolution of FSD and decrease the effects of external forcing bias in a few grid cells on FSDs. We added more details of this argument to the revised manuscript at L84-89 on Page 3 in the revised manuscript.

*The grid cell size of the three models evaluated in this study is 1° × 1°, ranging from 541 km$^2$ for the grid cells near the North Pole to 3802 km$^2$ for the grid cells on the far side of the North Pole in the study regions. Within every grid cell, the external forcing is homogeneous, and the model accuracy is dependent on the accuracy of external forcing. Compared to the satellite observation, much larger model study regions were selected, which could better represent the inhomogeneous effects of external forcing on the evolution of FSD and decrease the effects of external forcing bias in a few grid cells on FSDs.*

7. L90: What's the area of the uncropped images? 250 km^2? It would be interesting to know.

The size of the raw MEDEA image can be approximately more than 15 by 15 km, i.e., covering an area larger than 225 km$^2$ (Kwok, 2014), but often cloud-covered and missing/bad data. For the World-View image applied in this study, the size of raw images is more than 200 km$^2$, but we only purchase part of it (~40 km$^2$). The area of the uncropped images is added to the revised manuscript (L102, Page 4) and the size of cropped images is provided in the supplementary materials (Table S2).

*The original MEDEA images of area larger than 225 km$^2$ (Kwok, 2014) were cropped to remove cloud-covered areas and missing data. The size of the cropped images ranges between 30 km$^2$ and 250 km$^2$ (see Table S2 in the supporting information).*

8. L91-93: A reference for the WV images and more details about the sensor should be given.

Thanks. A reference for WV images and A more detailed information is added to the revised manuscript (L105-108 Page4).

*We also collected one WorldView-1 (WV1) and four WorldView-2 (WV2) images at the Chukchi Sea and Fram Strait sites (Fig. 1), which are collected in a 50 cm panchromatic band (WV1) at 0.52 m spatial resolution (Satellite Imaging Corporation, https://www.satimagingcorp.com/satellite-sensors/worldview-1/, last access: 16 February 2023) and 50 cm panchromatic band and 2 m 4-band Multispectral Bundle (WV2) at 0.47 m and 0.58 m spatial resolution respectively (Satellite Imaging Corporation, https://www.satimagingcorp.com/satellite-sensors/worldview-2/, last access: 16 February 2023).*

9. L98-102: how is "floe size" defined? There are multiple definitions out there.

Thanks for pointing out this. In this study, we define floe size as floe effective radius, the radius of a circle that has the same area as a floe. We added this information to the revised manuscript at L188.

*In this study, the floe effective radius $r = \sqrt{(a/\pi)}$ is used to define floe size, which is the radius of a circle that has the same area, $a$, as a floe.*

10. Section 3.2 is confusing. The intro paragraph discusses differences between the different models, before describing the models themselves in subsequent sub sub-sections. It would make more sense to describe the models first and then discuss their differences/similarities.

Thanks for your comments. The introduction paragraph gives a broad description and review of the recent FSD models including but not limited to models evaluated in this study. Section 3.2 introduces three specific models within them. In our opinion, it may be better to describe the differences between recent FSD models in the introduction section first and then introduce the details of the three specific models in the Data and Method section.

11. Section 3.2.1: More details about the WW3 configuration are needed. Is it a global or regional run? What ice attenuation parametrisation did you choose? Also, what do you mean "attenuation in the open ocean"?

Thanks for your comments. It is a global run. As in Roach et al. (2019), attenuation is the sum of two parts. "The first is an empirical fit to a multiple-floe scattering theory by Meylan et al. (2021) that accounts for floe size, thickness, and sea ice concentration. This theory finds that the scattering resembles the linear Boltzmann equation derived by Masson and Leblond (1989) with an error correction that accounts for multiple scattering. Attenuation from this theory becomes negligible when wavelengths are much greater than floe sizes; therefore, Roach et al. (2019) include an additional term based on observations to attenuate waves like the inverse of the wave period squared". More details will be given as a brief introduction to WW3 at L147-153 on Page5-6 in the revised manuscript.

*FSDv2-WAVE uses the slab ocean model (SOM) (Bitz et al., 2012) coupled with the global ocean surface wave model Wavewatch III v5.16 (WAVEWATCH III Development Group, 2016), and incorporates a new wave-dependent ice production scheme (Roach et al., 2019). Attenuation of wave energy in the MIZ is modelled using multiple wave scattering theory developed by Meylan et al. (2019), which accounts for the sea ice floe size, sea ice thickness and sea ice concentration. To address the deficiency of attenuation of wave when wavelengths are much greater than floe sizes in this theory, an additional attenuation scheme is applied as the reciprocal of wave period squared in FSDv2-WAVE.*

12. L149: "process" is the wrong word. Maybe "theory" or "model"?

Thanks. The wording is changed at L169 on Page 6 in the revised manuscript.

*WIPoFSD is a diagnostic power law FSD model (Bateson et al., 2020, 2022). The WIPoFSD model implements the wave-in-ice model (WIM), originally based on the ice-wave interaction theory described by…*

13. L153-154: I feel more details are needed about how the fixed power was determined. Was it the same data as those used later for comparisons with model outputs.

Thanks for your comments. It is not completely the same dataset. The exponent of - 2.56 was determined for the dataset considered in the Bateson et al. (2022) study rather than the more complete dataset considered in our study. To describe the differences between the dataset used by Bateson et al. (2022) and our study, hereafter, we refer to the observations used for calculating the power-law fit used in WIPoFSD (Bateson et al., 2022) as Obs1 and the observations used for mode-observation comparison in this study as Obs2. The difference between Obs1 and Obs2 are:

1) The different resolution: The resolution of the images used for deriving Obs1 was reduced from 2 m, while for Obs2, we keep its 1-m resolution.

2) The different number of sites: Obs1 is calculated from the data derived from three sites: the Chukchi Sea, East Siberian Sea and Fram Strait. Obs2 only includes 2 sites: the Chukchi Sea and Fram Strait.

3) The different number of cases in every site: For Obs1, there are 14 cases in the Chukchi Sea, 9 cases in the East Siberian Sea and 12 cases in the Fram Strait. For Obs2, there are 24 cases in the Chukchi Sea and 32 cases in the Fram Strait.

We now clearly explain the WIPoFSD model output and any choices about model parameters from the Bateson et al. (2022) study at L177-186 on Page 6-7 in the revised manuscript.

*There are three differences between the MEDEA imagery data used by Bateson et al. (2022) for calculating the power-law exponent (hereafter Obs1) and in this study for model evaluation (hereafter Obs2). 1) The image resolution of the satellite data used for deriving Obs1 was reduced to 2 m, while the original image resolution (1 m) was kept for Obs2. 2) Obs1 was derived from three sites: the Chukchi Sea, East Siberian Sea and Fram Strait, while Obs2 from 2 sites (the Chukchi Sea and Fram Strait). 3) The Obs1 datasets include 14 cases in the Chukchi Sea, 9 cases in the East Siberian Sea and 12 cases in the Fram Strait. The Obs2 datasets however include 24 cases in the Chukchi Sea and 32 cases in the Fram Strait. In addition to the exponent, the model also simulates FSD evolution through the floe size parameter $r_{var}$, varying between minimum floe radius $r_{min}$ and maximum floe radius $r_{max}$ in the distribution. $r_{var}$ evolves according to four FSD processes: lateral melt, wave-induced fracture, floe growth in winter and ice advection (Bateson et al., 2020, 2022). The full details of $r_{var}$, including a physical interpretation of this value, are provided in Bateson et al. (2020).*

Were all the floe size data from all the images collated into a single dataset and then a power law fit was performed or were power law fit done for each image and then averaged?

We applied power law fit for the floe size in each image first and then averaged. The range of floe sizes considered for the fits is 0 m-3000 m for Obs1.

Also is 3 significant figures appropriate?

Yes, 3 significant figures are appropriate. If we keep 5 significant figures and set $\alpha = 2.5574$, $[P(\alpha = 2.5574) - P(\alpha = 2.56)]/P(\alpha = 2.56) \approx -1\%$, which only shows a 1% smaller perimeter density.

What was the range of floe sizes considered for the fit(s)? What's the uncertainty on alpha? Was a goodness-of-fit test evaluation conducted?

Half cases pass the goodness-of-fit test.

A table of the power-law fit, uncertainty on alpha and p-value for the goodness-of-fit test of every case will be given in the revised supplement (Table S2).

14. Eq (2): I don't n(r) has been defined.

Thanks. The definition is added now at L189-192 on Page 7.

*The FSD is usually defined as the fractional-area distribution, $f(r)dr$, and number-density distribution, $n(r)dr$ (Rothrock and Thorndike, 1984; Toyota et al., 2006; Perovich and Jones, 2014; Horvat and Tziperman, 2015; Zhang et al., 2015; Hwang et al., 2017b; Bateson et al., 2020), corresponding to the area and the number of floes per unit ocean surface area with radius between $r$ and $r + dr$.*

15. L175-177: Rephrase these 2 sentences as they appear to contradict each other.

Thanks. We will move this sentence, "More details on the calculation of $P_i$ is provided in the supplementary materials Sect. S1", to the end of Section 3.3 (L218-219, Page 8).

*More details on the calculation of P is provided in the Supplementary Materials (S1).*

16. Eq (4): gamma needs to be defined.

Thanks for your comments. We added a clear definition of gamma at L203-206 on Page 7. When adding this definition, we realize that there was a mistake in the value of gamma for the calculation of Pi. As a result, we have revised all equations in both the supplement and main text. This modification led to changes in the value of Pi.

But there isn't any substantial difference in the comparison results when the value of gamma is modified.

*Here $\gamma$ is a floe shape parameter, the ratio of floe perimeter to twice the floe effective radius. For example, $\gamma = \pi$ is for circular floes. From the analysis of MEDEA-derived FSD results, the mean floe shape parameter $\gamma$ is 3.60 in the Chukchi Sea region and 3.69 in the Fram Strait region, which will be used for the calculation of P in this study. Due to the limitation of image resolution in capturing the shapes of small floes, the floes of $d < 5$ m are discarded for calculating $\gamma$.*

As r applied here is effective floe radius $r_{eff} = \sqrt{a/\pi}$, the radius of a circle that has the same area a as a floe. We assume $p = 2\gamma r_{eff}$. Here the gamma is defined as $\gamma = \frac{p}{2r_{eff}} = \frac{p}{2\sqrt{a/\pi}} = \frac{p\sqrt{\pi}}{2\sqrt{a}}$, where p and a are the perimeter and area of a floe respectively.

The gamma value we give in the original manuscript (the mean floe shape parameter $\gamma$ is 2.27 in the Chukchi Sea region and 2.23 in the Fram Strait region) is $\frac{4a}{d^2}$.

Now we have corrected both the equations and the Pi value shown in the text and Figures of results sections. Please see Lines 201, 204-206, 209, 217, 223-224, 229-234 and Figures 2, 3, 6 and 7 in the revised manuscript. For the supplementary materials, please see Line 54, Figure S1 and Table S1.

17. L197: How do you estimate central tendency and spread in the figures provided for Pi?

By applying equation $P_{obs} = \sum_{i=1}^{12} \frac{2\gamma A_{floe_i}}{\pi r_i A_{ice}}$, we calculate the $P_{obs}$ of every images. Then calculate the average of Pi and the standard deviation.

18. Fig 2e-k: there appears to be a plateauing in the observations for small floes. It could be due to a resolution issue as discussed in section 4.2, but it could also be a signature of the limitation of the power law. I know you are not trying to fit a power law here, but I believe this is an important point to make, especially when comparing to

Thanks for your comments. This is a very interesting point. We added discussion on the possibility of the limitation of the power law in the Discussion and Conclusion section in the revised manuscript (L372-377, Page 12-13).

*It is also worth noting that both 1-m and 0.5-m resolution observations show two distinct regimes of P for small floes ($r < O(10)$ m) and the larger floes (Figs 2e-2k, 4e and 4f). This situation is associated with several possible reasons: (i) Image resolution is not high enough to identify all small floes accurately; (ii) Lateral melt reduced the number of small floes (Hwang and Wang, 2022); (iii) Other statistic*

*models, e.g., log-normal distribution, is better to describe the FSDs rather than power laws (Montiel and Mokus, 2022).*

19. L212: use big O notation for order of magnitude.

Thanks. Revised at L 238 on Page 8 in the revised manuscript.

*The CPOM-FSD results also show a similar pattern, yet the model P values are in a much better agreement with the observations for large floes (r > O(10) m),*

20. L215-216: I believe an explanation was provided for the presence of the "uptick". It would be good to mention it here.

Thanks. The explanation is added (L241-244, Page 8).

*This type of 'uptick' in the prognostic models has been reported by Bateson et al. (2022) and Roach et al. (2018a), and is an artificial feature derived from the model setup of an upper floe size limit and an incomplete representation of fragmentation processes in the prognostic model for large floes.*

21. L254-255: That statement seems to be quite a stretch at this stage and sort of comes out of nowhere. Given the seasons considered, I would expect lateral melting to be a more important contributor to underestimated SIC.

Thanks. In the prognostic models, the floe welding rate is set to be proportional to the square of SIC in the two prognostic models. What we would like to explain here is that the underestimated SIC is a contributor to the underpredicted floe welding rate. We modified the sentences to make them clear at L281-282 on Page 10 in the revised manuscript.

*The underestimated SIC from the two prognostic models may be a contributor to the underpredicted floe welding rate during spring and early summer.*

22. L270: "floes welding are negligible during this season" seems to contradict the statement in the previous subsection (see comment 21).

Yes, the contribution of floe welding to the change of Pi in summer is negligible, i.e., floe welding does not happen in summer. But Bateson (2021) has assessed the effects of floes welding on the FSD in the CPOM-FSD model, suggesting that floe welding occurring in spring can influence the perimeter density in summer (Section 7.2.3 and Figure 7.9 in Bateson (2021)).

23. L334-339: again, I don't find the argument about underpredicting welding very convincing. What about potential overestimation in the data? See Roach et al (2018, The Cryosphere)

Thanks. Roach et al (2018) discuss the reason for the overestimation of sea ice concentration (SIC) as lateral melt. Similar to the response to comment 22, in this study, we discussed the underestimation of SIC will lead to the underestimated floe welding and thus the overestimation of perimeter density of small floes in summer. Our explanation may cause a misunderstanding.

Thanks for your comments. We modified the sentences to make these causal relationships clear at L379-380 on Page 13 in the revised manuscript.

*The strength of floe welding is strongly contributed from the SIC in the prognostic models evaluated in this study (Roach et al., 2018a, b; Bateson, 2021; Bateson et al., 2022).*

24. L243: Again, how was the fit performed? Across all floe sizes or a limited range? What about goodness-of-fit?

Thanks. I guess that you may mean L343.

This exponent is calculated across all floe sizes. By setting the floe size range of 0-3000m, 50% data pass the goodness-of-fit test. We added this information in Table S2 in the supplementary materials, including the power-law exponent, power-law range and p-value for the goodness-of-fit test of every case.

25. L245-247: What about the possibility that the power law is not appropriate?

Thanks. I guess that you may mean L345-347.

As you mentioned in comment 18, almost flat FSD in the observations for small floes could result from the limitation of image resolution (section 4.2) or a signature of the limitation of the power law. Besides, as in Hwang and Wang (2022), this behaviour may be consistent with the fact that small floes are more vulnerable to lateral melt relative to large floes, which causes the deviation of the small-floe distribution prior to the tail. A discussion of all these possibilities is added in this section in the revised manuscript at L372-377

*It is also worth noting that both 1-m and 0.5-m resolution observations show two distinct regimes of P for small floes (r < O(10) m) and the larger floes (Figs 2e-2k, 4e and 4f). This situation is associated with several possible reasons: (i) Image resolution is not high enough to identify all small floes accurately; (ii) Lateral melt reduced the number of small floes (Hwang and Wang, 2022); (iii) Other statistic models, e.g., log-normal distribution, is better to describe the FSDs rather than power laws (Montiel and Mokus, 2022).*

I don't see why a power fitted through historical data should predict a power law in the current dataset, even at the same location.

Through historical data, we would like to indicate that the spatial and seasonal variation of the power-law exponent does not only exist in our datasets but also in

previous studies. This could emphasize the view that a constant exponent is not appropriate.

**Typographical errors**

26. L58: delete "the".

    Thanks. Corrected.

27. "welding" is misspelled a couple of times.

    Thanks. Corrected.

**Citation in the response to Referee #2**:

Arntsen, A. E., Song, A. J., Perovich, D. K., and Richter-Menge, J. A.: Observations of the summer breakup of an Arctic sea ice cover, Geophys. Res. Lett., 42, 8057–8063, https://doi.org/10.1002/2015GL065224, 2015.

Bateson, A. W.: Fragmentation and melting of the seasonal sea ice cover, Ph.D. thesis, Department of Meteorology, University of Reading, United Kingdom, 293 pp., 2021.

Bateson, A. W., Feltham, D. L., Schröder, D., Wang, Y., Hwang, B., Ridley, J. K., and Aksenov, Y.: Sea ice floe size: its impact on pan-Arctic and local ice mass and required model complexity, 16, 2565–2593, https://doi.org/10.5194/tc-16-2565-2022, 2022.

Horvat, C., Blanchard-Wrigglesworth, E., and Petty, A.: Observing Waves in Sea Ice With ICESat-2, Geophys. Res. Lett., 47, https://doi.org/10.1029/2020GL087629, 2020.

Hwang, B. and Wang, Y.: Multi-scale satellite observations of Arctic sea ice: new insight into the life cycle of the floe size distribution, Philos. Trans. R. Soc. A Math. Phys. Eng. Sci., 380, https://doi.org/10.1098/rsta.2021.0259, 2022.

Kwok, R.: Declassified high-resolution visible imagery for Arctic sea ice investigations: An overview, Remote Sens. Environ., 142, 44–56, https://doi.org/10.1016/j.rse.2013.11.015, 2014.

Masson, D., & Leblond, P. H.: Spectral evolution of wind-generated surface gravity waves in a dispersed ice field. Journal of Fluid Mechanics, 202, 43–81, https://doi.org/10.1017/S0022112089001096, 1989.

Meylan, M. H., Horvat, C., Bitz, C. M., and Bennetts, L. G.: A floe size dependent scattering model in two- and three-dimensions for wave attenuation by ice floes, Ocean Model., 161, 101779, https://doi.org/10.1016/j.ocemod.2021.101779, 2021.

Perovich, D. K.: Aerial observations of the evolution of ice surface conditions during summer, J. Geophys. Res., 107, 8048, https://doi.org/10.1029/2000JC000449, 2002.

Perovich, D. K. and Jones, K. F.: The seasonal evolution of sea ice floe size distribution, J. Geophys. Res. Ocean., 119, 8767–8777, https://doi.org/10.1002/2014JC010136, 2014.

Roach, L. A., Horvat, C., Dean, S. M., and Bitz, C. M.: An Emergent Sea Ice Floe Size Distribution in a Global Coupled Ocean-Sea Ice Model, J. Geophys. Res. Ocean., 123, 4322–4337, https://doi.org/10.1029/2017JC013692, 2018.

Roach, L. A., Bitz, C. M., Horvat, C., and Dean, S. M.: Advances in Modeling Interactions Between Sea Ice and Ocean Surface Waves, J. Adv. Model. Earth Syst., 11, 4167–4181, https://doi.org/10.1029/2019MS001836, 2019.

---

## Author Response (AR2)

**Response to Referee #1**

Dear Referee,

We would like to thank you for reviewing our manuscript and giving valuable comments and suggestions. Please find our responses to your comments in blue text and revised manuscript in red text below.

The major change in this revision is:

We give a clarification of the two metrics for model evaluation that we have already been used in our analysis. The definition of $p(r)dr$ is the perimeter of floes per unit ice area with radius between $r$ and $r + dr$

$$P = \int_{r_{min}}^{r_{max}} p(r)dr$$

$P$ is the perimeter of floes per unit area with $r$ varying between minimum floe radius $r_{min}$ and maximum floe radius $r_{max}$ in the distribution.

The metrics evaluated in this study is not only $P$, a value of the integral of perimeter density per unit sea ice area over all floe sizes (e.g., the value given at Line 227-236 on Page 8) but also $p(r)$. $P$ captures the most useful information about FSD shape from the perspective of modelling into a singular value, making it easier to present comparisons between observations and models. Whilst information about FSD shape is lost when calculating $P$, we also apply $p(r)$ to evaluate the model performance (e.g., Figures 2e-2k, 4e-4f). In our manuscript, $p(r)$ is shown as the perimeter density distribution for floe size categories (perimeter of floes in floe size category $i$ per unit bin width per unit sea ice area). The text and figures have been revised when we show the results about $p(r)$ in the revised manuscript. It seems like the confusion between P and p(r) has resulted in a misunderstanding, so hopefully addressing this point should resolve most of your concerns around this point.

Best regards,

Yanan Wang and co-authors

This review is from Referee #1 of the original manuscript. The authors have made changes in the right direction, but there are still issues to address.

Each model produces a sea-ice floe size distribution (FSD) from which the sea-ice floe perimeter density (P) can be calculated (assuming a shape parameter, gamma). Each satellite image also yields a FSD and P. The authors compare P from models with P from satellite images, which is a perfectly proper and legitimate way to assess the skill of the models. But note that the relationship between P and FSD is a one-way street: given the FSD and shape parameter gamma, one can calculate P; but given P, one can say nothing about the FSD. The results of the analysis show that in general there is very poor

agreement between modelled P and observed P, and that's fine -- it simply means that the models are not able to reproduce the observed P. But even if the agreement were good, it would say nothing about the modelled FSDs. That point should be acknowledged or conveyed somewhere.

Based on our clarification of $p(r)$ and $P$ given above, $p(r)$ and $n(r)$ can be related by $p(r) = \frac{2\gamma rn(r)}{c_{ice}}$. Given $n(r)$, gamma and SIC, we can also calculate $p(r)$ and vice versa. This definition has been given in the revised manuscript. Please see Lines 190–194 on Page 7 in the revised manuscript.

*For the evaluation of FSD models, we consider the perimeter density distribution, $p(r)dr$ (units: km km$^{-2}$), which is defined here as the perimeter of floes per unit ice area with radius between $r$ and $r + dr$,*

$$P = \int_{r_{min}}^{r_{max}} p(r)dr,$$
*(1)*

*P is the perimeter per unit area of floes between $r_{min}$ and $r_{max}$ in radius.*

The perimeter density P is not a proxy for the FSD. A proxy is "a measurement of one physical quantity that is used as an indicator of the value of another" or equivalently "a measured variable used to infer the value of a variable of interest" (to quote two dictionary definitions). Now consider these examples:
1. All floes are circular with radius r0. The FSD is the Dirac delta function, f(r) = delta(r - r0), and P = 2/r0. Suppose r0 = 20 meters = 0.02 km. Then P = 100/km.
2. All floes are circular with uniform distribution on [0,L]. The FSD is f(r) = 1/L and P = 3/L. Suppose L = 30 meters = 0.03 km. Then P = 100/km.
3. All floes are circular with exponential distribution. The FSD is f(r) = (1/lambda)*exp(-r/lambda) and P = 1/lambda. Suppose lambda = 10 meters = 0.01 km. Then P = 100/km.
In all three examples, P = 100/km. If one is simply given that P = 100/km, it is not possible to "go backwards" and find the FSD. The FSD could be anything. Any distribution with a free parameter can be made to have P = 100/km by proper choice of the parameter. P is not a proxy for FSD.

We have clarified the terms P and p(r) in the updated, which we believe should address the above concerns. Based on the relationship between $p(r)$ and $n(r)$, $p(r) = \frac{2\gamma rn(r)}{c_{ice}}$, if the function of n(r) is different, p(r) is different. Yes, for these examples, the integral of p(r) over all floe sizes, i.e., P, are all 100/km, but the $p(r)$ is different.

Besides, we have also replaced the word 'proxy'. Please see Lines 195–198 on Page 7 in the revised manuscript.

*P is often used as a way to capture useful information about the FSD using a singular value and the concept of an overall perimeter density to characterize the FSD has been*

The authors misinterpret the work of Perovich and Jones (2014). At lines 237-238 of the marked-up revised manuscript, the authors write, "P is a useful proxy for FSD and widely used in previous observational studies (e.g. Perovich, 2002; Perovich and Jones, 2014; Arntsen et al., 2015)." In Perovich and Jones (2014) they ASSUME that the FSD follows a power law, and then use the observed P to determine the power-law exponent. In other words, P is not used as a proxy for the FSD -- the FSD is already fixed as a power law. P is used for finding the exponent.

Thanks for pointing out this oversight. Yes, P is not used as a proxy for the FSD in Perovich and Jones (2014). We have deleted the citation of this paper in this sentence (Line 195–198 on Page 7).

*P is often used as a way to capture useful information about the FSD using a singular value and the concept of an overall perimeter density to characterize the FSD has been used in several previous observational studies (e.g., Perovich, 2002; Arntsen et al., 2015),*

This is what the authors wrote in their response to my comment about perimeter density (PD) and FSD in the original manuscript:

"1) The same PD corresponding to different number FSDs or the same FSD corresponding to different PDs does not mean there is no connection between PD and FSD. In previous studies, Rothrock and Thorndike (1984) first defined the FSD as both number density n(r) and area fraction f(r). These two definitions are related by f(r)=gamma * r^2 * n(r). Now, if we consider a set of circular floes and a set of elliptical floes with semi-major axis "a" and semi-minor axis "b" with the same number FSD n(r) in a given region. But assume that pi*a*b NE pi*r^2. In this situation, although these two groups of floes have the same n(r), their areal FSD f(r) is different. Similarly, if we consider 5 circular floes with the same radius of 10 m and 100 elliptical floes with semi-major axis "a=1 m" and semi-minor axis "b = 5 m" in a given region. Then we can get the same areal FSD, different number FSDs. Even the traditional FSD concepts n(r) and f(r) still shows the same situation that there is not always a unique 1:1 relationship between them. Similarly, it is not abundant evidence to prove that PD is not a metric related to FSD by showing the same PD corresponding to different number FSDs or the same FSD corresponding to different PDs."

Let's analyze some of the statements in the paragraph above.
1. The equation "f(r) = gamma * r^2 * n(r)" is presented but then later "Even the traditional FSD concepts n(r) and f(r) still shows the same situation that there is not

always a unique 1:1 relationship between them." But the equation given by the authors shows that once the constant gamma is chosen, there is indeed a 1:1 relationship between n(r) and f(r).

*The reason that we mention "there is not always a unique 1:1 relationship between them" is because of that most examples in your previous comments are related to the variable shape of floe, i.e., γ is not the same for the two groups of circle and elliptical floes in these examples. However, we also responded that "as pointing out by previous study, the ratios between floe size properties is generally constant and almost independent of floe size (e.g., Rothrock and Thorndike, 1984; Toyota et al., 2006). Besides, it is common in FSD studies to make assumptions about floe shape, e.g., FSD models assume a fixed shape parameter γ. This should not be a cause for concern. "*

*Similarly, we have clarified the terms P and p(r) in the updated manuscript, which we believe should address the above concerns. Regarding the definitions P(r) and p(r) given in the revised manuscript, $p(r) = \frac{2\gamma r n(r)}{c_{ice}}$. Once the constant gamma and SIC are given, there is indeed a 1:1 relationship between n(r) and p(r) as well.*

2. "if we consider 5 circular floes with the same radius of 10 m and 100 elliptical floes with semi-major axis a=1 m and semi-minor axis b = 5 m in a given region. Then we can get the same areal FSD, different number FSDs." In the first case, the 5 circular floes of radius 10 m have a total area of 500*pi. In the second case, the 100 elliptical floes also have a total area of 500*pi. Yes, the total area is the same in both cases. And yes, the number of floes in the first case (5) is different than the number of floes in the second case (100). But the authors are mistaken or confused in their use of "FSD" in these examples. Since all the circular floes have radius 10 m, the FSD is n(r) = delta(r-10). Since all the elliptical floes have a=1 m, the FSD is n(a) = delta(a-1). The FSDs are all delta functions. The authors don't seem to understand the difference between "area" and "areal FSD" nor between "number" and "number FSD". This is troubling.

*As mentioned above, the metrics that we compared in this manuscript is not only P (e.g., the value given at Line 227-237 on Page 8) but also p(r) (e.g., Figures 2e-2k, 4e-4f). p(r) is perimeter density distribution rather than perimeter, which is also a function of floe size r.*

Image Resolution.
The models overestimate P for floes with r < 15 m (Fig. 2). The observed P is based on images with pixel size 1 meter. The authors analyze 5 images with pixel size 0.5 meters, resulting in larger P for small floes (Fig. 4). The authors conclude that inadequate image resolution may be one of the causes of the discrepancy between modelled and observed P for small floes.
Line 23: "the image resolution is not sufficient to detect small floes"

Lines 586-587: "The causes of such differences include ... the limitations within the observations such as the image resolution"
Lines 632-633: "It requires much higher resolution images (e.g., aerial photographs) and further research in the future to properly investigate the effects of the image resolution"
Lines 662-663: "the ability to resolve the shape of the FSD for small floes remains constrained by the limited resolution of satellite images"

During late spring and summer, there is a large number of floes with radii smaller than 15 m. During processing the images, even in the 0.5 m-resolution images, we noticed there are several groups of crowded small floes that remain unresolved. Please see the following example (100 x100 m) from the WV image ($\delta = 0.5$ m) on 5 June 2013 at 84.9°N, 0.1°E (Fram Strait). We do not know how much additional change we would see in P and p(r) if we had access to imagery at even higher resolutions. The limited size of the change from 1 m to 0.5 m compared to the size of the difference between models and observations suggests that image resolution cannot fully explain this difference. The image resolution could be one of the contributors but not the main contributor.

[Figure]

A few comments:
1. At lines 600-605, the authors admit that the increase in P in going from 1-m images to 0.5-m images "is still far too small to explain the difference between the observations and model outputs." Nevertheless, the next sentence says: "This suggests that the image resolution could be one of the contributors to the overestimation of modelled P for small floes, but it is still inconclusive whether the limited image resolution is the main contributor or other factors such as model parameterisations contribute to the difference." Really? Can't we conclude that image resolution is NOT the main contributor, given that halving the pixel size from 1 meter to 0.5 meters gives a change in P that is "far too small to explain the difference between observations and model outputs"?

Please see the example given in our response above. We do not know how much additional change we would see in P and p(r) if we had access to imagery at higher

resolutions. What we would like to explain is that image resolution could be one of the contributors but not the main contributor. These sentences have been revised. Please see Lines 378–382 on Page 13 in the revised manuscript.

*However, this difference is still too small to explain the difference between the observations and model outputs, varying between 20.42 km km$^{-2}$ and 218.95 km km$^{-2}$ in Figs 2e–2k (See Table S1). We do not know how much additional change we would see in P and $p(r)$ if we had access to imagery at even higher resolutions. This suggests that based on the recent satellite imagery, the image resolution could be one of the contributors but is not the main contributor to the overestimation of modelled p of small floes.*

2. Judging by Figure 2, the models simulate floe sizes down to about r=3 meters. The authors suggest that the image resolution of 1 meter (or 0.5 meters) is not sufficient (see quotes above). Apparently centimeter-scale imagery is needed (e.g. from aerial photographs). Wouldn't that result in a mismatch in the scale of comparison, if images resolved floe sizes of r=0.1 meters but models only resolved r=3 meters?

As outlined in Roach et al. (2018a) and Bateson et al. (2022), 12 Gaussian-spaced floe size categories are applied in FSDv2-WAVE and CPOM-FSD simulations. In the smallest floe size category, floes with radii between 0.07-5.31m. In Figure 2, 2.69 metres is the midpoint of this bin rather than the smallest floe size simulated in models. The minimum floe size for the prognostic models is technically 0.13 m. Additionally, the image resolution of 1 m does not equal the smallest size of derived floe of 1 m. The distance between floes can also influence the separation of floes from images, especially when during late spring and summer. Please see the example of unresolved floes in 0.5m-resolution image given in our response above.

3. In order to explain or account for the difference in modelled P vs. observed P for small floes, the authors look for a potential shortcoming in the data (its resolution) rather than a potential shortcoming in the models. This seems like the wrong approach. In my opinion, 1-meter (or 0.5-meter) image resolution is good enough! I'm not convinced that we need centimeter-scale imagery to properly characterize the FSD and P.

Our results in Section 4.2 indicate that the minimum floe radius that could be retrieved accurately from the 1-m images was 15 m. However, during late spring and summer, there is a large number of floes with radii smaller than 15 m. Please see the example given in our response above. During processing the images, even the 0.5 m-resolution images, we noticed there are several groups of crowded small floes that cannot be separated. We do not know how much additional change we would see in P if we had access to imagery at even higher resolutions. But based on the comparison of 1-m to 0.5-m resolution imagery, we do not believe that this effect is the main contribution to the differences between models and observations. This discussion in the paper is not to

assume that the problem is with observations, but to rule this out so that we can reach some conclusions on model performance

Lines 448-450. "floe welding rate is set to be proportional to the square of SIC in the two prognostic models ... In this section, we present the model-observation comparison results for SIC to validate floe welding for the prognostic models."
The model-observation comparison of SIC does not validate the floe welding parameterization. There are no observations of floe welding presented in this paper. The comparison of SIC validates SIC, not floe welding.

Yes, we agree that floe welding is not validated directly in this study. Roach et al. (2018b) provide observational evidence that floe welding correlates well with sea ice concentration, which is used to give this floe welding parameterization, e.g., the floe welding rate is set to be proportional to the square of SIC in the two prognostic models. This result just provides some possibilities of underestimated SIC leading to the decrease of P through floe welding in the two prognostic models. We have cited the observational evidence of the correlation between floe welding and SIC in the revised manuscript. Please see Lines 276–280 on Pages 9–10 in the revised manuscript.

*The FSD models considered in this study include parameterisations with dependencies on SIC. For example, the floe welding rate is set to be proportional to the square of SIC in the two prognostic models, FSDv2-WAVE and CPOM-FSD based on observations that the welding rate correlates with SIC (Roach et al., 2018b). It is therefore useful to evaluate how well the models simulate the observed SIC and consider the extent to which errors in the simulated SIC could explain the differences between models and observations in simulating floe perimeter density.*

Lines 479-481. "FSDv2-WAVE slightly underestimates SIC by 2-4% compared to the observations in the MIZ (SIC<80%). CPOM-FSD strongly underestimate the SIC by 13%-15% in the MIZ compared to the observations. This difference can be attributed to different atmospheric forcing that is used in the models (Schroder et al., 2019)." I don't see any evidence that the difference in SIC can be attributed to the different atmospheric forcing (NCEP-2 vs JRA55b) used in the models. This sentence simply makes a declaration without any further justification. I looked up the Schroder reference and it uses NCEP-2 but not JRA55b, so I don't see how it supports the claim being made here.

Schroder et al. (2019) didn't analyse the effects of specific atmospheric conditions on Arctic SIC. Schroder et al. (2019) suggested there is a strong influence of the spring and summer atmospheric conditions on the simulated summer sea ice extent in CICE, so the different atmospheric forcing used (NCEP-2 and JRA55b in our study) can be expected to be a significant factor in explaining the difference in SIC between FSDv2-WAVE and CPOM-FSD. This sentence has been revised at Lines 287-288 on Page 10.

*This difference in SIC between FSDv2-WAVE and CPOM-FSD can be attributed to different atmospheric forcing that is used in the models (Schröder et al., 2019).*

Lines 483-484. "The underestimated SIC from the two prognostic models may be a contributor to the underpredicted floe welding rate during spring and early summer." Since the floe welding rate is proportional to the square of SIC in the two prognostic models (line 448), underestimation of SIC translates directly into underpredicted floe welding rate in the models. It's not "may be a contributor to" but rather "is the cause of".

Thanks, we have revised this sentence. Please see Lines 289–290 on Page 10 in the revised manuscript.

*The underestimated SIC from the two prognostic models will result in a too small floe welding rate during spring and early summer.*

Incidentally, we don't know if the floe welding rate is, in reality, proportional to SIC-squared. That has not been validated in this paper.

As in our response above, Roach et al. (2018b) provide observational evidence that floe welding correlates well with sea ice concentration, which is used to give this floe welding parameterization. We have also included this information in our revised manuscript. Please see Lines 276–280 on Pages 9–10 in the revised manuscript.

*The FSD models considered in this study include parameterisations with dependencies on sea ice concentration. For example, the floe welding rate is set to be proportional to the square of SIC in the two prognostic models, FSDv2-WAVE and CPOM-FSD based on observations that the welding rate correlates with SIC (Roach et al., 2018b). It is therefore useful to evaluate how well the models simulate the observed SIC and consider the extent to which errors in the simulated SIC could explain the differences between models and observations in simulating floe perimeter density.*

Lines 484-485. "A negative bias in spring SIC shown in the prognostic models may partially explain the overestimation of P especially for small floes (Fig. 2)." First, the overestimation of P is ONLY for small floes (Fig. 2) so it's misleading to say "especially for small floes."

Based on our updated definition of P and p(r), we have revised these sentences as follows. Please see Lines 290–292 on Page 10 in the revised manuscript.

*A negative bias in spring SIC shown in the prognostic models may partially explain the overestimation of $P$, and in particular the overestimation of $p(r)$ for small floes and the underestimation of $p(r)$ for large floes (Fig. 2).*

Second, a negative bias in SIC could be due to too few small floes or too few large floes. Simply knowing that SIC is too low does not automatically imply an overestimation or underestimation of P.

Regarding your second comment, again, Roach et al. (2018b) provide observational evidence that floe welding correlates well with sea ice concentration, which is used to give this floe welding parameterization. Bateson (2021) has assessed the effects of floe welding on the p(r) in the CPOM-FSD model, suggesting that floe welding occurring in spring can influence the p(r) in summer. A low ice concentration reduces the floe welding during spring and consequently results in an initial over-fragmented state in early summer. Therefore, a negative bias in spring SIC shown in the prognostic models can partially explain the large value of p(r) for small floes and the small value of p(r) for larger floes in the two prognostic models (Fig. 2). We have cited the observational evidence of the correlation between floe welding and SIC (Lines 276–280 on Pages 9–10) and the effects of floe welding on p(r) for small and large floes in the revised manuscript (Lines 393–396 on Page 13).

*The FSD models considered in this study include parameterisations with dependencies on sea ice concentration. For example, the floe welding rate is set to be proportional to the square of SIC in the two prognostic models, FSDv2-WAVE and CPOM-FSD based on observations that the welding rate correlates with SIC (Roach et al., 2018b). It is therefore useful to evaluate how well the models simulate the observed SIC and consider the extent to which errors in the simulated SIC could explain the differences between models and observations in simulating floe perimeter density.*

*A low ice concentration reduces the floe welding during spring and consequently results in an initial over-fragmented state in early summer. Therefore, a negative bias in spring SIC shown in the prognostic models can partially explain the overestimation of $p(r)$ for small floes and the underestimation of $p(r)$ for larger floes in the two prognostic models (Fig. 2).*

Lines 652-654. "we examined the P in the northern regions where wave-induced breakup is negligible. In these regions, most modelled P match our observations better." There are no observations in the northern CS region.

The reasons of this comparison do not only depend on whether there are observations in the northern CS region or not. As we mentioned in the previous response, "*In the northern CS, the change in P arising from all FSD evolution processes is almost zero in the two prognostic models during our research period. For checking the P in these regions, we recognize the P value in the northern region is the value at the end of early spring or the beginning of our research period. This comparison could help us determine whether the differences between observations and models arise from FSD evolution processes in summer (lateral melt and wave fracture) or in spring.* "

Figure 6. The caption refers to changes in FSD but the figure shows changes in P.

Thanks. We have corrected the caption of Figure 6.

*Figure 6: Monthly changes of **P** simulated by the two prognostic models over the period May to July during 2000–2014. (a) Change of FSD arising from lateral melt for FSDv2-WAVE in the Chukchi Sea. (b) is same as (a) but for wave induced **P** change. (c) and (d) are same as (a)and (b) but in the Fram Strait. (e)–(h) is same as (a)–(d) but for CPOM-FSD. The blue and red box in (a) and (c) show the northern and southern region of the two study regions. Black dots indicate the location of observations in the study regions.*

Table 2. The caption refers to three models but the table lists only two models.

Thanks. We have corrected the caption of Table 2 in the revised manuscript.

*Table 2. Statistical summary for the two prognostic FSD models against the NSIDC SIC and ASI SIC. NSIDC and ASI SIC data used for the comparison are between April and August for the analysis period of 2000–2014.*

**Response to Referee #2**

Dear Dr Montiel,

We would like to thank you for reviewing our manuscript and giving valuable feedbacks. Please find our responses to your comments in blue text and revised manuscript in red text below.

Best regards,

Yanan Wang and co-authors

I have now gone through the revised manuscript and the authors' responses and am happy to recommend that the manuscript is accepted.

1. I only have one comment, which the authors and/or editor may want to take into consideration. From the viewpoint of the reader, I still think that having a discussion about model differences at the start of section 3.2 before the models are fully described does not flow well.

We have revised Section 3.2 in the revised manuscript (Lines 130–184 on Pages 5-6).

*3.2 Sea ice models with floe size distribution*

*In this study, three FSD models are evaluated. An overview of the configuration of these three FSD models is given in Table 1. In Sects. 2.4.2–2.4.4, we will briefly introduce the major differences between the three models in simulating FSD related processes.*

*3.2.1 FSDv2-WAVE model*

*FSDv2-WAVE model is based on a sub-grid scale floe size and thickness distribution (FSTD) model by Horvat and Tziperman (2015, 2017).*

*……*

*3.2.3 WIPoFSD model*

*WIPoFSD is a diagnostic power law FSD model incorporated into the CPOM-CICE (Bateson et al., 2020, 2022) and has the same horizontal grid and run period as CPOM-FSD.*

2. In addition, I suggest that the authors add the reference Mokus and Montiel (2022, The Cryosphere) in line 376, as that paper really complements the Montiel and Mokus (2022, Phil Trans A) paper and these should be taken in tandem.

We have added in the revised manuscript (Lines 385–387 on Page 13).

*Other statistical models, e.g., log-normal distribution, is better to describe the FSDs rather than power laws (Montiel and Mokus, 2022; Mokus and Montiel, 2022).*

3. Finally, the year of the reference to "Meylan et al (2019)" in line 150 should be 2021.

We have revised in the revised manuscript (Lines 142-143 on Page 5).

*Attenuation of wave energy in the MIZ is modelled using multiple wave scattering theory developed by Meylan et al. (2021).*

---

## Author Response (AR3)

Dear Referee,

We would like to thank you for reviewing our manuscript and giving valuable feedbacks. Please find our responses to your comments in blue text and revised manuscript in red text below.

Best regards,

Yanan Wang and co-authors

The authors have added the notation p(r) to refer to the perimeter density (per unit ice area) for floes of radius r. The notation P refers to the integral of p(r) over all values of r. This distinction is certainly helpful for understanding the analysis. The most remarkable result from this work, in my opinion, is the complete lack of agreement between data and models, and between the models themselves (Figures 2 and 3).

Below are my comments on the latest revision of the manuscript. Page and line numbers refer to the "clean" file
tc-2022-130-manuscript-version4.pdf

Page 3, line 68. "we derived a new FSD dataset from 1-m resolution MEDEA imagery" Note that Denton and Timmermans (2022) did the same thing. Please acknowledge their work.
Denton and Timmermans (2022), Characterizing the sea-ice floe size distribution in the Canada Basin from high-resolution optical satellite imagery, The Cryosphere, 16, 1563–1578,
https://doi.org/10.5194/tc-16-1563-2022

We have added an introduction of recent observations and cited this paper. Please see Lines 57–61 on Page 2 in the updated manuscript.

*To enhance the understanding of the FSD evolution in various seasons and regions in the Arctic, a wide range of observations from aerial vehicles (e.g., Perovich and Jones, 2014; Toyota et al., 2016), optical satellites, e.g., Landsat (e.g., Rothrock and Thorndike, 1984; Gherardi and Lagomarsino, 2015; Wang et al., 2016), MEDEA (e.g., Denton and Timmermans, 2021; Hwang and Wang, 2022), MODIS (Toyota et al., 2016; Stern et al., 2018a), and Synthetic Aperture Radar (SAR), e.g., TerraSAR-X (Hwang et al., 2017b; Stern et al., 2018a), have been used to derive floe sizes.*

Page 7, lines 190-191. When introducing p(r) just before equation (1), it would be helpful to state that p(r) is proportional to r*n(r).

Thanks. We have revised in the revised manuscript (Lines 195-200 on Page 7).

*For the evaluation of FSD models, we consider the perimeter density distribution, $p(r)dr$ (units: km km$^{-2}$), which is proportional to $r \cdot n(r)dr$. Perimeter density distribution defined here as the perimeter of floes per unit ice area with radius between $r$ and $r + dr$. The integral of $p(r)$ between $r_{min}$ and $r_{max}$ in radius is defined as total perimeter density,*

$$P = \int_{r_{min}}^{r_{max}} p(r)dr,$$
*(1)*

*which is the perimeter per unit ice area of floes between $r_{min}$ and $r_{max}$.*

Page 7, lines 195-196. "P is often used as a way to capture useful information about the FSD using a singular value"
As stated in my first and second reviews, P does not convey anything about the FSD. Any number of different FSDs can give rise to the same P. It is not possible to go backwards from P and say anything about the FSD from which it was calculated. P is a legitimate metric to use for comparing models and observations, but it does not "capture useful information about the FSD".

We have revised these sentences as follows. Please see Lines 201–203 on Page 7 in the revised manuscript.

*The concept of a perimeter density to characterize the overall floe size has been used in several previous observational studies (e.g., Perovich, 2002; Arntsen et al., 2015),*

Page 7, lines 199-201. "In this study, P (units: km km-2) is used as it captures the most useful information about FSD shape... Whilst information about FSD shape is lost when calculating P..."
Again, P does not capture anything about FSD shape. Furthermore, it seems odd to claim that P captures the most useful information about FSD shape and then admit in the next sentence that information about FSD shape is lost when calculating P.

We have revised these sentences as follows. Please see Lines 204-205 on Page 7 in the revised manuscript.

*In this study, P (units: km km$^{-2}$) is used to present comparisons between observations and models.*

Page 8, line 237. "regional difference (Figs. 2c and 2d)."
I believe this should be Figs. 2b and 2d (not 2c).

We have revised in the revised manuscript (Line 241 on Page 8).

*WIPoFSD (Chukchi Sea: 120.93 ± 1.66 km km⁻², Fram Strait: 138.99 ± 12.98 km km⁻²)* $WIPoFSD$ *(Chukchi Sea: 120.93 ± 1.66 km km⁻², Fram Strait: 138.99 ± 12.98 km km⁻²)*
*and CPOM-FSD (Chukchi Sea: 59.55 ± 19.13 km km⁻², Fram Strait: 61.19 ± 29.95 km km⁻²) also show a general overestimation of P to the observations and the opposite regional difference (Figs. 2b and 2d).*

Page 9, line 266. "In Sect. 3.1"
I believe this should be 4.1

We have revised in the revised manuscript (Lines 270 on Page 9).

*In Sect. 4.1, the three models all show larger value of $p(r)$ for small floes (r < 10–30 m) than the observations (Figs. 2e–2k).*

Page 9, line 266. "than the observations (Figs. 3e-k)."
I believe this should be Figs. 2e-k (not 3).

We have revised in the revised manuscript (Lines 270 on Page 9).

*In Sect. 4.1, the three models all show larger value of $p(r)$ for small floes (r < 10–30 m) than the observations (Figs. 2e–2k).*

Page 9, line 267. "This large value of modelled p(r) for small floes may be attributed to the limited image resolution in retrieving small floes." This makes no sense, because the modelled p(r) has nothing to do with the image resolution. The models and the observations are independent. The sentence should be phrased something like this: "The limited image resolution may hinder the retrieval of small floes. To test this..."

We have revised in the revised manuscript (Lines 271 on Page 9).

*The limited image resolution may hinder the retrieval of small floes. To test this, we investigate $p(r)$ derived from MEDEA…*

Pages 9-10, lines 276-278. "the floe welding rate is set to be proportional to the square of SIC in the two prognostic models, FSDv2-WAVE and CPOM-FSD based on observations that the welding rate correlates with SIC (Roach et al., 2018b)."
Correlation implies a linear relationship. If "the welding rate correlates with SIC" then it's proportional to SIC, not SIC^2. If Roach et al (2018b) found that the welding rate is proportional to SIC^2 then just write "the floe welding rate is set to be proportional to the square of SIC in the two prognostic models, FSDv2-WAVE and CPOM-FSD, based on the work of Roach et al (2018b)."

We have revised in the revised manuscript (Lines 279-281 on Page 9).

*For example, the floe welding rate is set to be proportional to the square of SIC in the two prognostic models, FSDv2-WAVE and CPOM-FSD based on the work of Roach et al. (2018b).*

Page 11, lines 333-334. "The P values from the northern regions are considerably smaller than the values from the southern regions for the two models"
I'm looking at Table 3. I see that the above statement is true for FSDv2-WAVE in the CS region, and for CPOM-FSD in the FS region, but it is not true for FSDv2-WAVE in the FS region nor for CPOM-FSD in the CS region.

We have revised in the revised manuscript (Lines 336-337 on Page 11).

*The P values from the northern regions are smaller than the values from the southern regions for the two models (Fig. 7).*

Page 11, line 336. "comparable to the observation values (Fig. 7a and Table 3)." Table 3 says nothing about observation values. Just write "(Fig. 7a)."

We have revised in the revised manuscript (Line 339 on Page 11).

*In the Chukchi Sea region, the SIC values from the northern region are clustered between 90% and 100%, and the P values for both models are comparable to the observation values (Fig. 7a).*

Page 27, line 691, second line of Figure 6 caption. Change "FSD" to "P"

We have revised Figure 6 caption.

*Figure 6: Monthly changes of **P** simulated by the two prognostic models over the period May to July during 2000–2014. (a) Change of **P** arising from lateral melt for FSDv2-WAVE in the Chukchi Sea. (b) is same as (a) but for wave induced **P** change. (c) and (d) are same as (a)and (b) but in the Fram Strait. (e)–(h) is same as (a)–(d) but for CPOM-FSD. The blue and red box in (a) and (c) show the northern and southern region of the two study regions. Black dots indicate the location of observations in the study regions.*

Supplementary Materials

Page 2, line 53. "Based on Eq. 12 in the main text"
There is no equation 12 in the main text.

Thanks. It should be Eq. 2 in the main text. We have revised in the updated supplementary materials (Line 53 on Page 2).

*Based on Eq. 2 in the main text*